# *trim-21* promotes proteasomal degradation of CED-1 for apoptotic cell clearance in *C. elegans*

Lei Yuan[†], Peiyao Li[†], Huiru Jing, Qian Zheng, Hui Xiao*

College of Life Sciences, Shaanxi Normal University, Xi'An, China

**Abstract** The phagocytic receptor CED-1 mediates apoptotic cell recognition by phagocytic cells, enabling cell corpse clearance in *Caenorhabditis elegans*. Whether appropriate levels of CED-1 are maintained for executing the engulfment function remains unknown. Here, we identified the *C. elegans* E3 ubiquitin ligase tripartite motif containing-21 (TRIM-21) as a component of the CED-1 pathway for apoptotic cell clearance. When the NPXY motif of CED-1 was bound to the adaptor protein CED-6 or the YXXL motif of CED-1 was phosphorylated by tyrosine kinase SRC-1 and subsequently bound to the adaptor protein NCK-1 containing the SH2 domain, TRIM-21 functioned in conjunction with UBC-21 to catalyze K48-linked poly-ubiquitination on CED-1, targeting it for proteasomal degradation. In the absence of TRIM-21, CED-1 accumulated post-translationally and drove cell corpse degradation defects, as evidenced by direct binding to VHA-10. These findings reveal a unique mechanism for the maintenance of appropriate levels of CED-1 to regulate apoptotic cell clearance.

## Editor's evaluation

This article will be of high interest to scientists interested in phagocytosis, the process of removal and degradation of dead cells and pathogens. The authors identify multiple signaling components that affect the protein level of a critical phagocytosis receptor, which disrupts the degradation of dead cells. The data are extensive and overall support the conclusions of the article, providing new insight into the regulation of phagocytosis.

*For correspondence:
huixiao@snnu.edu.cn

†These authors contributed equally to this work

Competing interest: The authors declare that no competing interests exist.

## Introduction

The removal of apoptotic cells (ACs) is an integral part of the apoptotic program and is critical for tissue remodeling, suppression of inflammation, and regulation of immune responses (*Nagata, 2018*). When *Caenorhabditis elegans* cells undergo apoptosis, the activation of CED-8 by CED-3 produces phosphatidylserine (PS) on the cell surface, acting as an 'eat-me' signal that is recognized by phagocytes to activate two parallel, redundant genetic pathways (*Huang et al., 2012*; *Suzuki et al., 2013*). The *ced-2/5/12* pathway mediates the activation of the small GTPase CED-10/Rac1, leading to rearrangement of the actin cytoskeleton for cell corpse engulfment (*Wang et al., 2003*). In the *ced-1/6/7* pathway, TTR-52 binds to both exposed PS and the extracellular domain of CED-1, acting as a bridging molecule to cross-link PS with CED-1 (*Wang et al., 2010*; *Zhou et al., 2001*). CED-7, a homolog of the mammalian ABC transporter, functions in both dying and engulfing cells, is required for the enrichment of CED-1 around cell corpses (*Mapes et al., 2012*; *Wu and Horvitz, 1998*). After the phagocytic receptor CED-1 recognizes PS, the adaptor protein CED-6 (GULP) interacts with the cytoplasmic tail of CED-1 through its phosphotyrosine-binding domain (PTB) to transduce engulfment signals to downstream effectors (*Liu and Hengartner, 1998*; *Su et al., 2002*), including the large

GTPase dynamin (DYN-1) (*Yu et al., 2006*), clathrin, and clarithin adaptors AP2 or epsin (*Chen et al., 2013*; *Shen et al., 2013*), to promote actin rearrangement for the internalization of ACs. The *ced-1/6/7* and *ced-2/5/12* pathways lead to the internalization of ACs and formation of phagosomes. The phagosomes then fuse with lysosomes to form phagolysosomes that digest corpses using lysosomal acid hydrolases (*Kinchen et al., 2008*).

CED-1 shares a sequence similarity with several cell surface proteins, including multiple EGF-like domains 10 (MEGF10) (*Kay et al., 2012*) and Jedi in mammals (*Wu et al., 2009*) and Draper in *Drosophila* (*Ziegenfuss et al., 2008*), all of which have been implicated in the recognition of ACs. Our previous work showed that the CED-1 receptor is recycled from the phagosome back to the cell membrane by the retromer complex. The recycling of CED-1 by the retromer complex is critical for AC clearance as the loss of function of the retromer results in the lysosomal degradation of CED-1 (*Chen et al., 2010*). In *Drosophila* glial cells, Draper intracellular domain promotes phagocytosis through an ITAM-domain-SFK-Syk-mediated signaling cascade (*Ziegenfuss et al., 2008*). And the transcriptional factor STAT92e promotes the clearance of degenerating axonal debris by directly activating the transcriptional expression of Draper (*Musashe et al., 2016*; *Purice et al., 2016*). Recently, the transcriptional factor Serpent, a GATA factor homolog (*Waltzer et al., 2016*), was observed to be required for sufficient phagocytosis of ACs in *Drosophila* embryonic macrophages through the Draper regulation (*Kurant et al., 2008*; *Shlyakhover et al., 2018*). Our recent study also showed that Bfc serves as a Serpent co-factor to upregulate transcription of the engulfment receptor Croquemort, and thus boosts macrophage efferocytosis in response to excessive apoptosis in *Drosophila* (*Zheng et al., 2021*). These studies indicate that specific levels of engulfment receptors, including CED-1/Draper, are required for this engulfment function. Whether CED-1 is regulated by other pathway to maintain appropriate levels for executing engulfment function remains unknown.

Here, we report the identification of an E3 ubiquitin ligase, tripartite motif containing-21 (TRIM-21), as a novel regulator of cell corpse clearance. We observed that TRIM-21 acts in conjunction with UBC-21, an E2 ubiquitin-conjugating enzyme, to catalyze K48-linked poly-ubiquitination of CED-1, thus promoting CED-1 degradation. In addition, we found that tyrosine kinase SRC-1 phosphorylated Y1019 in the YXXL motif of CED-1, which was subsequently bound by the adaptor protein NCK-1. The binding of CED-1 to CED-6 or NCK-1 is required for the ubiquitination of CED-1 by TRIM-21. Moreover, we discovered that failure by TRIM-21 to degrade CED-1 led to excessive CED-1, which was sufficient to affect phagosomal acidification by binding to vacuolar-type H$^+$-ATPase subunit (VHA-10), resulting in cell corpse degradation defects. Our results reveal a novel ubiquitin-mediated regulatory mechanism that regulates the amount of CED-1 to facilitate cell corpse clearance. The expression of human TRIM21 can substitute for *C. elegans* TRIM-21 function in removing cell corpses, indicating the conserved roles of human TRIM21 in this process.

## Results

### The ubiquitin-proteasome pathway participates in the degradation of CED-1

Based on our previous work, which showed that the CED-1 receptor is recycled to the cell membrane by the Retromer complex (*Chen et al., 2010*), we sought to better understand which pathways participate in CED-1-mediated phagocytosis. To this end, we blocked the lysosome binding with chloroquine (CQ); this prevented the formation of acidic conditions required for lysosomal enzymatic function. We found that pretreatment of wild-type (WT) *C. elegans* with CQ led to a marked increase in the endogenous levels of CED-1 (*Figure 1—figure supplement 1A*). These results suggest that although CED-1 is recycled to the cell membrane during phagocytosis of the cell corpse (*Chen et al., 2010*), the remaining CED-1 still undergoes lysosome-mediated degradation.

Given that the ubiquitin-proteasome pathway is the other major protein degradation pathway in eukaryotic cells (*Varshavsky, 2017*), we next blocked this pathway, either through chemical inhibition or RNAi silencing, to test whether CED-1 homeostasis is regulated by the ubiquitin-proteasome pathway. In WT worms pre-treated with different concentrations of the proteasome inhibitor MG132, we observed that endogenous CED-1 accumulated to higher levels with increasing doses of MG132 (*Figure 1A*). We then generated *C. elegans* RNAi knocked down for either the sole E1 enzyme gene *uba-1* or the single-copy ubiquitin gene *ubq-2* and found that endogenous levels of CED-1

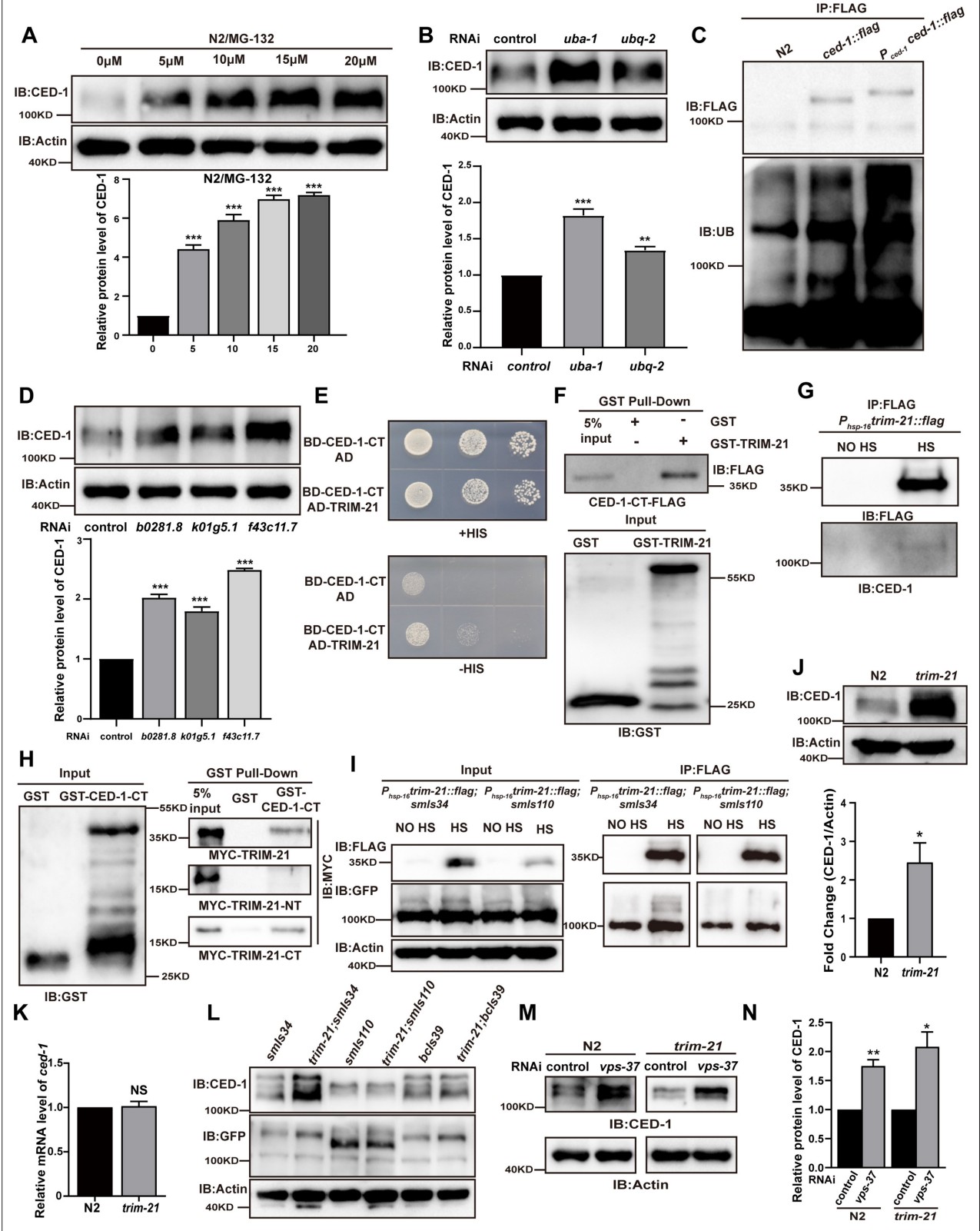

**Figure 1.** TRIM-21 is the E3 ubiquitin ligase to mediate the degradation of CED-1. (**A, B, D**) The endogenous CED-1 was examined by immunoblot analysis in N2 treated with different concentrations of MG-132 (**A**), control RNAi, *uba-1* RNAi, and *ubq-2* RNAi (**B**), control RNAi, *b0281.8* RNAi, *k01g5.1* RNAi, and *f43c11.7* RNAi (**D**). Graphs show the quantification of the protein level of CED-1. Data were from three independent experiments. (**C**) Ubiquitination of CED-1 was examined in *C. elegans*. FLAG IP was performed, followed by detection of ubiquitination with anti-ubiquitin antibodies.

*Figure 1 continued on next page*

*Figure 1 continued*

CED-1 has four isoforms; $P_{ced-1}ced-1::flag$ overexpressed CED-1 isoform a, and *ced-1::flag* was inserted as a FLAG tag into the endogenous *ced-1* locus. (E–H) The interaction between CED-1-CT and TRIM-21 was examined by yeast two-hybrid analyses (+HIS, the medium lacking Trp and Leu; -HIS, the medium lacking Trp, Leu, and His) (E), GST pull-down assays (F), FLAG IP in vivo (G), and the CED-1-CT-TRIM-21 interaction occurred through the coiled-coil domain of TRIM-21 (H). (I) FLAG-IP of worm lysates were prepared from $P_{hsp-16}trim-21::flag$ strains carrying $smIs34(P_{ced-1}ced-1::gfp)$ and $smIs110(P_{ced-1}ced-1DC::gfp)$. (J) The endogenous CED-1 was examined by immunoblot analysis in N2 and *trim-21(xhw12)* mutants. The graph shows CED-1 level. Data were from three independent experiments. (K) *ced-1* mRNA in N2 and *trim-21(xhw12)* mutants was determined by qRT-PCR. Data were from three independent experiments. (L) The exo/endogenous CED-1 expression was examined by immunoblot analyses in *trim-21(xhw12)* mutants carrying *smIs34*, *smIs110*, and $bcIs39(P_{lim-7}ced-1::gfp)$. (M) Endogenous CED-1 was examined by immunoblot analysis in N2 and *trim-21* treated with control RNAi and *vps-37* RNAi. (N) Graph shows CED-1 levels, which were quantified using ImageJ software. Data were from three independent experiments. The unpaired *t*-test was performed in this figure. *p<0.05, **p<0.01, ***p<0.001, NS, no significance. All bars indicate means and SEM.

The online version of this article includes the following source data and figure supplement(s) for figure 1:

**Source data 1.** Comparison of the levels of proteins in different samples.

**Figure supplement 1.** The coiled-coil domain of TRIM-21 interacts with the intracellular domain of CED-1.

**Figure supplement 1—source data 1.** Variation in protein levels of CED-1 after N2 treatment of chloroquine or MG-132 and CED-1-CT interacting with TRIM-21 CC domain.

**Figure supplement 2.** Sequence alignment of *C. elegans* TRIM-21 and human TRIM21, and schematic illustration of the mutation and tag insertions generated by CRISPR-Cas9.

**Figure supplement 2—source data 1.** The interactions between hTRIM21 or TRIM-21 with MEGF10-CT and hTRIM21 with CED-1-CT.

were significantly increased compared to WT controls in both of these transient suppression lines (*Figure 1B*), suggesting that the proteasomal pathway was involved in CED-1 degradation. The endogenous levels of CED-1 following treatment with MG132 were significantly higher than those after treatment with *uba-1* or *ubq-2* knockdown, likely because MG132 caused loss of proteasome function, whereas *uba-1* and *ubq-2* knockdown only decreased the ubiquitination modification. To examine the relationship between lysosome- and proteasome-dependent CED-1 degradation, we treated *C. elegans* with CQ and MG132 overlay and found that endogenous levels of CED-1 were significantly increased compared to those after CQ or MG132 treatment (*Figure 1—figure supplement 1B*). It is possible that proteasomal degradation independently affects the stability of another protein that influences lysosomal degradation of CED-1. To determine whether CED-1 could be ubiquitylated in *C. elegans*, we generated an integrated transgenic strain, $xwhIs27(P_{ced-1}ced-1::flag)$, that overexpressed CED-1-FLAG under the control of its native promoter. We also used CRISPR-Cas9 to insert a FLAG tag into the C-terminus of the endogenous *ced-1* gene locus, resulting in strain *xwh18(ced-1::flag)* (*Figure 1—figure supplement 2C*). Anti-FLAG immunoprecipitates with antibodies against ubiquitin (Ub) revealed several poly-ubiquitinated forms of CED-1 (*Figure 1C*), confirming that CED-1 was indeed poly-ubiquitinated in the worms. These observations confirmed that CED-1 acted as a substrate for poly-ubiquitylation and was degraded by the ubiquitin-proteasome pathway.

## E3 ubiquitin ligase TRIM-21 interacts with CED-1 to mediate its degradation

In order to identify which ubiquitin E3 ligase(s) could mediate the degradation of CED-1, we analyzed the endogenous level of CED-1 by Western blot (WB) screening (in triplicate) of *C. elegans* treated with individual RNAi constructs to mediate the knockdown of 170 individual ubiquitin ligases (*Kipreos, 2005*; *Table 1*). We found that the inactivation of three potential E3 ligases, *b0281.8*, *k01g5.1*, and *f43c11.7*, resulted in the accumulation of significantly higher endogenous CED-1 protein levels in all three replicates (*Figure 1D*). Among these three candidates, only *b0281.8* interacted with CED-1 in yeast two-hybrid (Y2H) assays (*Figure 1E*, *Figure 1—figure supplement 1C*). The analysis of the B0281.8 sequence showed that this E3 UB-ligase was an ortholog of human TRIM21 containing an N-terminal RING finger domain, a BBox domain, and a C-terminal α-helical coiled-coil domain (*Figure 1—figure supplement 2A and B*), which led us to designate this gene *trim-21* in *C. elegans*.

Recombinant CED-1 protein was pulled down by GST-TRIM-21 but not by GST, indicating that TRIM-21 could directly interact with CED-1 (*Figure 1F*). To further determine whether TRIM-21 also interacts with CED-1 in vivo, we generated an integrated transgenic strain $xwhIs28(P_{hsp-16}trim-21::flag)$ that expressed a TRIM-21-FLAG fusion protein under the control of a heat shock promoter and found

**Table 1.** The CED-1 protein level, regulated by RNAi of *C. elegans* genes encoding E3 ubiquitin ligases.

| *C. elegans* E3 ubiquitin ligases (RNAi) | Endogenous level of CED-1 | | |
|---|---|---|---|
| | 1st | 2nd | 3rd |
| control | - | - | - |
| T09B4.10 | + | - | ++ |
| R10A10.2 | ++ | - | - |
| T24D1.3 | + | - | - |
| Y51F10.2 | ++ | - | - |
| F10G7.10 | ++ | - | + |
| C34F11.1 | ++ | - | - |
| M110.3 | + | - | - |
| D2089.2 | + | - | - |
| R06F6.2 | + | - | - |
| B0281.8 | ++ | ++ | + |
| ZK1240.1 | + | - | ++ |
| ZK1320.6 | + | - | - |
| F43C11.8 | + | - | - |
| ZK1240.9 | + | - | - |
| F45H7.6 | ++ | - | - |
| K01G5.1 | + | ++ | ++ |
| F40G9.12 | ++ | - | - |
| M88.3 | + | - | - |
| R05D3.4 | + | - | - |
| ZK637.14 | + | - | ++ |
| F43C11.7 | + | ++ | ++ |
| C09E7.5 | ++ | ++ | - |
| T02C1.2 | + | - | - |
| Y47D3A.22 | ++ | - | - |
| Y47D3B.11 | + | - | - |
| C09E7.9 | ++ | - | - |
| K12B6.8 | ++ | ++ | - |
| T08D2.4 | + | - | - |
| Y45G12B.2 | + | - | - |
| M142.6 | ++ | ++ | - |
| C32D5.10 | ++ | ++ | - |
| C36A4.8 | ++ | - | - |

that CED-1 was immunoprecipitated from heat shock-treated worm lysates using FLAG antibodies, but not from untreated control lysates (*Figure 1G*). These results confirmed that TRIM-21 interacts with CED-1 in vivo.

To identify which regions of CED-1 and TRIM-21 were required for binding interactions, we generated a series of truncation constructs for both CED-1 and TRIM-21 (*Figure 1—figure supplement 1D*

and E), and then used Y2H and GST pull-down assays to check for the loss of interaction or binding. The results showed that the amino acid region between residues 931–1007 in the intracellular, but not extracellular, domain of CED-1 was sufficient to bind TRIM-21 (*Figure 1—figure supplement 1F*), and the coiled-coil domain of TRIM-21 was required for its interaction with CED-1 (*Figure 1H*, *Figure 1—figure supplement 1G–I*). Co-immunoprecipitation (co-IP) assays using an integrated *C. elegans* strain *smIs34* expressing a full-length CED-1::GFP fusion protein under the native CED-1 promoter and a second integrated transgenic line, *smIs110,* which expressed a GFP-fused CED-1 with the C-terminal region deleted (CED-1ΔC::GFP), also driven by the native promoter, revealed that TRIM-21 interacted with the full-length CED-1, but not with the CED-1 variant harboring a C-terminal deletion (*Figure 1I*). These data demonstrated that CED-1-TRIM-21 binding was mediated by the interaction between the CED-1 intracellular domain and the TRIM-21 coiled-coil domain. Interestingly, we found that hTRIM21 interacted with MEGF10-CT and CED-1-CT, whereas TRIM-21 interacted with CED-1-CT and MEGF10-CT, suggesting that TRIM-21 is an ancient E3 ligase in which the coiled-coil domain recognizes its substrate CED-1 (*Figure 1—figure supplement 2D and E*).

To further investigate the structural basis of TRIM-21 binding, we used CRISPR-Cas-9 to generate a mutant allele of *trim-21* (*xwh12*) with a nonsynonymous mutation that led to a premature stop codon at the N-terminus in WT worms (*Figure 1—figure supplement 2C*). WB analysis indicated that CED-1 protein accumulated to significantly higher levels in worms carrying the *trim-21*(*xwh12*) allele than in those carrying the WT allele (*Figure 1J*), but the transcription level of CED-1 was unaltered (*Figure 1K*), which further suggested that TRIM-21 was required for CED-1 degradation. We then crossed the *trim-21*-bearing strain *xwh12* with *smIs34* (expressing CED-1::GFP driven by the native promoter), *bcIs39* (expressing sheath cell-specific CED-1::GFP under the *lim-7* promoter), and *smIs110* (expressing CED-1ΔC::GFP). We observed that full-length CED-1::GFP in *smIs34* and *bcIs39*, but not CED-1ΔC::GFP in *smIs110*, were higher in the progeny carrying *trim-21*(*xwh12*) than in the controls (*Figure 1L*), which demonstrated that the CED-1 intracellular domain was required for its binding and degradation by TRIM-21. Additionally, the endogenous levels of CED-1 were higher in *trim-21; smIs34* and slightly higher in *trim-21; bcIs39* but unchanged in *trim-21; smIs110* as compared to the control (*Figure 1L*). This may be because *smIs34*, *smIs110*, and *bcIs39* are three transgenic integrated strains containing different copy numbers of extrachromosomal arrays integrated into the chromosome. This may lead to differences in the levels of CED-1 overexpression in *smIs34* and *bcIs39*. In *bcIs39*, exogenous CED-1 may be expressed at a higher level than that in *smIs34*, possibly masking the ability of TRIM-21 to degrade endogenous CED-1. In *smIs110,* CED-1ΔC::GFP (a GFP-fused CED-1 with the C-terminal region deleted) was still capable of recognizing neighboring ACs, suggesting that it competes with endogenous CED-1 for signal transduction, including recruiting TRIM-21 for its degradation. We next examined whether lysosomal pathway degradation of CED-1 was affected by *trim-21*. Following knockdown of VPS-37, which blocks lysosomal degradation of CED-1 (*Chen et al., 2010*), in WT worms and the *trim-21* mutant, the endogenous levels of CED-1 were significantly higher than those in the control (*Figure 1M and N*). Thus, TRIM-21 is an E3 ubiquitin ligase that interacted with CED-1 to mediate its degradation independently of lysosome-dependent degradation of CED-1.

## UBC-21 and TRIM-21 together mediate CED-1 poly-ubiquitination

To identify which E2 ubiquitin-conjugating enzyme(s) could mediate CED-1 degradation, we elucidated the endogenous level of CED-1 by WB screening (in triplicate) of *C. elegans* treated with 22 E2 ubiquitin-conjugating enzymes (*Kipreos, 2005*). We found that only the inactivation of one candidate E2 enzyme, *ubc-21,* led to the increased levels of endogenous CED-1 (*Table 2*) and that UBC-21 apparently interacted with TRIM-21 in Y2H assays (*Figure 2A*). GST pull-down assays showed that the recombinant TRIM-21 protein could be pulled down by GST-UBC-21, but not by GST alone (*Figure 2B*), indicating that TRIM-21 directly interacted with UBC-21 in vitro. Using TRIM-21 truncation constructs in Y2H assays, we further observed that the coiled-coil domain of TRIM-21 interacted with UBC-21 (*Figure 2C and D*). Therefore, the relevant E2 ubiquitin-conjugating enzyme that interacted with TRIM-21 was UBC-21.

To further investigate whether UBC-21 (E2) and TRIM-21 (E3) catalyzed poly-ubiquitin modification of CED-1, we performed in vitro ubiquitination assays using a CED-1 variant consisting of only the C-terminal intracellular domain (CED-1-CT) and found that it was poly-ubiquitinated by purified HA-UBQ-2, UBA-1, UBC-21, and TRIM-21. Notably, poly-ubiquitination of CED-1-CT was completely

**Table 2.** The CED-1 protein level, regulated by RNAi of *C. elegans* genes encoding E2 ubiquitin-conjugating enzymes.

| *C. elegans* E2 ubiquitin-conjugating enzymes (RNAi) | Endogenous level of CED-1 | | |
|---|---|---|---|
| | 1st | 2nd | 3rd |
| control | - | - | - |
| *ubc-1* | - | - | - |
| *ubc-2* | - | - | + |
| *ubc-3* | - | + | - |
| *ubc-6* | - | + | - |
| *ubc-7* | + | + | - |
| *ubc-8* | - | + | - |
| *ubc-9* | - | - | - |
| *ubc-12* | - | - | - |
| *ubc-13* | - | + | - |
| *ubc-14* | - | - | - |
| *ubc-15* | - | - | - |
| *ubc-16* | + | - | - |
| *ubc-17* | - | - | - |
| *ubc-18* | - | - | + |
| *ubc-19* | - | - | - |
| *ubc-20* | - | - | - |
| *ubc-21* | + | + | + |
| *ubc-22* | - | - | - |
| *ubc-23* | + | - | - |
| *ubc-24* | - | - | + |
| *ubc-25* | - | - | - |
| *ubc-26* | - | - | + |

abolished by the removal of either UBC-21 or TRIM-21 (*Figure 2E*), indicating that UBC-21 and TRIM-21 mediated the poly-ubiquitination of CED-1-CT in vitro.

To determine whether CED-1 was ubiquitinated in *C. elegans* in vivo, we inserted *flag* and *ha* tags into the endogenous *ced-1* and *ubq-2* loci using CRISPR-Cas9 to obtain the *C. elegans* strains *xwh18(ced-1::flag)* and *xwh20(ha::ubq-2)*, respectively (*Figure 1—figure supplement 2C*). We then crossed the *xwh18* and *xwh20* strains to obtain progeny carrying tagged forms of these proteins and performed anti-FLAG immunoprecipitation to test whether CED-1 was modified with UBQ-2. Antibody detection of HA in immunoprecipitation lysates obtained from the recombinant progeny revealed a smear of high-molecular mass species (*Figure 2F*), indicating that CED-1 was poly-ubiquitinated in the worms. We then introduced *ced-1::flag* and *ha::ubq-2* into the *ubc-21(xwh15)* (*Figure 1—figure supplement 2C*) and *trim-21(xwh12)* knockout strains and found that poly-ubiquitination of CED-1 was greatly reduced in both the *ubc-21(lf)* and *trim-21(lf)* mutants (*Figure 2F and H*). We found that inclusion of K48R- (conserved function in regulating protein degradation) but not K63R- (signal transduction) ubiquitin abolished poly-ubiquitination of CED-1-CT in vitro (*Figure 2E*). In addition, we generated strains carrying individual, non-synonymous K48R or K63R conversion mutations in UBQ-2 (*Figure 1—figure supplement 2C*). WB assays to detect ubiquitination of CED-1 by anti-FLAG immunoprecipitation showed that K48R-, but not K63R-, ubiquitin abolished CED-1 poly-ubiquitination in vivo (*Figure 2G and H*), indicating that CED-1 was modified by K48-linked, but not K63-linked

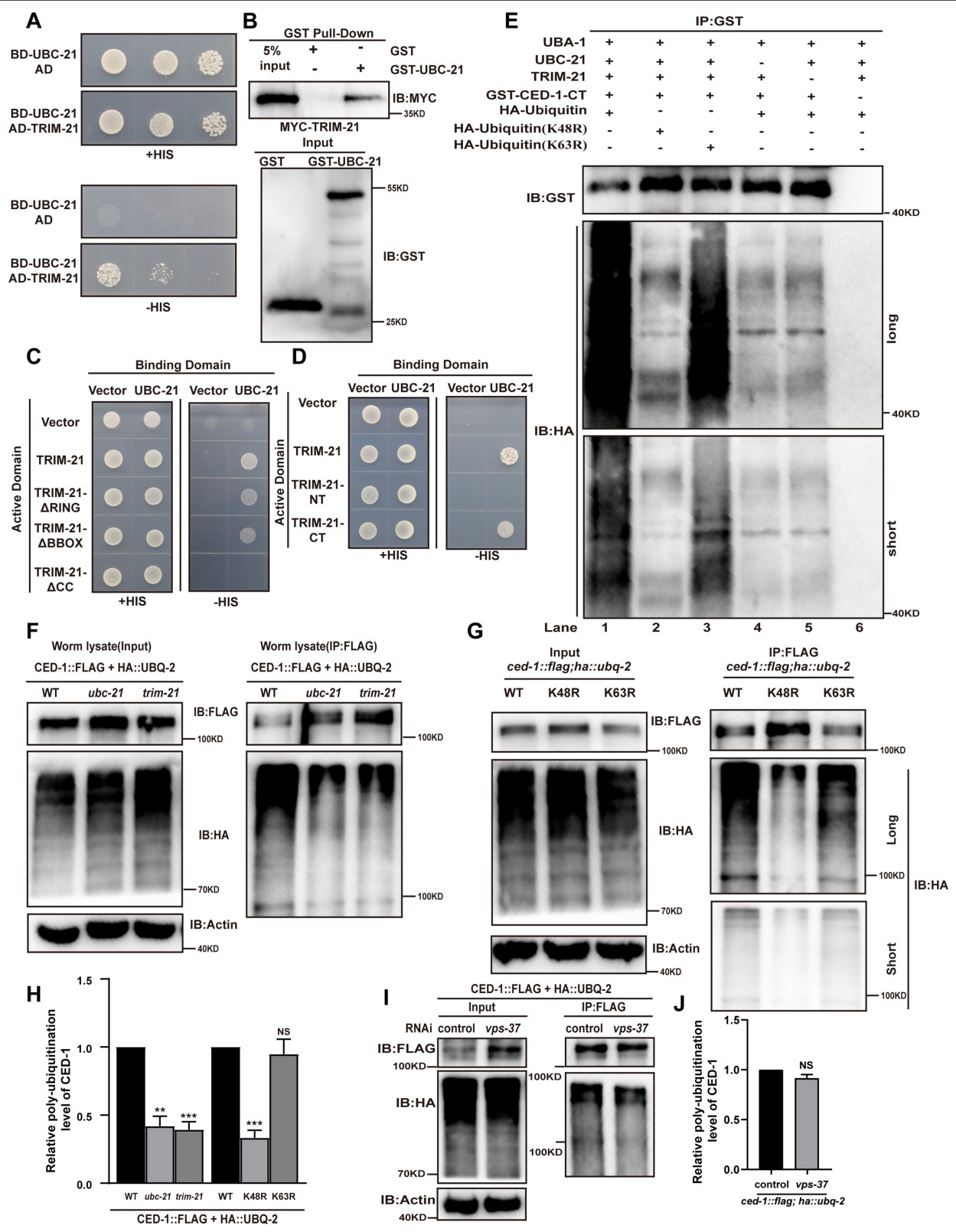

**Figure 2.** UBC-21 and TRIM-21 mediate the poly-ubiquitination of CED-1. (**A–D**) The interaction between TRIM-21 and UBC-21 was examined by yeast two-hybrid analyses (**A**), GST pull-down assays (**B**), and the UBC-21-TRIM-21 interaction occurs through the coiled-coil domain of TRIM-21 (**C, D**). (**E**) Ubiquitination of recombinant GST-CED-1-CT was examined in vitro using different forms of HA-ubiquitin (WT, K48R, K63R). GST IP was performed, followed by detection of ubiquitination with anti-HA antibodies. Ubiquitination of CED-1-CT was observed (compare lane 1 with lanes 4–6),

*Figure 2 continued on next page*

*Figure 2 continued*

and inclusion of K48R- but not K63R-ubiquitin disrupted poly-ubiquitination of CED-1-CT (compare lanes 1 and 3 with lane 2). Long and short designate long exposure time and short exposure time, respectively. (**F**) Ubiquitination of CED-1 was examined in WT, *ubc-21(xwh15),* and *trim-21(xwh12)* mutant worms carrying *ced-1::flag* and *ha::ubq-2.* FLAG IP was performed, followed by detection of ubiquitination with anti-HA antibodies. (**G**) Ubiquitination of CED-1 was examined in worms carrying both *ced-1::flag* and *ha::ubq-2*(WT, K48R, and K63R). Long and short designate long exposure time and short exposure time, respectively. (**H**) Graph of the ubiquitination level (**F, G**) quantified using ImageJ software. The ratio of ubiquitin versus CED-1 was determined and normalized to onefold in the WT. Data were from three independent experiments. (**I**) Ubiquitination of CED-1 was examined for both *ced-1::flag* and *ha::ubq-2* treated with control or *vps-37* RNAi. FLAG IP was performed, followed by detection of ubiquitination with anti-HA antibodies. (**J**) Graph shows level of ubiquitination. The ratio of ubiquitin versus CED-1 was determined and normalized to onefold in the control. Data were from three independent experiments. An unpaired *t*-test was performed in this figure. **p<0.01, ***p<0.001. All bars indicate means and SEM.

The online version of this article includes the following source data for figure 2:

**Source data 1.** The interaction between TRIM-21 and UBC-21, and the relative poly-ubiquitination level of CED-1 *in vitro* and *in vivo*.

poly-ubiquitination. We analyzed whether VPS-37 affects ubiquitination of CED-1. Ubiquitination of CED-1 by *vps-37* knockdown was similar to that in the control (***Figure 2I and J***), suggesting that lysosomal degradation of CED-1 does not affect poly-ubiquitination of CED-1. Collectively, these observations demonstrated that UBC-21 and TRIM-21 together catalyzed K48-linked poly-ubiquitination of CED-1.

## TRIM-21-mediated CED-1 degradation requires core cell death machinery, phosphatidylserine sensitivity, and the CED-7/CED-6 pathway

To investigate which signals participate in triggering the degradation of CED-1, we next examined the effects of loss of function in the components of the core cell death machinery using knockout lines with individual mutants or RNAi suppression for cell death, engulfment, and PS sensitivity. To this end, we conducted WB assays to measure the endogenous levels of CED-1 in the cell death mutants *ced-3(n717)* and *ced-4(n1162)*, the PS exposure mutant *ced-8(n1891),* the engulfment mutants *ced-7(n1892)*, *ced-6(n1813)*, *ced-2(n1994)*, and *ced-5(n1812)*, and in *ttr-52* RNAi-treated worms. The results showed that CED-1 accumulated at higher levels in the *ced-3(n717)*, *ced-4(n1162)*, *ced-8(n1891)*, *ced-7(n1892),* and *ced-6(n1813)* mutants, as well as in the *ttr-52* RNAi-treated worms (***Figure 3A***, ***Figure 3—figure supplement 1A***), indicating that the core cell death machinery, surface-exposed PS on ACs, and CED-7/CED-6 pathways were required for CED-1 degradation, whereas the *ced-2/5/12* pathway was not.

Since CED-6 acts downstream of CED-1 to transduce engulfment signals, we then detected the CED-1 levels in *dyn-1* RNAi- and *ap-2* RNAi-treated worms to test whether downstream factors of CED-6 also affected CED-1 degradation. No significant differences in CED-1 levels were observed between control RNAi-treated and *dyn-1* or *ap-2* RNAi-treated worms (***Figure 3—figure supplement 1B***), suggesting that CED-6 may perform the last step necessary for CED-1 degradation. Based on these results, we tested whether TRIM-21-mediated CED-1 degradation also required participation by CED-6. By crossing the integrated transgenic strain *xwhIs29(P$_{ced-1}$trim-21::flag)* expressing TRIM-21 under the control of the CED-1 promoter, and *xwhIs28(P$_{hsp-16}$trim-21::flag)* expressing TRIM-21 under the control of the heat shock promoter with the *ced-6(n1813)* or *ced-6(xwh25)* mutant worms (***Figure 3—figure supplement 1C***), we found that the endogenous levels of CED-1 were significantly increased in both *ced-6 (lf)* mutants (***Figure 3B***, ***Figure 3—figure supplement 1D***). These findings indicated that the CED-6 function was required for TRIM-21-mediated CED-1 degradation in engulfing cells.

## CED-1 degradation requires CED-6-mediated TRIM-21 recruitment

To address the mechanism by which CED-6 affects CED-1 degradation mediated by TRIM-21, we first tested whether CED-6 could directly interact with TRIM-21. Using GST pull-down assays, we found that recombinant CED-6 protein was effectively co-immunoprecipitated with GST-TRIM-21, but not with GST alone (***Figure 3C***), indicating that CED-6 directly interacted with TRIM-21 in vitro. We then used the *xwhIs28(P$_{hsp-16}$tirm-21::flag)* strain to further address whether CED-6 could interact with TRIM-21 in vivo. The results showed that TRIM-21 could be immunoprecipitated from lysates of heat shock-treated worms using antibodies targeting CED-6, but not from lysates of untreated controls

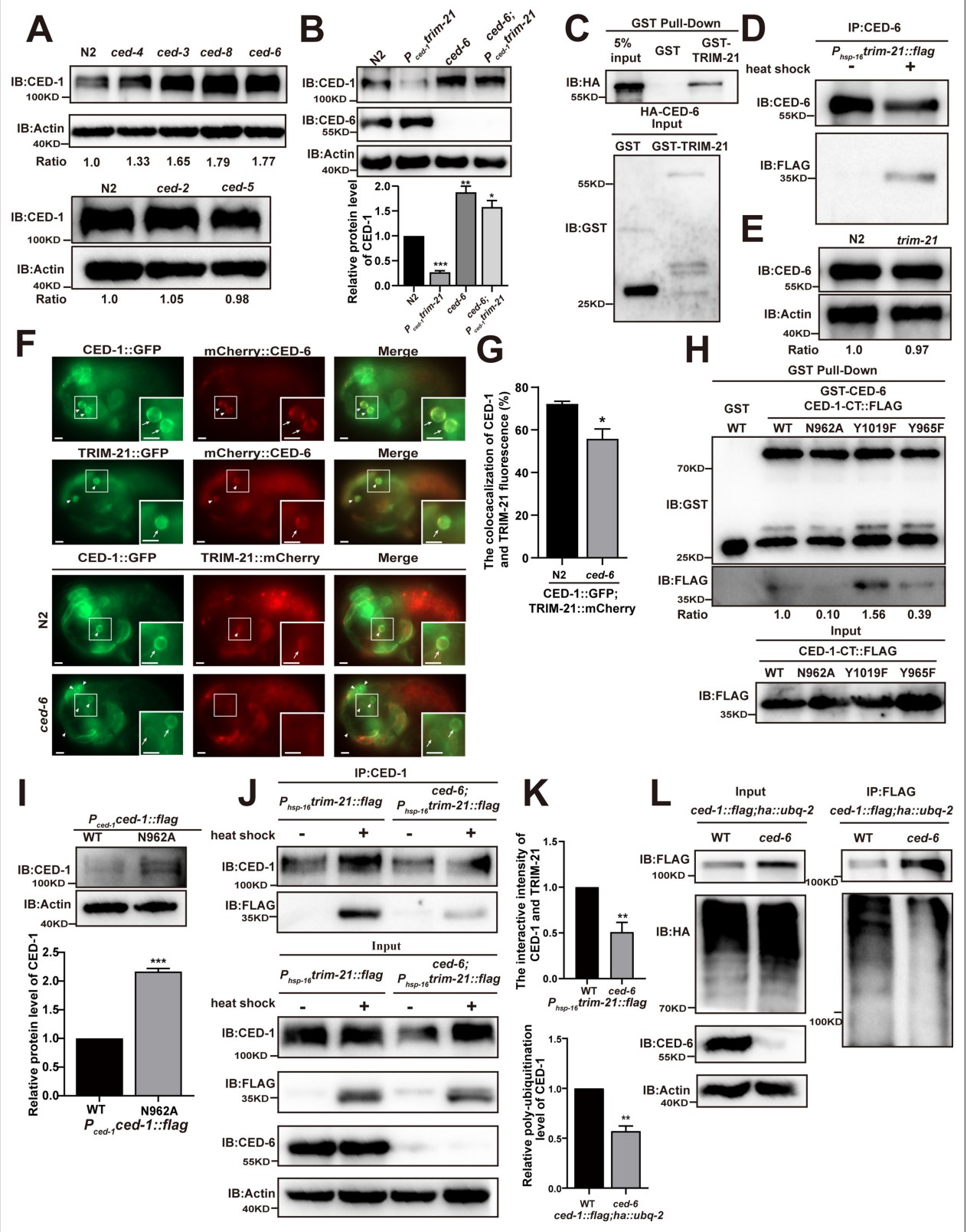

**Figure 3.** CED-6 mediates the degradation of CED-1 by recruitment of TRIM-21 to CED-1. (**A, B**) Endogenous CED-1 was examined by immunoblot analysis in N2 and different null alleles mutants (**A**), and in N2 and indicated strains (**B**). Endogenous CED-1 levels are shown at the bottom. Data were from three independent experiments. (**C, D**) The interaction between CED-6 and TRIM-21 was detected by GST pull-down assays (**C**), and CED-6 IP in vivo using *xwhls28(P_hsp-16trim-21::flag)* worms (**D**). (**E**) The endogenous CED-6 was examined by immunoblot analysis in N2 and *trim-21(xwh12)* mutants.

*Figure 3 continued on next page*

*Figure 3 continued*

The endogenous CED-6 levels were quantified at the bottom. Data were from three independent experiments. (**F**) Co-localization of CED-1::GFP and mCherry::CED-6, TRIM-21::GFP and mCherry::CED-6 in N2 embryos, CED-1::GFP and TRIM-21:: in N2 and null alleles mutant *ced-6(xwh25)* embryos. Boxed regions are magnified (2×) in insets. Bars, 2 µm. (**G**) Quantification of CED-1::GFP and TRIM-21::mCherry co-localization on cell corpses in N2 and *ced-6* mutant embryos. At least 100 cell corpses were scored for each strain and the data were repeated three times. The percentage referred to the ratio of TRIM-21::mCherry to CED-1::GFP. (**H**) The interactions between CED-6 and CED-1-CT (WT, N962A, Y1019F, and Y965F) were examined by GST pull-down assays. The quantity of GST pull-down FLAG-CED-1-CT/input FLAG-CED-1-CT is shown at the bottom. Data were from three independent experiments. (**I**) The exogenous CED-1 level in null alleles mutant *ced-1(e1735)* carrying $P_{ced-1}ced-1::flag(xwhEx34)$ and $P_{ced-1}ced-1(N962A)::flag(xwhEx35)$ is shown. The graph shows the quantification of the CED-1 protein level. Data were from three independent experiments. (**J**) CED-1 IP was performed, followed by detection of interaction between CED-1 and TRIM-21. The interaction was observed in N2 and null alleles mutant *ced-6(xwh25)*. (**K**, above) The graph shows the quantification of the protein level of TRIM-21/CED-1. The ratio of TRIM-21 versus CED-1 was determined and normalized to onefold in N2. (**L**) FLAG IP was performed, followed by detection of ubiquitination in WT and null alleles mutant *ced-6(xwh25)* carrying both *ced-1::flag* and *ha::ubq-2* with anti-HA antibodies. (**K**, below) The graph shows the relative poly-ubiquitination of CED-1. The ratio of ubiquitin versus CED-1 was determined and normalized to onefold in WT. Data were from three independent experiments. The unpaired *t*-test was performed in this figure. **p<0.01, ***p<0.001. All bars indicate means and SEM.

The online version of this article includes the following source data and figure supplement(s) for figure 3:

**Source data 1.** Related protein levels in indicated strains and related proteins interactions.

**Figure supplement 1.** The phosphotyrosine-binding domain (PTB) domain of CED-6 interacts with coiled-coil domain of TRIM-21 and NPXY motif of CED-1.

**Figure supplement 1—source data 1.** The protein level of CED-1 in indicated strains and related proteins interactions.

(*Figure 3D*). Therefore, we determined that CED-6 directly interacted with TRIM-21 in vivo. To identify the region(s) of TRIM-21 and CED-6 participating in their binding interactions, we used Y2H and GST pull-down assays to examine CED-6 interactions with various TRIM-21 truncation variants. These experiments showed that the PTB domain of CED-6 was sufficient to bind TRIM-21, whereas the coiled-coil domain of TRIM-21 was required for its interaction with CED-6 (*Figure 3—figure supplement 1E and F*). Notably, we detected no increase in the endogenous protein levels of CED-6 in *trim-21* (*xwh12*), indicating that TRIM-21 did not function as an E3 ligase in CED-6 degradation (*Figure 3E*). Furthermore, fluorescence microscopy revealed that a TRIM-21::mCherry or TRIM-21::GFP fusion reporter co-localized to the surface of cell corpses with CED-1::GFP and mCherry::CED-6 reporters in *C. elegans* (*Figure 3F*). We next examined whether CED-6 mediates TRIM-21 recruitment to the AC surface by detecting and quantifying TRIM-21::GFP recruitment to ACs in *ced-6* mutants. We found that recruitment of TRIM-21::GFP was significantly reduced in *ced-6* mutants compared to in the WT (*Figure 3—figure supplement 1G*). In addition, co-localization of CED-1::GFP and TRIM-21::mCherry was greatly decreased in *ced-6* mutants compared to that in the WT (*Figure 3F and G*). These results indicate that CED-6 mediated recruitment of TRIM-21 to the AC surface.

The intracellular domain of CED-1 contains two conserved putative tyrosine phosphorylation sites, the NPXY (residues 962–965) and YXXL motifs (residues 1019–1022), which can interact with proteins containing PTB and Src Homology 2 (SH2) domains, respectively (*Zhou et al., 2001*). Y2H assays showed that the CED-6 PTB domain was sufficient to interact with the intracellular domain of CED-1(CED-1-CT) (*Figure 3—figure supplement 1I*). Moreover, recombinant CED-6 was successfully pulled down by GST-CED-1-CT, but not by GST (*Figure 3—figure supplement 1H*). To investigate whether CED-6 is required for TRIM-21 binding to CED-1 to mediate its degradation, we first mutated the conserved residues of N962 to alanine and of Y965 to phenylalanine in the NPXY motif and introduced mutations into the $P_{ced-1}ced-1$ reporter. We then determined the number of ACs in developmental stages in embryos and found that N962A and Y965F did not attenuate the engulfment defects of *ced-1(e1735)* mutants (*Table 3*), indicating that both residues were necessary for complete engulfment of CED-1, which was consistent with the results of a previous study (*Zhou et al., 2001*).

We then asked whether the N962A and Y965F mutants of CED-1 could affect its binding to CED-6. GST pull-down and Y2H assays showed that the N962A CED-1 mutants exhibited severely impaired binding to CED-6, whereas the Y965F and Y1019F mutants of CED-1 bound successfully (*Figure 3H*, *Figure 3—figure supplement 1I*), suggesting that although the interaction occurred through CED-6 binding to the CED-1 NPXY motif, tyrosine phosphorylation was not a requirement for NPXY binding to CED-6. Next, we addressed whether N962A mutations in the NPXY domain of CED-1 affected its protein levels. We found that the CED-1 levels were higher in the N962A mutants than in the

**Table 3.** Cell corpse phenotypes in N2, *ced-1(e1735),* and overexpression of *ced-1* or *ced-1* site mutants in *ced-1(e1735)*.

| Transgene | No. of somatic cell corpses (developmental stages) | | | | | |
|---|---|---|---|---|---|---|
| | Comma | 1.5F | 2F | 2.5F | 3F | 4F |
| N2 (-) | 9.73 ± 0.47 | 12.27 ± 0.48 | 11.40 ± 0.39 | 6.67 ± 0.29 | 2.6 ± 0.32 | 0.67 ± 0.12 |
| *ced-1(e1735)* | 21.67 ± 0.85 *** | 28.80 ± 0.79 *** | 34.73 ± 0.99 *** | 34.20 ± 1.45 *** | 31.20 ± 1.84 *** | 30.93 ± 1.23 *** |
| $P_{ced-1}$ced-1 line 1 / ced-1(e1735) | 9.73 ± 0.36 NS | 12.47 ± 0.28 NS | 9.00 ± 0.30 NS | 8.20 ± 0.72 NS | 2.07 ± 0.43 NS | 0.93 ± 0.31 NS |
| $P_{ced-1}$ced-1 line 2 / ced-1(e1735) | 9.47 ± 0.31 NS | 11.73 ± 0.31 NS | 11.00 ± 0.74 NS | 7.87 ± 0.53 NS | 2.53 ± 0.48 NS | 1.07 ± 0.18 NS |
| $P_{ced-1}$ced-1 line 3 / ced-1(e1735) | 10.73 ± 0.31 NS | 11.87 ± 0.58 NS | 11.40 ± 0.78 NS | 6.40 ± 0.46 NS | 1.73 ± 0.33 NS | 0.53 ± 0.26 NS |
| $P_{ced-1}$ced-1(N962A) line 1/ced-1(e1735) | 23.60 ± 0.51 *** | 32.00 ± 0.99 *** | 35.73 ± 0.71 *** | 31.53 ± 1.14*** | 30.87 ± 1.60*** | 30.93 ± 1.05*** |
| $P_{ced-1}$ced-1(N962A) line 2/ced-1(e1735) | 23.93 ± 0.38 *** | 30.40 ± 1.12*** | 35.93 ± 1.19 *** | 33.47 ± 1.11 *** | 27.13 ± 0.80*** | 30.93 ± 1.46*** |
| $P_{ced-1}$ced-1(N962A) line 3/ced-1(e1735) | 22.40 ± 0.49 *** | 32.40 ± 1.07 *** | 34.80 ± 0.99 *** | 33.07 ± 0.71*** | 32.00 ± 0.91*** | 32.00 ± 0.60*** |
| $P_{ced-1}$ced-1(Y965F) line 1/ced-1(e1735) | 12.73 ± 0.89 ** | 16.67 ± 0.70 *** | 18.53 ± 1.61 *** | 12.67 ± 0.62*** | 11.00 ± 1.82 *** | 3.33 ± 0.50 *** |
| $P_{ced-1}$ced-1(Y965F) line 2/ced-1(e1735) | 15.67 ± 0.50 *** | 18.20 ± 0.48 *** | 21.00 ± 0.80 *** | 11.27 ± 0.69 *** | 7.87 ± 0.52 *** | 2.73 ± 0.33 *** |
| $P_{ced-1}$ced-1(Y965F) line 3/ced-1(e1735) | 12.07 ± 0.73 * | 15.00 ± 0.52 *** | 17.27 ± 1.54 ** | 11.33 ± 0.51 *** | 17.27 ± 1.54*** | 3.27 ± 0.47 *** |
| $P_{ced-1}$ced-1(Y1019F) line 1/ced-1(e1735) | 16.47 ± 0.69 *** | 19.13 ± 0.70 *** | 20.27 ± 0.61 *** | 11.87 ± 0.63 *** | 10.07 ± 0.89*** | 3.07 ± 0.35 *** |
| $P_{ced-1}$ced-1(Y1019F) line 2/ced-1(e1735) | 15.07 ± 0.43 *** | 16.60 ± 0.51 *** | 17.80 ± 0.54 *** | 12.13 ± 0.53*** | 5.53 ± 0.55 *** | 2.13 ± 0.21 *** |
| $P_{ced-1}$ced-1(Y1019F) line 3/ced-1(e1735) | 14.33 ± 0.29 *** | 19.73 ± 0.49 *** | 20.53 ± 0.79 *** | 11.60 ± 0.57 *** | 8.20 ± 0.46 *** | 3.00 ± 0.37 *** |

At least 15 embryos were scored at each stage for each strain. *p<0.05, **p<0.01, ***p<0.001, NS, no significance.

The online version of this article includes the following source data for table 3:

**Source data 1.** The number of somatic different developmental stages cell corpses in indicated strains.

WT CED-1 controls (*Figure 3I*). In addition, we found that the binding of TRIM-21 to CED-1 was substantially reduced in the *ced-6(xwh25)* background (*Figure 3J and K*). Moreover, we found that the poly-ubiquitination of CED-1 was also reduced in *ced-6(xwh25)* mutants (*Figure 3K and L*). These results indicated that CED-6 mediated TRIM-21 binding interactions with CED-1 to ensure CED-1 degradation in vivo.

## CED-1 degradation requires phosphorylation of its YXXL motif by SRC-1

Since the two putative tyrosine phosphorylation sites, YXXL and NPXY, in the CED-1 intracellular domain have been reported to be partially redundant for the engulfment of CED-1 (*Zhou et al., 2001*), we next sought to identify which tyrosine kinase(s) could mediate phosphorylation of these motifs. To screen tyrosine kinases in *C. elegans*, we generated individual RNAi knockdowns of 90 potential tyrosine kinases and screened for germline cell death in three replicate experiments. We found that the inactivation of 13 candidate tyrosine kinases increased the cell corpse number in the germline (*Table 4*). Among these, SRC-1 was the only CED-1-CT-interacting tyrosine kinase identified by the Y2H assay (*Figure 4A*). Furthermore, we found that the recombinant SRC-1 protein could be pulled down by GST-CED-1-CT, but not by GST alone (*Figure 4B*), indicating that SRC-1 directly interacted with the CED-1 C-terminal intracellular domain. Tagging CED-1-CT with TurboID ligase,

**Table 4.** Cell corpse phenotypes caused by RNAi of *C. elegans* genes encoding tyrosine kinases.

| *C. elegans* tyrosine kinases (RNAi) | No. of germ cell corpses (mean ± SEM) | *C. elegans* tyrosine kinases (RNAi) | No. of germ cell corpses (mean ± SEM) |
|---|---|---|---|
| Control | 2.667 ± 0.2425 | T06C10.6 | 2.478 ± 0.4484 |
| F49B2.5 | 4.459 ± 0.3435 | T13H10.1 | 4.854 ± 0.2828** |
| Y47G6A.5 | 3.735 ± 0.2873 | T25B9.4 | 4.577 ± 0.385 |
| Y48G1C.10 | 2.806 ± 0.2948 | W01B6.5 | 1.75 ± 0.3096 |
| C35E7.10 | 3.143 ± 0.5084 | Y4C6A.k | 2.75 ± 0.3708 |
| F22D6.1 | 2.529 ± 0.5363 | ZK593.9 | 1.188 ± 0.2453 |
| F23C8.7 | 2.235 ± 0.5391 | T25B9.5 | 2.063 ± 0.17 |
| F26E4.5 | 2.286 ± 0.3097 | W08D2.8 | 4.545 ± 0.2995 |
| F53G12.6 | 3.559 ± 0.3409 | Y69E1A.3 | 2.063 ± 0.2657 |
| F59A3.8 | 2.619 ± 0.4654 | F11E6.8 | 1.5 ± 0.2739 |
| T21G5.1 | 1.933 ± 0.4306 | T22B11.4 | 1.111 ± 0.1962 |
| ZC581.7 | 1.529 ± 0.2443 | Y116A8C.24 | 1.313 ± 0.2846 |
| W04G5.6 | 2.4 ± 0.3352 | T08G5.2 | 3.879 ± 0.3191 |
| C34F11.5 | 2.333 ± 0.2323 | M01B2.1 | 3.591 ± 0.3984 |
| F46F5.2 | 1.733 ± 0.3157 | T01G5.1 | 2.688 ± 0.3125 |
| M176.9 | 1.4 ± 0.3055 | C16D9.2 | 3.105 ± 0.4319 |
| R05H5.4 | 4.829 ± 0.4056** | C24G6.2 | 2.125 ± 0.482 |
| Y62F5A.10 | 2 ± 0.3086 | F40A3.5 | 1.789 ± 0.4811 |
| ZK622.1 | 4 ± 0.6249 | T10H9.2 | 4.05 ± 0.397 |
| C08H9.5 | 5.275 ± 0.4236*** | Y38H6C.20 | 1.842 ± 0.3356 |
| C08H9.8 | 3.826 ± 0.469 | F09G2.1 | 2 ± 0.2425 |
| M176.6 | 4.741 ± 0.4356* | B0302.1 | 4.571 ± 0.3864 |
| M176.7 | 1.579 ± 0.2791 | D1073.1 | 1.5 ± 0.2415 |
| R09D1.12 | 3.969 ± 0.4804 | M79.1 | 5.172 ± 0.4915** |
| R09D1.13 | 2.438 ± 0.3287 | F59F5.3 | 2 ± 0.3162 |
| ZK938.5 | 5.515 ± 0.4809*** | B0198.3 | 1.818 ± 0.3872 |
| B0252.1 | 2.04 ± 0.3628 | F54F7.5 | 2 ± 0.3208 |
| C01G6.8 | 3.063 ± 0.359 | C16B8.1 | 1.938 ± 0.335 |
| M03A1.1 | 3 ± 0.3291 | C25F6.4 | 3.192 ± 0.4039 |
| B0523.1 | 4.188 ± 0.4002 | F11D5.3 | 1.813 ± 0.2617 |
| F57B9.8 | 1.563 ± 0.3412 | F58A3.2 | 1.938 ± 0.2495 |
| W03A5.1 | 2.875 ± 0.2869 | F59F3.1 | 2.063 ± 0.359 |
| C15H7.3 | 4.293 ± 0.3479 | F59F3.5 | 3.375 ± 0.3146 |
| T17A3.8 | 4.273 ± 0.4661 | T14E8.1 | 2.188 ± 0.3788 |
| R151.1 | 3.103 ± 0.2595 | F08F1.1 | 2.438 ± 0.3158 |
| C01C7.1 | 2.438 ± 0.4741 | F09A5.2 | 2.625 ± 0.3637 |
| C18H7.4 | 5.093 ± 0.4046*** | ZK1067.1 | 4.9 ± 0.2969** |
| C25A8.5 | 2.7 ± 0.4872 | C30F8.4a | 3.313 ± 0.3619 |

*Table 4 continued on next page*

*Table 4 continued*

| *C. elegans* tyrosine kinases (RNAi) | No. of germ cell corpses (mean ± SEM) | *C. elegans* tyrosine kinases (RNAi) | No. of germ cell corpses (mean ± SEM) |
|---|---|---|---|
| *C55C3.4* | 2.85 ± 0.4881 | *M142.1* | 3.313 ± 0.3502 |
| *F01D4.3* | 4.188 ± 0.366 | *Y55D5A.5a.2* | 1.188 ± 0.4002 |
| *F22B3.8* | 8.643 ± 0.8517*** | *T17A3.1* | 2.5 ± 0.3028 |
| *K07F5.4* | 3.95 ± 0.3507 | *W02A2.6* | 3.063 ± 0.193 |
| *K09B11.5* | 5.532 ± 0.3231*** | *Y50D4B.6* | 2.813 ± 0.3191 |
| *R11E3.1* | 2.565 ± 0.4066 | *T22B11.4* | 5.522 ± 0.6188*** |
| *T04B2.2* | 5.892 ± 0.3948*** | *Y92H12A.1* | 9.625 ± 0.6575*** |
| *T06C10.3* | 2.579 ± 0.3182 | | |

At least 15 adult worms were scored for each RNAi treatment. *$p<0.05$, **$p<0.01$, ***$p<0.001$.

The online version of this article includes the following source data for table 4:

**Source data 1.** Germ cell corpses in N2 treated with tyrosine kinases RNAi.

an enzyme-catalyzed proximity biotin label for detecting direct protein interaction, further confirmed that the CED-1 C-terminal domain could interact with SRC-1 (**Figure 4C**). Subsequent Y2H assays indicated that both the SH2 and SH3 domains of SRC-1 were required for interaction with the CED-1 intracellular domain (**Figure 4D**, **Figure 4—figure supplement 1A**).

In *Drosophila*, the Src family kinase Src42A significantly increases Draper phosphorylation and is essential for glial cell phagocytic activity (**Ziegenfuss et al., 2008**). To test whether SRC-1 indeed mediated CED-1 tyrosine phosphorylation, we performed in vitro kinase assays by incubating CED-1-CT-FLAG with SRC-1 in a reaction mixture containing ATP. Using the Pro-Q Diamond Phosphoprotein Gel Stain, we detected the phosphorylation signal when CED-1-CT-FLAG was incubated with SRC-1, but not in control samples (**Figure 4—figure supplement 1B**). We then used an anti-phosphotyrosine (anti-P-TYR-100) antibody to confirm that the tyrosine phosphorylation site in CED-1-CT was phosphorylated by SRC-1 (**Figure 4—figure supplement 1B**). In *Drosophila*, the transcriptional factor STAT92e promotes clearance of degenerating axonal debris in the glia by directly activating the transcriptional expression of Draper (**Musashe et al., 2016**; **Purice et al., 2016**). We found that knockdown of *sta-2*, a homolog of *STAT92e* in *C. elegans*, led to accumulation of cell corpses in the germline and decreased expression of CED-1 at the transcriptional and translational levels (**Figure 4—figure supplement 1C–E**), indicating that conserved STAT92e in *Drosophila* promotes transcriptional expression of Draper. In light of these results, we next investigated whether SRC-1 was required for CED-1 degradation. We used CRISPR-Cas9 to generate a mutant allele of *src-1(xwh26)*, in which a premature stop codon was introduced at the N-terminus in WT worms (**Figure 4—figure supplement 1F**). We found that endogenous levels of CED-1 were higher in the *src-1(xwh26)* knockout worms than those in the control (**Figure 4E**). Moreover, we found that poly-ubiquitination of CED-1 was attenuated in *src-1*-silenced worms (**Figure 4F**). These results confirmed that SRC-1 was required for CED-1 degradation via the proteasome pathway. Moreover, we observed that *src-1* knockdown led to increased cell corpse numbers in both *ced-2(n1994)* and *ced-1(e1735)* mutants (**Figure 4G and H**), suggesting that *src-1* did not act specifically within either pathway to regulate cell corpse removal.

To identify the specific sites in the CED-1 intracellular domain (CED-1-CT) phosphorylated by SRC-1, we performed in vitro kinase assays, excising the CED-1-CT bands from the SDS-PAGE gel and purifying them for analysis by liquid chromatography–tandem mass spectrometry (LC-MS/MS). This assay revealed that one amino acid had substantially greater phosphorylation than the other sites: Tyr-1019 (**Figure 4—figure supplement 1G**). Notably, this residue was the same as the predicted tyrosine phosphorylation site in the YXXL motif (residues 1019–1022) of the CED-1 intracellular domain (**Zhou et al., 2001**). To test the effects of abolishing this site, we generated a non-synonymous mutation that converted the conserved YXXL tyrosine residue (Y1019) to phenylalanine and found that the Y1019F variant did not rescue the engulfment defects observed in *ced-1(e1735)* mutants (**Table 3**). These results indicated that this residue was necessary for the engulfment activity mediated by CED-1, which

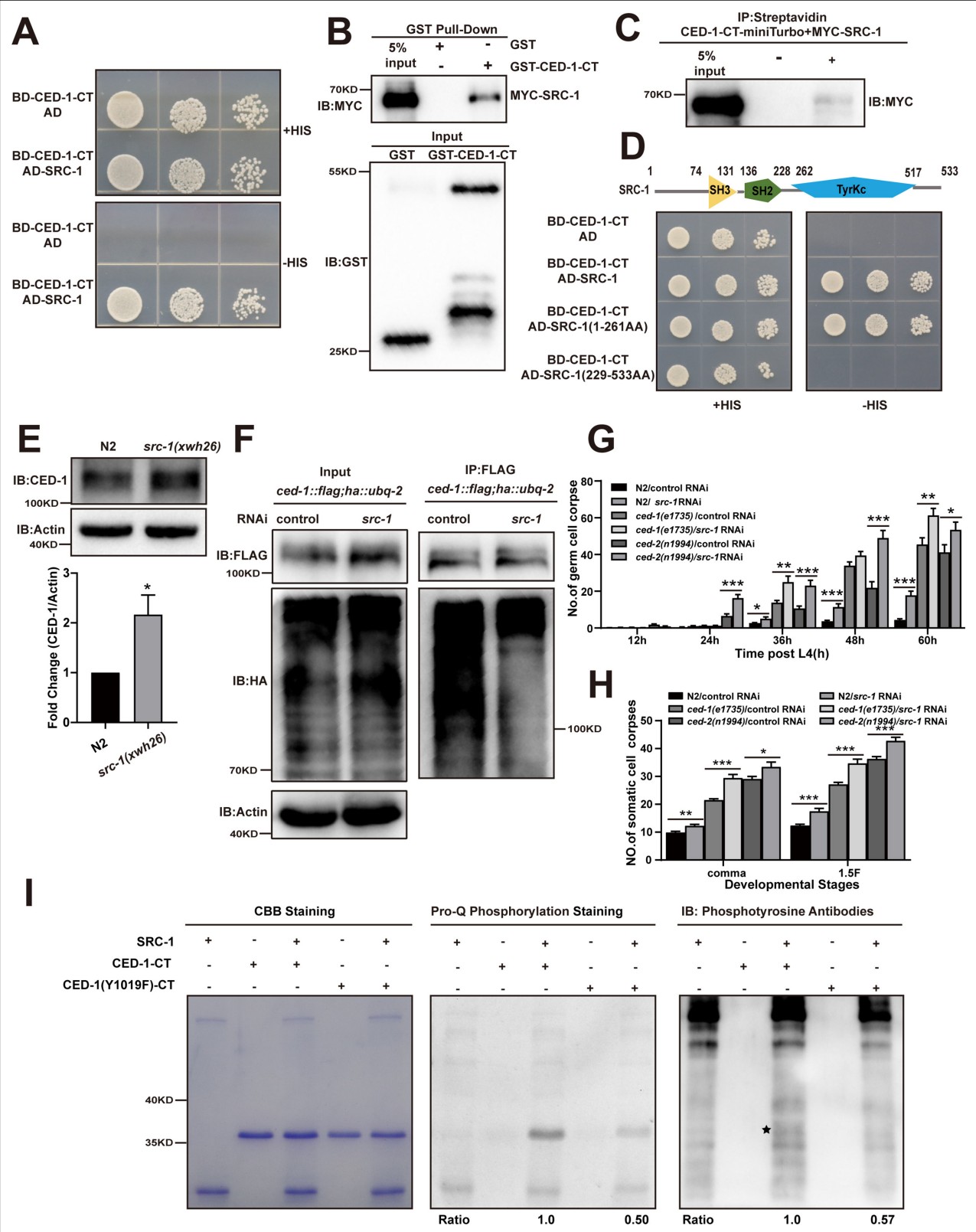

**Figure 4.** The phosphorylation of YXXL motif in CED-1 by SRC-1 is required for CED-1 degradation. (**A–D**) The interaction between CED-1-CT 1617 and SRC-1 was examined by yeast two-hybrid (Y2H) analyses (**A**), GST pull-down assays (**B**), co-IP by 0.5 mM biotin in 293T cells (**C**), and the CED-1-SRC-1 interaction occurs through the SH3 and SH2 domain of SRC-1 in Y2H (**D**). (**E**) The endogenous CED-1 level was detected in N2 and *src-1(xwh26)*. The graph shows quantification of the protein level of CED-1. (**F**) Ubiquitination of CED-1 was examined in worms carrying both *ced-1::flag* and *ha::ubq-2*

*Figure 4 continued on next page*

Figure 4 continued

treated with control or *src-1* RNAi. FLAG IP was performed, followed by detection of ubiquitination with anti-HA antibodies. (**G, H**) Different stages (hr post L4) of germ cell corpses (**G**) and comma, 1.5F stage embryo corpses (**H**) were quantified (mean ± SEM) in indicated strains, in which *ced-1(e1735)* and *ced-2(n1994)* are null alleles mutants. Fifteen adult worms or embryos were scored at each stage for each strain. (**I**) Phosphorylation of CED-1-CT (WT or Y1019F) by SRC-1 was analyzed using CBB staining (left), Pro-Q phosphorylation staining (middle) and phosphotyrosine antibody (right). * shows phosphotyrosine bands. Quantities of Pro-Q phosphorylation staining CED-1-CT level/CBB staining CED-1-CT level and anti-phosphotyrosine CED-1 level/CBB staining CED-1-CT level are shown at the bottom. Data were from three independent experiments. An unpaired *t*-test was performed. *$p<$ 0.05, **$p<0.01$, ***$p<0.001$. All bars indicate means and SEM.

The online version of this article includes the following source data and figure supplement(s) for figure 4:

**Source data 1.** Related protein levels in indicated strains, related proteins interactions and phosphorylation of CED-1-CT *in vitro*.

**Figure supplement 1.** The YXXL motif in CED-1 is phosphorylated by SRC-1.

**Figure supplement 1—source data 1.** Related protein levels and poly-ubiquitination in indicated strains, mRNA levels of *ced-1* and germ cell corpses in N2 treated with *sta-2* RNAi, related proteins interactions and phosphorylation of CED-1-CT *in vitro*.

is consistent with the findings of a previous study (*Zhou et al., 2001*). We then asked whether the Y1019F mutation affected CED-1 phosphorylation by SRC-1. We found less Pro-Q Diamond staining and decreased phosphorylation of CED-1, assessed by anti-PY antibody, when the CED-1(Y1019F) variant was incubated with SRC-1, thus confirming that SRC-1 phosphorylated CED-1 at the Y1019 tyrosine residue (*Figure 4I*).

We then sought to determine whether the Y1019F mutation in CED-1 also affected its binding to CED-6 or TRIM-21. Y2H and GST pull-down assays showed that the CED-1(Y1019F) variant could still bind to CED-6 or TRIM-21 (*Figure 3H*, *Figure 4—figure supplement 1H and I*), which suggested that tyrosine phosphorylation was not required for YXXL-mediated binding to CED-6 or TRIM-21. Next, to address whether mutations in the N962 and Y1019 residues affected CED-1 protein stability, we introduced N962A(*xwh21*) and Y1019F(*xwh22*) mutations individually into the *C. elegans* genome at the *ced-1* locus in the *ced-1::flag(xwh18)* strain (*Figure 1—figure supplement 2C*). We found that the degradation rates of the N962A and Y1019F CED-1 mutants were lower than those of WT CED-1 (*Figure 4—figure supplement 1J and K*). Moreover, we found that poly-ubiquitination of CED-1 was also reduced in the N962A and Y1019F mutants (*Figure 4—figure supplement 1L and M*). These results confirmed that SRC-1 phosphorylated the tyrosine residue of the CED-1 YXXL motif, which is required for CED-1 degradation.

## The SH2 domain-containing adaptor protein NCK-1 is required for CED-1 degradation

Previous studies have proposed that the tyrosine sites in the YXXL motif (residues 1019–1022) of the CED-1 intracellular domain are phosphorylated and subsequently bound by an adaptor protein containing the SH2 domain, which shares a partially redundant function with another PTB domain protein in CED-1-mediated cell corpse clearance (*Zhou et al., 2001*). Interestingly, we found that SRC-1 function was partially redundant with that of CED-6 in CED-1 degradation (*Figure 5A*). To identify the adaptor protein containing an SH2 domain that could bind to the YXXL motif of CED-1, we generated RNAi knockdowns of 60 SH2 domain proteins and screened for germline cell death in three replicate experiments. This screen revealed that the inactivation of *nck-1* increased the number of germline corpses in all three experiments (*Table 5*). We also found that NCK-1 interacted with CED-1-CT in Y2H assays (*Figure 5B*) and that recombinant NCK-1 protein was efficiently pulled down by GST-CED-1-CT and GST-TRIM-21, but not by GST (*Figure 5C*), indicating that NCK-1 could directly interact with the CED-1 intracellular domain and TRIM-21. These results confirmed co-IP with CED-1-CT or TRIM-21 tagged with TurboID ligase (*Figure 5D and E*). In addition, we found that the endogenous levels of CED-1 were increased, whereas CED-1 poly-ubiquitination was reduced in *nck-1* knockdown worms (*Figure 5F and G*).

We observed that *nck-1* RNAi also resulted in increased cell corpse counts in *xwhIs49(rde-1; P_{ced-1}::rde-1)* transgenic worms with phagocyte-specific RNAi suppression (*Figure 5H*), suggesting that *nck-1* functions in cell corpse removal. Moreover, we found that *nck-1* RNAi increased cell corpse numbers in both *ced-2(n1994)* and *ced-1(e1735)* mutants, suggesting that *nck-1* does not act specifically within either pathway to regulate cell corpse removal (*Figure 5I– and J*). To further confirm these

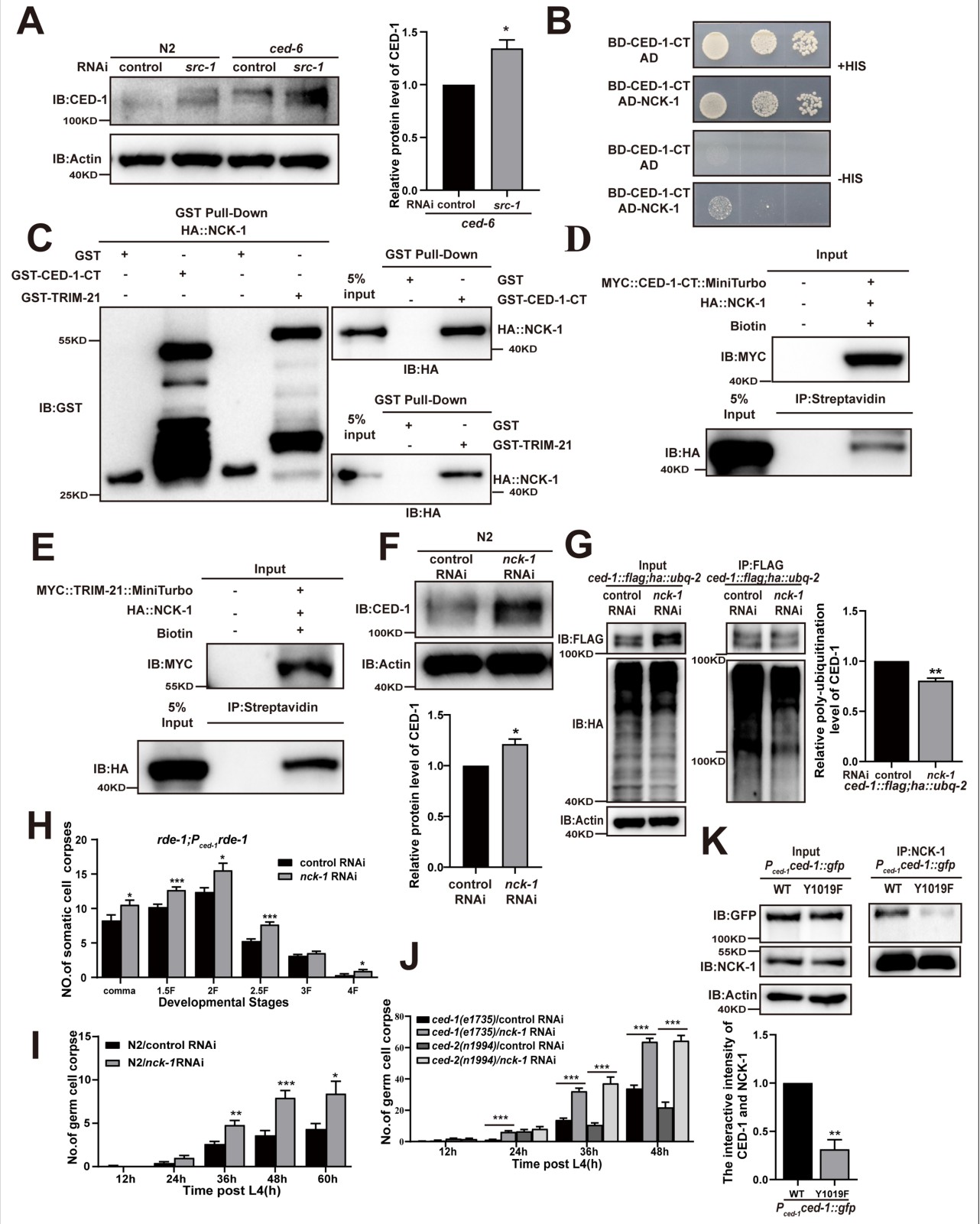

**Figure 5.** The adaptor NCK-1 is required for CED-1 degradation. (**A**) The endogenous CED-1 was examined by immunoblot analysis in N2 and null alleles mutant *ced-6(n1813)* treated with control or *src-1* RNAi. The graph shows the quantification of the level of CED-1 in *ced-6(n1813)* treated with control and *src-1* RNAi. Data were from three independent experiments. (**B–E**) The interactions between CED-1-CT–NCK-1 were examined by yeast two-hybrid (Y2H) (**B**), CED-1-CT–NCK-1 and TRIM-21–NCK-1 levels were detected by GST pull-down assays (**C**), co-IP by 0.5 mM biotin in 293T cells (**D**,

*Figure 5 continued on next page*

*Figure 5 continued*

E). (**F**) The endogenous CED-1 was examined by immunoblot analysis in N2 treated with control or *nck-1* RNAi. The graph shows quantification of the level of CED-1. (**G**) Ubiquitination of CED-1 was examined in worms carrying both *ced-1::flag* and *ha::ubq-2* treated with control or *nck-1* RNAi. FLAG IP was performed, followed by detection of ubiquitination with anti-HA antibodies. The graph shows quantification of the level of ubiquitination. The ratio of ubiquitin versus CED-1 was determined and normalized to onefold in the control. Data were from three independent experiments. (**H–J**) The embryonic or gonadal cell corpses were quantified in the indicated strains treated with control or *nck-1* RNAi, the development stages of embryo cell corpses in tissue-specific expression strain *rde-1; P$_{ced-1}$rde-1* (**H**), the germ cell corpses with different stages post-L4 in N2 (**I**), and null alleles mutant *ced-1(e1735)* and null alleles mutant *ced-2(n1994)* (**J**). 15 adult worms or embryos were scored at each stage for each strain. (**K**) NCK-1 IP was performed, followed by detection of the interaction between CED-1 (WT, Y1019F) and NCK-1 in *P$_{ced-1}$ced-1::gfp (WT, Y1019F)* worms with anti-GFP antibodies. The graph shows the protein level of CED-1/NCK-1. The ratio of CED-1 versus NCK-1 was determined and normalized to onefold in N2. Data were from three independent experiments. An unpaired *t*-test was performed in this figure. *p<0.05, **p<0.01, ***p<0.001. All bars indicate means and SEM.

The online version of this article includes the following source data and figure supplement(s) for figure 5:

**Source data 1.** Related protein levels and poly-ubiquitination in indicated strains, cell corpses in indicated strains and related proteins interactions.

**Figure supplement 1.** Role of NCK-1 in CED-1 degradation.

**Figure supplement 1—source data 1.** Related protein levels and poly-ubiquitination in indicated strains and germ cell corpses in *nck-1* mutants.

findings, we used CRISPR-Cas9 to generate a mutant allele of *nck-1(xwh51)* in which a premature stop codon was introduced at the N-terminus in WT worms (*Figure 5—figure supplement 1A*). We found that the endogenous levels of CED-1 were increased (*Figure 5—figure supplement 1B*), as were the cell corpse numbers, (*Figure 5—figure supplement 1C*) in *nck-1(xwh51)* mutants. In addition, the binding of TRIM-21 to CED-1 was substantially reduced in the *nck-1(xwh51)* background (*Figure 5—figure supplement 1D and E*). Moreover, CED-1 poly-ubiquitination was reduced in *nck-1(xwh51)* worms (*Figure 5—figure supplement 1E and F*). To investigate whether phosphorylation of CED-1 mediated its interaction with NCK-1, we created a Y1019F mutant of CED-1 (cannot be phosphorylated) under control of the CED-1 promoter. The interaction of NCK-1 to CED-1 was significantly reduced in Y1019F mutants compared to in the WT (*Figure 5K*), indicating that phosphorylation of CED-1 is required to maintain its interaction with NCK-1. These results confirmed that the adaptor protein NCK-1 was required for CED-1 degradation through the proteasome pathway.

## Loss of UBC-21 and TRIM-21 affects apoptotic cell clearance through phagosome maturation

To determine the roles of UBC-21 and TRIM-21 in cell corpse engulfment, we used CRISPR-Cas9 to generate *ubc-21(xwh16)* and *trim-21(xwh13)* knockout worms. We found that both knockout lines contained significantly more cell corpses than WT at various embryonic stages and in adult germline cells (*Figure 6A*, *Figure 6—figure supplement 1A*). We found that *ubc-21* had significantly more germ cell corpses than *trim-21*, particularly at 48 and 60 hr post-L4, suggesting that additional E3 was involved or that UBC-21 has roles in addition to collaborating with TRIM-21. In addition, cell corpses persisted significantly longer in *trim-21(xwh13)* mutants than in WT worms (*Figure 6B*, *Figure 6—figure supplement 1B*). However, the number of cell deaths in *trim-21(xwh13)* embryos was indistinguishable from that in the wild-type (*Figure 6—figure supplement 1C*). We also found that overexpression of TRIM-21 under the control of the CED-1 or heat shock promoter resulted in increased cell corpse numbers (*Figure 6—figure supplement 1D–F*). These findings indicated that the accumulation of cell corpses resulted from defective corpse clearance rather than excessive apoptosis.

To examine the subcellular localization and expression patterns of TRIM-21, we integrated a TRIM-21::mCherry fusion protein, under the control of its native promoter *xwhIs33(P$_{trim-21}$trim-21::mcherry)*, into the WT *C. elegans* genome. We observed that TRIM-21::mCherry was ubiquitously expressed in the worms and appeared to be localized in the cytoplasm. In early larvae, TRIM-21::mCherry was mainly observed in pharyngeal muscle cells and body wall muscle cells (*Figure 6—figure supplement 1G*). A previous study showed that overexpression of CED-1 reduces cell corpses in *snx-1* mutants that impair CED-1 recycling (*Chen et al., 2010*). However, we found that overexpression of CED-1 enhanced the cell corpse phenotype in the *trim-21* mutant (*Figure 6—figure supplement 1H*). To test whether TRIM-21 could reverse the engulfment defects in a *trim-21(xwh13)* mutant, we observed the extra-chromosomal expression of TRIM-21::GFP driven by its native promoter (*P$_{trim-21}$trim-21::gfp*) and found complete rescue of cell corpse clearance in *trim-21(xwh13)* worms (*Figure 6—figure supplement 1I and J*). We found a high level of conservation between TRIM-21 from *C. elegans* and its

**Table 5.** Cell corpse phenotypes caused by RNAi of *C. elegans* genes encoding SH2 domain proteins.

| *C. elegans* SH2 domain proteins (RNAi) | No. of germ cell corpses (mean ± SEM) | | |
| --- | --- | --- | --- |
| | 1st | 2nd | 3rd |
| Control | 4.27 ± 0.44 | 3.47 ± 0.46 | 3.27 ± 0.55 |
| *chin-1* | 5.53 ± 0.62 | 7.27 ± 0.69*** | 4.73 ± 0.64 |
| *shc-1* | 4.93 ± 0.53 | 4.93 ± 0.64 | 5.33 ± 0.73* |
| *csk-1* | 5.07 ± 0.71 | 6.67 ± 0.78** | 7.00 ± 2.20 |
| *F39B2.5* | 4.07 ± 0.49 | 5.80 ± 0.82* | 5.73 ± 0.73 |
| *sli-1* | 4.27 ± 0.68 | 5.80 ± 0.77* | 4.80 ± 0.66 |
| *sem-5* | 8.40 ± 1.08** | ND | ND |
| *rin-1* | 5.13 ± 0.80 | 5.33 ± 0.64* | 6.00 ± 0.90* |
| *F13B12.6* | 2.47 ± 0.42 | ND | ND |
| *vav-1* | 4.73 ± 0.65 | 7.67 ± 0.56*** | 4.13 ± 0.43 |
| *gap-3* | 6.20 ± 0.85 | 5.60 ± 0.59* | 5.20 ± 0.76 |
| *nck-1* | 6.80 ± 0.76** | 7.00 ± 1.02** | 6.53 ± 0.36*** |
| *tns-1* | 5.53 ± 0.85 | ND | 5.13 ± 0.74 |
| *C18A11.4* | 6.20 ± 0.98 | 4.07 ± 0.42 | 6.93 ± 0.94** |
| *Y43C5B.2* | 5.87 ± 0.80 | 4.33 ± 0.68 | 4.20 ± 0.96 |
| *K11E4.2* | 5.00 ± 0.78 | 4.13 ± 0.40 | 4.00 ± 0.56 |
| *plc-3* | 2.67 ± 0.34 | 2.13 ± 0.48 | 2.13 ± 0.48 |
| *sta-2* | 3.80 ± 0.50 | 9.67 ± 2.67* | 7.33 ± 0.87*** |
| *Y116A8C.38* | 3.47 ± 0.46 | 4.40 ± 0.58 | 4.20 ± 0.55 |
| *soem-1* | 5.33 ± 0.42 | 3.27 ± 0.49 | 5.60 ± 0.61* |
| *Y52D5A.2* | 5.27 ± 0.66 | 4.93 ± 0.75 | 4.47 ± 0.55 |
| *sta-1* | 5.53 ± 0.58 | 5.40 ± 0.65* | 4.27 ± 0.62 |
| *aap-1* | 3.47 ± 0.56 | 3.40 ± 0.41 | 4.67 ± 0.71 |
| *shc-2* | 4.67 ± 0.54 | 3.93 ± 0.46 | 5.33 ± 0.62* |
| *Y37D8A.4* | 5.93 ± 0.79 | 5.20 ± 0.78 | 4.73 ± 68 |
| *ptp-1* | 9.53 ± 1.05 | 7.26 ± 0.91 | 3.6 ± 0.58 |
| *emb-5* | ND | ND | ND |

15 adult worms were scored for each RNAi treatment. Data were from three independent experiments. *p<0.05, **p<0.01, ***p<0.001, ND, no data.

The online version of this article includes the following source data for table 5:

**Source data 1.** Germ cell corpses in N2 treated with SH2 domain proteins RNAi.

homolog, hTRIM21, in humans. Expression of hTRIM21 driven by the *trim-21* promoter effectively rescued the defective corpse clearance phenotype of *trim-21(xwh13)* (***Figure 6—figure supplement 1K***), showing that human TRIM21 could functionally substitute for worm TRIM-21 in the cell corpse removal process.

To demonstrate that *trim-21*, *ubc-21*, and *ced-1* function in the same genetic pathway, we found that *ubc-21; trim-21* double mutants in either the *ubc-21(xwh16)* or *trim-21(xwh13)* strains did not exhibit altered defects in cell corpse clearance (***Figure 6A***, ***Figure 6—figure supplement 1A***). These data indicated that TRIM-21 and UBC-21 acted together to promote AC clearance. In addition, the

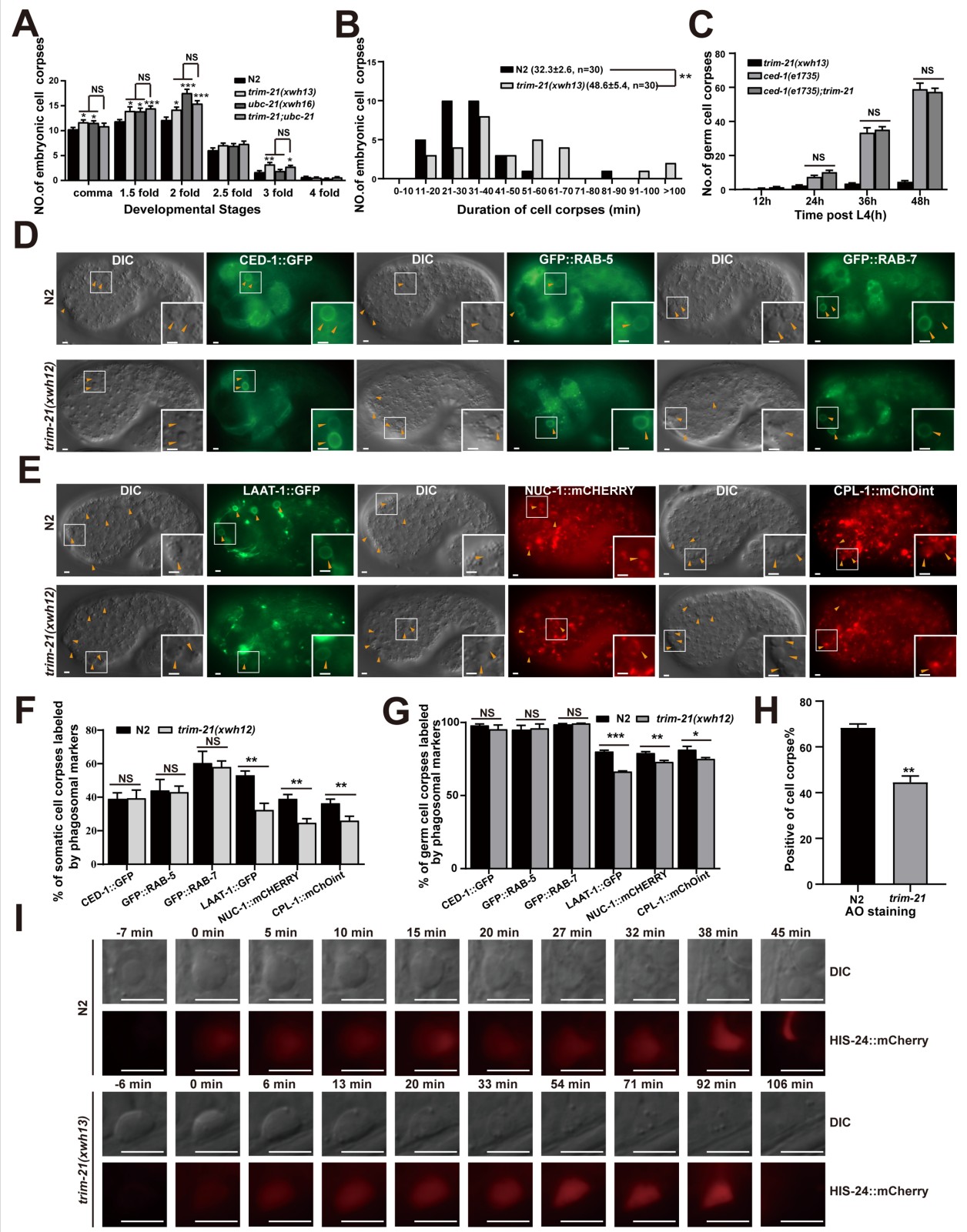

**Figure 6.** TRIM-21 acts in the CED-1 pathway to regulate phagosome maturation. (**A**) Different stages of embryonic corpses were quantified (mean ± SEM) in the indicated mutants. Fifteen embryos were scored at each stage for each strain. (**B**) Four-dimensional microscopy analysis of cell corpse duration was performed in N2 and *trim-21(xwh13)*. The persistence of 30 cell corpses from embryos was monitored. The mean duration (± SEM) is shown in parenthesis. (**C**) *trim-21(xwh13)*, null alleles mutant *ced-1(e1735)* and double null alleles mutant *trim-21(xwh13); ced-1(e1735)* germ cell corpses

*Figure 6 continued on next page*

*Figure 6 continued*

were quantified in different adult stages (hr post L4). Fifteen adult worms were scored at each stage for each strain. (**D, E**) The cell corpse labeling by the phagosome markers CED-1::GFP, GFP::RAB-5, GFP::RAB-7 (**D**), LAAT-1::GFP, NUC-1::mCHERRY, and CPL-1::mChOint (**E**) in N2 and *trim-21(xwh12)* embryos were captured using Imager M2 (Zeiss). Bars, 2 µm. (**F, G**) The cell corpses positive for phagosome markers in N2 and *trim-21(xwh12)* embryos (**F**) and germlines (**G**) were quantified. At least 100 cell corpses were scored for each strain. Data were from three independent experiments. (**H**) The cell corpse labeled by 0.1 mg/ml acridine orange was quantified (mean ± SEM) in N2 and *trim-21(xwh12)* adult worms. Data were from three independent experiments. (**I**) Time-lapse chasing of button-like cell corpses in DIC and HIS-24::mCherry-positive phagolysosomes in N2 and *trim-21 (xwh13)* germlines. The time point that the HIS-24::mCherry ring was first detected on a cell corpses was set as 0 min. Bars, 5 µm. An unpaired *t*-test was performed in this figure. *p<0.05, **p<0.01, ***p<0.001, NS, no significance. All bars indicate means and SEM.

The online version of this article includes the following source data and figure supplement(s) for figure 6:

**Source data 1.** Cell corpses and cell corpses duration in indicated strains, cell corpses labeled by phagosome markers and AO stainging in *trim-21* mutants.

**Figure supplement 1.** Loss of TRIM-21 and UBC-21 affect phagosome maturation.

**Figure supplement 1—source data 1.** Cell corpses and cell corpses duration in indicated strains, cell corpses labeled by phagosome markers in *ubc-21* mutants and the persistence of cell corpses labeled with HIS-24::mCherry in *trim-21* mutants.

loss of *trim-21* or *ubc-21* function did not further increase the accumulation of corpses in *ced-1(e1735)* germline cells (**Figure 6C**, **Figure 6—figure supplement 1L**), suggesting that *trim-21*, *ubc-21*, and *ced-1* probably function in the same genetic pathway. Fluorescence microscopic observation of the *ubc-21(xwh16)* and *trim-21(xwh13)* germline cells and embryos indicated that cell corpses were surrounded by CED-1::GFP phagocytic receptor–reporter fusion protein, just as in WT (**Figure 6D, F, and G**, **Figure 6—figure supplement 1M and N**), which suggested that recognition and initiation of engulfment were unaffected by the knockout of either gene. Next, we individually integrated a panel of GFP- or mCHERRY-tagged phagosomal markers into the *ubc-21(xwh16)* and *trim-21(xwh13)* mutants, respectively, and found that phagosome association with LAAT-1, a lysosomal membrane protein (**Liu et al., 2012**), was significantly reduced in both the *ubc-21(xwh16)* and *trim-21(xwh13)* mutants (**Figure 6E–G**, **Figure 6—figure supplement 1M and N**). Moreover, the signal from NUC-1, a lysosomal DNase, and CPL-1, a lysosomal cathepsin protease, was also significantly decreased on phagosomes in the *ubc-21(xwh16)* and *trim-21(xwh13)* mutants (**Figure 6E–G**, **Figure 6—figure supplement 1M and N**). These data showed that the loss of *ubc-21(xwh16)* and *trim-21(xwh13)* impaired phagosome maturation at a later stage.

To test whether the persistence of cell corpses in *trim-21(xwh13)* mutants resulted from a failure in the acidification of phagosomes containing cell corpses (i.e., a late stage of corpse clearance), acridine orange (AO) staining was used to highlight compromised cell corpses. We found that AO staining was greatly reduced in *trim-21(xwh13)* mutants compared with the WT worms (**Figure 6H**), suggesting that the loss of *trim-21(xwh13)* affected the acidification of cell corpse-containing phagosomes. To directly monitor the degradation process of cell corpses, we introduced a germline-specific transgenic marker for chromatin, H2B::mCHERRY, into *trim-21(xwh16)* mutants. We found that in the wild-type the chromatin in early germ cell corpses was condensed and disappeared within 60 min (54.6 ± 4.44 min; n = 5). In contrast, although the chromatin in early germ cell corpses of *trim-21(xwh13)* mutants was also condensed, it diffused throughout the corpse in the later stages of phagosome maturation, and the mCHERRY signal persisted for 90 min (90.2 ± 8.10 min, n = 5) (**Figure 6I**, **Figure 6—figure supplement 1O**), indicating that chromatin degradation was greatly delayed in the cell corpses of the mutant strain. Collectively, these findings demonstrated that the accumulation of cell corpses caused by the *trim-21* mutation was due to defects in cell corpse digestion in engulfing cells.

## Excessive CED-1 binding to the V-ATPase subunit VHA-10 in *trim-21* mutant worms negatively affects the maturation and acidification of cell corpse-containing phagosomes

Since endogenous CED-1 was increased by the *trim-21* mutation, we therefore proposed that excessive CED-1 in *trim-21* mutant worms negatively affected the maturation and acidification of cell corpse-containing phagosomes. To identify factors related to excess CED-1 accumulation that could inhibit phagosome maturation and acidification, we used an immunoprecipitation assay to screen for CED-1-interacting proteins in *C. elegans*. For this purpose, we established an integrated line that

expressed Flag-tagged CED-1-CT (CED-1-CT Flag) under the control of a heat shock promoter in WT animals. Next, we immunoprecipitated CED-1-CT Flag with an anti-Flag antibody and used LC-MS/MS to identify proteins bound to CED-1-CT (*Figure 7A*). In total, 15 proteins were identified as interaction partners of CED-1-CT (*Table 6*), with endosome system proteins representing the most attractive candidates. Interestingly, VHA-10 subunit G was identified as an interaction partner of CED-1-CT (*Figure 7B*). We further confirmed the direct interaction between CED-1-CT and VHA-10 by GST pull-down assay and proximity biotin labeling (*Figure 7C and D*).

VHA-10 is an ortholog of human ATPase H$^+$ transporting V1 subunit G2. This VHA subunit performs the same function across a multitude of essential cellular processes, such as acidification of lysosomes and intracellular organelles (*Collins and Forgac, 2020*). V-ATPase-mediated acidification of lysosomes is required for the activation of lysosomal hydrolases and ultimately leads to the enzymatic degradation of the cell corpse (*Ernstrom et al., 2012*). We found that *vha-10* knockdown by RNAi resulted in the increased germ cell corpse number (*Figure 7E*) due to delayed clearance of dying or dead cells (*Figure 7F*). We next examined the effects of VHA-10 on TRIM-21-mediated corpse clearance and found that *vha-10* knockdown did not affect corpse clearance defects in the *trim-21(xwh13)* mutant strain, suggesting that *trim-21* and *vha-10* likely function in the same genetic pathway (*Figure 7E*). In addition, the knockdown of *vha-10* impaired phagosome maturation at a late stage as well as the acidification of phagosomes containing cell corpses (*Figure 7G and H*), which was similar to the phenotype of *trim-21* loss-of-function mutants.

To address whether the dysfunction of *vha-10* was responsible for the defects in cell corpse degradation in *trim-21* mutants, we overexpressed VHA-10 by placing it under the control of the *ced-1* promoter (P$_{ced-1}$*vha-10::mcherry*) in *trim-21(xwh13)* mutant animals. Interestingly, VHA-10 overexpression fully attenuated the increased cell corpse phenotype and completely restored the acidification of phagosome defects in *trim-21(xwh13)* mutants (*Figure 7I and J*), indicating that VHA-10 dysfunction in engulfing cells caused defects in cell corpse degradation exhibited by *trim-21* mutants.

CED-1 is recycled from phagosome membranes to plasma membranes via the retromer complex, and a failure in recycling results in lysosomal degradation of CED-1 (*Chen et al., 2010*). We next investigated CED-1 recycling in the *trim-21* mutant. Recruitment and release of CED-1::GFP to cell corpses in *trim-21* (8.29 ± 0.48 min, n = 5) was similar to that in the WT (8.31 ± 0.25 min, n = 5) but not to that in *snx-1* (>30 min, n = 5) (*Figure 7—figure supplement 1A*). Previous studies suggested that CED-1 specifically initiates engulfment and controls phagosome maturation during cell corpse clearance (*Yu et al., 2008*). To confirm whether CED-1 was indeed degraded by proteasomes in addition to lysosomal degradation during phagosome maturation, we examined CED-1::GFP levels on the plasma membranes, phagosomal membranes, and phagolysosomal membranes in *trim-21* and *snx-1* mutants and in double mutants lacking both TRIM-21 and SNX-1. The CED-1::GFP levels in the *trim-21* mutants were comparable to those on the N2 in plasma membranes, phagosomal membranes, and phagolysosomal membranes, whereas *snx-1* mutants and *trim-21; snx-1* double mutants exhibited lower CED-1::GFP levels (*Figure 7—figure supplement 1B–E*).

In addition, co-localization of CED-1 and VHA-10 on phagosomes was dramatically increased in *trim-21* and *snx-1* mutants compared to in the WT (*Figure 7—figure supplement 2A–C*). To further verify whether the interaction between CED-1 and VHA-10 varied in the *trim-21* or *snx-1* mutants, we inserted an HA tag into the endogenous *vha-10* loci using CRISPR-Cas9 to generate *C. elegans* strains *xwh52(ha::vha-10)* (*Figure 7—figure supplement 2D*). A similar CED-1-VHA-10 interaction was observed in *snx-1* mutants and the WT; this interaction was greatly enhanced in *trim-21* mutants (*Figure 7—figure supplement 2E and F*), suggesting that excessive CED-1 in the *trim-21* mutants bound VHA-10, unlike in the *snx-1* mutants. Based on our findings, we proposed the ubiquitination of CED-1 by TRIM-21 occurs on forming phagosomes. Therefore, the transmembrane receptor CED-1 is presumably accumulated on the phagosomal surface due to the loss of function of TRIM-21. While we found that the level of CED-1 on phagosomes and phagolysosomes was not altered in *trim-21* mutants, whereas the co-localization and interaction of CED-1-VHA-10 were enhanced. A possible explanation for this might be that the level of excessive CED-1 is localized on the individual phagosomal surface in each phagocyte and goes through the dynamic phagosome maturation process. As a result, a substantial difference of CED-1 localized to the phagosomes and phagolysosomes between *trim-21* mutants and WT worms cannot be captured by the fluorescent image quantification method. Although levels of CED-1 and interaction of CED-1-VHA-10 were increased in *trim-21* mutants lysis

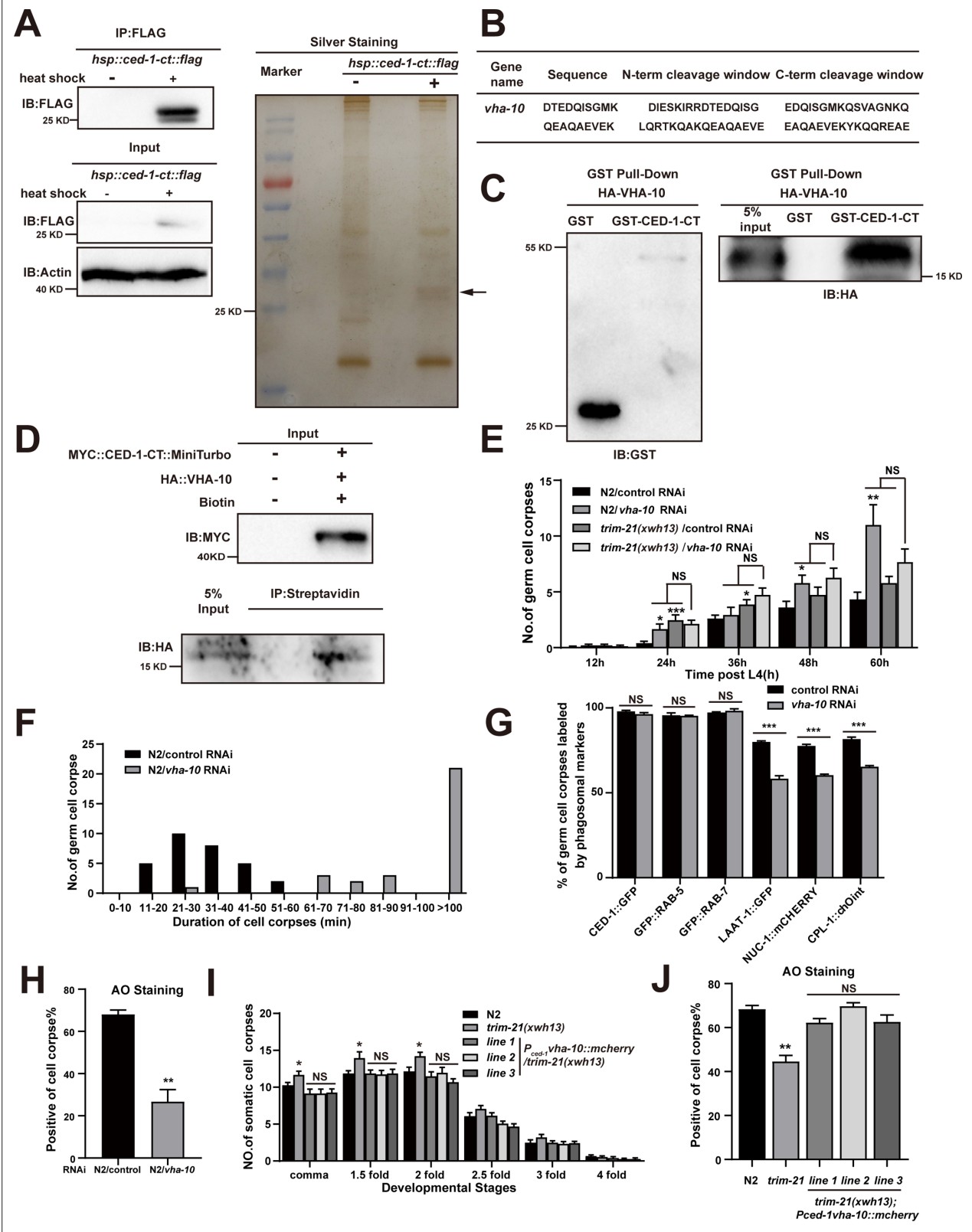

**Figure 7.** Excessive CED-1 binding to VHA-10 in *trim-21* mutant worms negatively affects the maturation and acidification of cell corpse-containing phagosomes. (**A**) The FLAG IP was performed on *P$_{hsp-16}$ced-1-ct::flag* worms (heat shock for 1 hr at 33°C), followed by identification of proteins that interact with CED-1 in MS. The immunoblot analysis (left) and silver staining (right) results are shown. (**B**) The peptides were identified by MS analysis. (**C**, **D**) The interaction between CED-1-CT and VHA-10 was examined by GST pull-down assays (**C**) and co-IP assays in 293T cells treated with 0.5 mM biotin

*Figure 7 continued on next page*

*Figure 7 continued*

(D). (E) The different adult stages of germ cell corpses were quantified in N2 and *trim-21(xwh13)* treated with control or *vha-10* RNAi. 15 adult worms were scored at each stage for each strain. (F) Four-dimensional microscopy analyses of 30 germ cell corpse duration were performed in N2 treated with control or *vha-10* RNAi. (G) The germ cell corpses positive for phagosome markers in N2 treated with control or *vha-10* RNAi were quantified. At least 100 cell corpses were scored for each strain. Data were from three independent experiments. (H) The cell corpse labeled by 0.1 mg/ml acridine orange was quantified in N2 treated with control or *vha-10* RNAi adult worms. At least 100 cell corpses were scored for each strain. Data were from three independent experiments. (I) Embryonic cell corpses were quantified in the indicated strains. Fifteen embryos at different stages were scored for each strain. (J) Cell corpse labeled by 0.1 mg/ml acridine orange was quantified (mean ± SEM) in the indicated strains of adult worms. At least 100 cell corpses were scored for each strain. Data were from three independent experiments. An unpaired *t*-test was performed in this figure. *$p<0.05$, **$p<0.01$, ***$p<0.001$, NS, no significance. All bars indicate means and SEM.

The online version of this article includes the following source data and figure supplement(s) for figure 7:

**Source data 1.** The immunoblot and silver staining in $P_{hsp-16}$*ced-1-ct::flag* worms, related proteins interactions, cell corpses in indicated strains, cell corpses duration and cell corpses labeled by phagosome markers in N2 treated with *vha-10* RNAi and AO staining in indicated strains.

**Figure supplement 1.** TRIM-21-mediated proteasome degradation of CED-1 independent of lysosomal degradation of CED-1 by loss of function of the retromer complex.

**Figure supplement 1—source data 1.** Time-lapse mointoring of CED-1::GFP on phagosomes in N2, *trim-21* and *snx-1* embryos.

**Figure supplement 2.** Excessive CED-1 in *trim-21* mutant worms binds VHA-10.

**Figure supplement 2—source data 1.** The percentage of VHA-10::mCherry and CED-1::GFP and the interaction between CED-1 and VHA-10 in N2, *trim-21*, *snx-1* mutants.

than in WT controls by the WB detection method, this reflects the change of a large number of phagosomes from massive amounts of phagocytes. Another possible explanation is that the excessive CED-1 in *trim-21* mutants accumulated on the phagosomal surface affects multiple stages of phagosome maturation process, which slows down the whole maturation process rather than a specific step, undetectable at the stage that we captured. As the mechanism determining which portions of CED-1 are ubiquitinated by TRIM-21 is unknown, we cannot distinguish portion of CED-1 regulated by TRIM-21 from the portion of CED-1 regulated by other mechanisms in *trim-21* mutants, making it difficult to follow where the excessive CED-1 is localized in *trim-21* mutants.

A crucial step in phagosome maturation is the gradual acidification of the phagosomal lumen because an acidic environment promotes the activity of hydrolytic enzymes that degrade phagosomal contents. In *C. elegans*, cell corpse-containing phagosome acidification begins fairly early, with Rab5 positive early phagosomes staining weakly with AO, indicating that there are multiple modes of acidification depending on the stage of maturation (*Kinchen et al., 2008*). Later study revealed that RAB-2 and RAB-14 act partially redundantly to promote phagosome acidification and recruit lysosomes for phagolysosome formation for cell corpse degradation, whereas RAB-7 mediates fusion of lysosomes to phagosomes but is largely dispensable for the acidification of phagosomes in *C. elegans*, indicating that acidification of cell corpse-containing phagosomes does not appear to be dependent on efficient phagosome–lysosome fusion (*Guo et al., 2010*; *Lu and Zhou, 2012*; *Wang and Yang, 2016*; *Yu et al., 2008*). However, the mechanism by which V-type ATPases regulate acidification of phagosomes containing ACs or AC degradation has not been thoroughly investigated. We discovered that excessive CED-1 binding to the V-ATPase in *trim-21* mutant worms reduces acidification of cell corpse-containing phagosomes. However, future research is needed to determine the precise stage at which phagosome acidification is affected, as well as how excessive CED-1 binding to VHA-10 in *trim-21* mutants affects cell corpse degradation. Taken together, excess CED-1, which accumulated in *trim-21* mutants, bound to the VHA-10 subunit of V-ATPase, negatively affected the acidification of phagosomes, and consequently blocked cell corpse degradation. Additionally, TRIM-21-mediated proteasome degradation of CED-1 occurs independently of lysosomal degradation of CED-1 through loss of function of the retromer complex.

## Discussion

Recent studies indicate that a sufficient amount of CED-1/Draper is critical for its engulfment function (*Hilu-Dadia et al., 2018*; *Kurant et al., 2008*; *MacDonald et al., 2006*; *Manaka et al., 2004*). Whether appropriate levels of CED-1 are maintained for executing engulfment function remains unknown.

**Table 6.** The factors identified by liquid chromatography–tandem mass spectrometry (LC-MS/MS) to be associated with the CED-1 pathway in *C. elegans*.

| Gene names | Number of proteins | Peptides | Unique peptides | Sequence coverage (%) | Mol. weight (kDa) | Sequence length | Sequence coverage No HS (%) | Sequence coverage HS (%) | LFQ intensity No HSP | LFQ intensity HS |
|---|---|---|---|---|---|---|---|---|---|---|
| hsp-16.1 | 2 | 2 | 2 | 18.6 | 16.253 | 145 | 0 | 18.6 | 0 | 63691000 |
| vha-10 | 1 | 2 | 2 | 15.1 | 14.485 | 126 | 0 | 15.1 | 0 | 1.71E+08 |
| iffb-1 | 1 | 1 | 1 | 1 | 120.42 | 1074 | 0 | 1 | 0 | 0 |
| his-35 | 4 | 1 | 1 | 5.5 | 13.418 | 127 | 0 | 5.5 | 0 | 2.69E+08 |
| CELE_T10C6.7 | 1 | 1 | 1 | 3.8 | 37.523 | 317 | 0 | 3.8 | 0 | 0 |
| rpn-8 | 1 | 1 | 1 | 2.5 | 40.687 | 362 | 0 | 2.5 | 0 | 0 |
| sumv-1 | 1 | 1 | 1 | 1.1 | 112.21 | 1024 | 0 | 1.1 | 0 | 0 |
| CELE_F55H12.4 | 1 | 1 | 1 | 3.8 | 22.043 | 208 | 0 | 3.8 | 0 | 86366000 |
| pdi-6 | 1 | 1 | 1 | 3.4 | 47.727 | 440 | 0 | 3.4 | 0 | 19651000 |
| C17C3.3 | 1 | 1 | 1 | 3.5 | 35.904 | 316 | 0 | 3.5 | 0 | 0 |
| gmeb-3 | 1 | 1 | 1 | 6.6 | 42.769 | 376 | 0 | 6.6 | 0 | 34430000 |
| F38B2.4 | 1 | 1 | 1 | 4.3 | 22.597 | 210 | 0 | 4.3 | 0 | 46574000 |
| npr-10 | 2 | 1 | 1 | 3 | 40.576 | 362 | 0 | 3 | 0 | 0 |
| otub-2 | 2 | 1 | 1 | 2.4 | 56.275 | 499 | 0 | 2.4 | 0 | 0 |
| CELE_Y62H9A.5 | 1 | 1 | 1 | 4.2 | 18.496 | 165 | 0 | 4.2 | 0 | 79508000 |

LFQ indicated the protein signal intensity after correction by LFQ algorithm. As a result of relative quantification of proteins, it is usually used for screening of differential proteins between different samples.

The online version of this article includes the following source data for table 6:

**Source data 1.** The factors identified by LC-MS/MS.

Here, we identified TRIM-21 in *C. elegans* as an important negative regulator of the engulfment receptor CED-1 and as a previously unrecognized component of the CED-1 pathway for AC clearance. In the absence of TRIM-21, the CED-1 protein accumulated to excess levels. We found that TRIM-21 ubiquitinates CED-1 directly, thereby targeting it for proteasomal degradation. The TRIM-21-mediated degradation of CED-1 appears to be activated by CED-1 NPXY motif binding to the CED-6 adaptor protein or after phosphorylation of the CED-1 YXXL motif by the tyrosine kinase, SRC-1, and subsequent binding by the adaptor protein, NCK-1. ITAM of Draper is phosphorylated by Src42a and binds to Shark, a tyrosine kinase homologous to Syk, to promote phagocytic signaling in *Drosophila* (*Ziegenfuss et al., 2008*). This signaling pathway of Draper is conserved among the mammalian phagocytic receptors Jedi-1 and MEGF10, which can independently interact with Syk through ITAMs and promote phagocytosis (*Scheib et al., 2012*). In contrast to *Drosophila* and mammals, *C. elegans* CED-1 lacks ITAM and contains only a single YXXL at its intracellular terminus. Our data indicate that SRC-1 phosphorylates YXXL of CED-1, and that phosphorylated CED-1 binds NCK-1, which has not been observed in *Drosophila* and mammals. Our results show that SRC-1 does not act within a single specific engulfment pathway, which differs from the findings of previous studies that reported that SRC-1 acts as a bridging molecule to link CED-2 and INA-1 (*Hsu and Wu, 2010*). Since the strong loss-of-function allele of *src-1* exhibits embryonic lethality, one possible explanation for this discrepancy may be differences in the efficiency of *src-1* knockdown. In *trim-21* knockout worms, CED-1 degradation mediated by TRIM-21 was essential for AC clearance because the accumulation of excess CED-1 protein blocked cell corpse degradation via binding with VHA-10 subunit of the proton pump for phagosomal acidification. Therefore, after CED-1 mediates signal transduction by interacting with cytoplasmic adaptor proteins, some amount of CED-1 should be rapidly 'cleared' to facilitate cell corpse degradation (*Figure 8*).

Previous studies have suggested that CED-1 acts specifically in initiating engulfment and controls phagosome maturation during cell corpse clearance (*Yu et al., 2008*). Our previous work revealed that loss of function of the retromer results in lysosomal degradation of CED-1. Here, we found that TRIM-21 mediated proteasome degradation of CED-1 independently of lysosomal degradation of CED-1 through loss of function of the retromer complex. We also found enhanced co-localization of CED-1 and VHA-10 on the phagosomal surface in both *trim-21* and *snx-1* mutants (*Figure 7—figure supplement 2A–C*); however, the CED-1 levels on the phagosomal surface are reduced in snx-1 and unaltered in trim-21 (*Figure 7—figure supplement 1D and E*). Given that V-type ATPases are trafficked to the phagosome and function to acidify its contents during the phagosome maturation process (*Kinchen and Ravichandran, 2008*), one possible explanation for these results is that CED-1 failed to recycle from phagosomes and cytosol back to the plasma membrane in *snx-1* mutants, resulting in the enhanced co-localization of CED-1 and VHA-10 on the phagosomal surface despite no increased interaction between CED-1 and VHA-10. Unlike in *snx-1* mutants, the enhanced co-localization of CED-1 and VHA-10 on the phagosomal surface is at least partially due to increased interaction between CED-1 and VHA-10 in *trim-21* mutants. Another possible explanation is that the amount of excessive CED-1 degraded by TRIM-21 might be less than the amount of CED-1 recycled by retromer as *trim-21; snx-1* double mutants exhibited lower CED-1 levels (*Figure 7—figure supplement 1B–E*), which requires further investigation. Therefore, we propose that TRIM-21 functions downstream of CED-6 and NCK-1 to mediate CED-1 degradation, preventing excessive CED-1 from affecting phagosome acidification and maturation, whereas retromer-mediated effective recycling of CED-1 contributes to efficient engulfment of cell corpses. Our findings establish the function of TRIM-21 in AC clearance by mediating the degradation of the engulfment receptor CED-1, representing the first identification of CED1-TRIM-21 signaling in invertebrates. Moreover, this finding suggests that CED-1 family engulfment receptors may be similarly regulated in other organisms.

In mammals, the E3 ubiquitin-protein ligase TRIM21 mediates a wide range of processes, including innate and adaptive immunity, through its interactions with numerous proteins (*McEwan et al., 2013*; *Zhang et al., 2013*). A critical molecular feature of TRIM21 is its binding to the fragment crystallizable region (Fc) of antibodies for recognition of numerous pathogens via its PRYSPRY domain, which results in the ubiquitination and degradation of these pathogens through the proteasome pathway (*James et al., 2007*). The TRIM-21 in *C. elegans* lacks the PRYSPRY domain for pathogen antibody binding, which is in line with the complete reliance of *C. elegans* on its innate immune system for defense against pathogens (*Kim and Ewbank, 2018*). Given that the regulation of response to unfolded proteins by

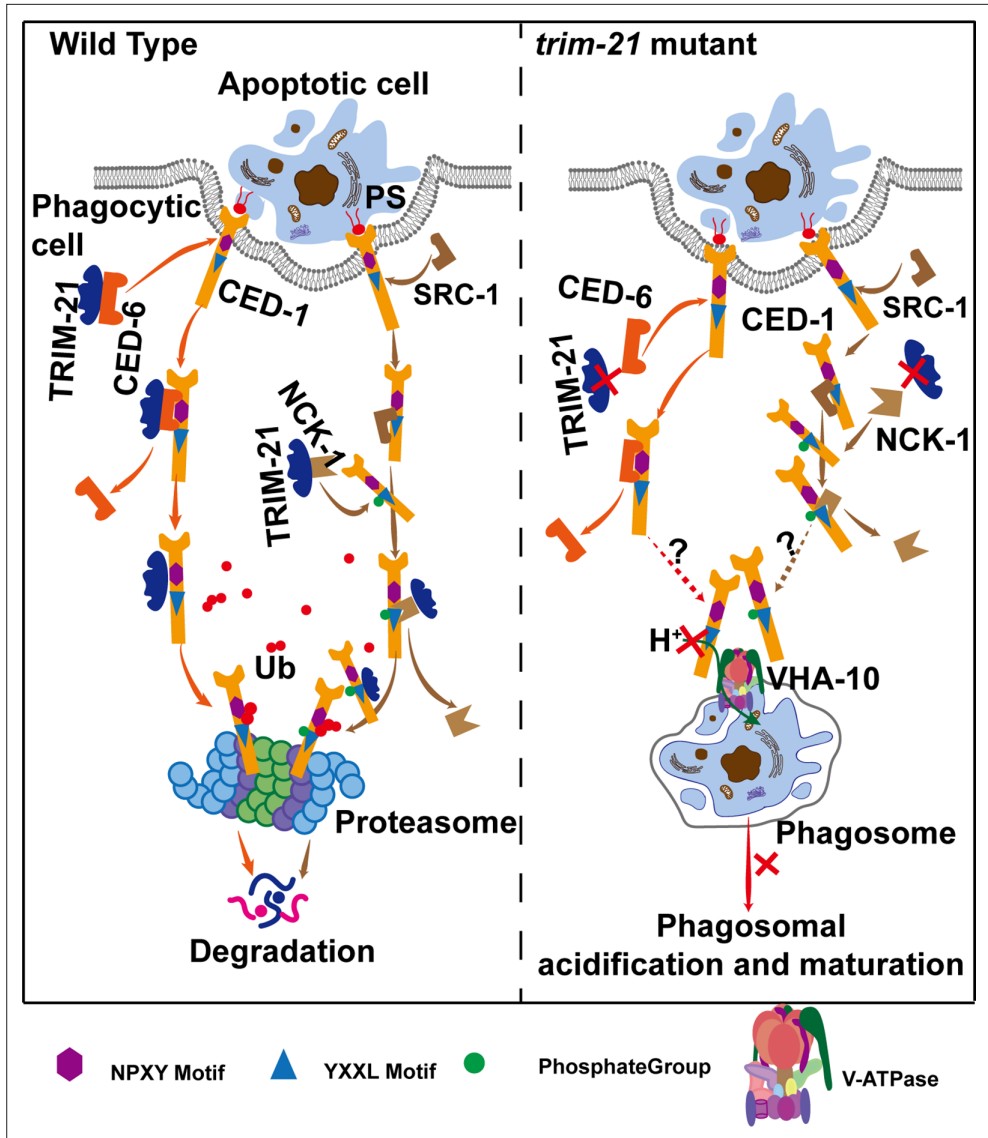

**Figure 8.** Model of TRIM-21-mediated CED-1 degradation through the proteasome pathway. After phagocytic receptors (CED-1) recognize apoptotic cells and cell corpse engulfment, TRIM-21 is recruited to the surfaces of phagosomes through two parallel pathways and ubiquitinates part of CED-1 for proteasomal degradation. One pathway occurs after the NPXY motif of CED-1 binds to the adaptor protein CED-6, and TRIM-21 released from CED-6 and ubiquitinates CED-1. The other occurs when a tyrosine residue in the YXXL motif of CED-1 is phosphorylated by tyrosine kinase SRC-1, and the phosphorylated CED-1 YXXL motif recruits an adaptor protein NCK-1 (containing the SH2 domain), followed by TRIM-21 being released from NCK-1 and ubiquitinating CED-1. In the absence of TRIM-21, part of CED-1 fails to be ubiquitinated for proteasomal degradation and accumulates in phagosomes to bind to VHA-10 and affect phagosomal acidification and maturation, thus resulting in cell corpse degradation defects. It is currently unknown how the excess CED-1 in *trim-21* mutants specifically binds to VHA-10.

CED-1 is essential for innate immunity in *C. elegans* (*Haskins et al., 2008*), further studies are necessary to determine the contribution of *trim-21* in innate immunity. Our work suggests that TRIM-21 is an ancient E3 ligase in which the coiled-coil domain recognizes its substrate CED-1. hTRIM21 was originally discovered as an 'autoantigen' (RO52 or SS-A) in patients with systemic lupus erythematosus (SLE) (*Ben-Chetrit et al., 1988*), and its increased expression was also found in Sjögren's syndrome (*Ben-Chetrit et al., 1990*). Interestingly, we found that hTRIM21 interacts with MEGF10-CT and CED-1-CT, and TRIM-21 too interacted with CED-1-CT and MEGF10-CT. MEGF10, the mammalian homolog of CED-1, has also been shown to participate in the pathogenesis of Alzheimer's disease

(*Singh et al., 2010*). Thus, the identification of TRIM-21 as a negative regulator of CED-1 will have important implications for understanding its role in the pathogenesis of human autoimmune diseases, such as SLE, and neurodegenerative diseases, such as Alzheimer's disease, in addition to the mechanistic insights into its role in AC clearance.

# Materials and methods

## Key resources table

| Reagent type (species) or resource | Designation | Source or reference | Identifiers | Additional information |
|---|---|---|---|---|
| Strain, strain background (*Caenorhabditis elegans*) | *ced-1::flag(xwh17)* I 2× | This paper | SNU19 | *Figure 1*<br>*Figure 4—figure supplement 1*; Available from the Xiao Lab |
| Strain, strain background (*C. elegans*) | *xwhIs27[P$_{ced-1}$ced-1::flag, sur-5::gfp]* | This paper | SNU31 | *Figure 1*<br>; Available from the Xiao Lab |
| Strain, strain background (*C. elegans*) | *xwhIs28[P$_{hsp-16}$trim-21::flag, sur-5::gfp]* | This paper | SNU32 | *Figure 1*<br>*Figure 3*<br>*Figure 3—figure supplement 1*<br>*Figure 5—figure supplement 1*;<br>Available from the Xiao Lab |
| Strain, strain background (*C. elegans*) | *trim-21(xwh12)* II 2× | This paper | SNU12 | *Figure 1*<br>*Figure 6—figure supplement 1*;<br>Available from the Xiao Lab |
| Strain, strain background (*C. elegans*) | *trim-21(xwh13)* II 6× | This paper | SNU13 | *Figure 1*<br>*Figure 6*<br>*Figure 6—figure supplement 1*<br>*Figure 7*;<br>Available from the Xiao Lab |
| Strain, strain background (*C. elegans*) | *smIs34[P$_{ced-1}$ced-1::gfp, rol-6(su1006)]* | Dr. Chonglin Yang | CU1546 | *Figure 1* |
| Strain, strain background (*C. elegans*) | *smIs110[P$_{ced-1}$ced-1DC::gfp]* | Dr. Chonglin Yang (*Chen et al., 2013*) | N/A | *Figure 1* |
| Strain, strain background (*C. elegans*) | *bcIs39[P$_{lim-7}$ced-1::gfp, lin-15(+)]* | Dr. Chonglin Yang | MD701 | *Figure 1* |
| Strain, strain background (*C. elegans*) | *trim-21(xwh12); smIs34[P$_{ced-1}$ced-1::gfp, rol-6(su1006)]* | This paper | SNU33 | *Figure 1*<br>*Figure 6*<br>*Figure 7—figure supplement 1*;<br>Available from the Xiao Lab |
| Strain, strain background (*C. elegans*) | *trim-21(xwh12); bcIs39[P$_{lim-7}$ced-1::gfp, lin-15(+)]* | This paper | SNU34 | *Figure 1*;<br>Available from the Xiao Lab |
| Strain, strain background (*C. elegans*) | *trim-21(xwh12); smIs110[P$_{ced-1}$ced-1DC::gfp]* | This paper | SNU35 | *Figure 1*;<br>Available from the Xiao Lab |
| Strain, strain background (*C. elegans*) | *ced-1::flag(xwh17); ha::ubq-2(xwh20)* | This paper | SNU22 | *Figure 2*<br>*Figure 3*<br>*Figure 4*<br>*Figure 4—figure supplement 1*<br>*Figure 5*<br>*Figure 5—figure supplement 1*;<br>Available from the Xiao Lab |
| Strain, strain background (*C. elegans*) | *ubc-21(xwh15)* X 3× | This paper | SNU15 | Available from the Xiao Lab |
| Strain, strain background (*C. elegans*) | *ubc-21(xwh16)* X 6× | This paper | SNU16 | *Figure 6*<br>*Figure 6—figure supplement 1*;<br>Available from the Xiao Lab |
| Strain, strain background (*C. elegans*) | *ubc-21(xwh15); ced-1::flag(xwh17); ha::ubq-2(xwh20)* | This paper | SNU36 | *Figure 2*;<br>Available from the Xiao Lab |
| Strain, strain background (*C. elegans*) | *trim-21(xwh12); ced-1::flag(xwh17); ha::ubq-2(xwh20)* | This paper | SNU37 | *Figure 2*;<br>Available from the Xiao Lab |
| Strain, strain background (*C. elegans*) | *trim-21(xwh12); ced-1::flag(xwh17); ha::ubq-2-K48R(xwh23)* | This paper | SNU25 | *Figure 2*;<br>Available from the Xiao Lab |

*Continued on next page*

*Continued*

| Reagent type (species) or resource | Designation | Source or reference | Identifiers | Additional information |
|---|---|---|---|---|
| Strain, strain background (*C. elegans*) | *trim-21(xwh12); ced-1::flag(xwh17); ha::ubq-2-K63R(xwh24)* | This paper | SNU26 | **Figure 2**; Available from the Xiao Lab |
| Strain, strain background (*C. elegans*) | *xwhIs29[P$_{ced-1}$trim-21::flag, sur-5::gfp]* | This paper | SNU38 | **Figure 3**; Available from the Xiao Lab |
| Strain, strain background (*C. elegans*) | *ced-6(xwh25); xwhIs29[P$_{ced-1}$trim-21::flag, sur-5:: gfp]* | This paper | SNU39 | **Figure 3**; Available from the Xiao Lab |
| Strain, strain background (*C. elegans*) | *smIs34[P$_{ced-1}$ced-1::gfp, rol-6(su1006)]; xwhIs30[P$_{ced-1}$trim-21::mcherry, rol-6(su1006)]* | This paper | SNU43 | **Figure 3**; Available from the Xiao Lab |
| Strain, strain background (*C. elegans*) | *smIs34[P$_{ced-1}$ced-1::gfp, rol-6(su1006)]; xwhIs31[P$_{ced-1}$mcherry::ced-6, rol-6(su1006)]* | This paper | SNU44 | **Figure 3**; Available from the Xiao Lab |
| Strain, strain background (*C. elegans*) | *xwhIs31[P$_{ced-1}$mcherry::ced-6, rol-6(su1006)]; xwhIs32[P$_{ced-1}$trim-21::gfp, rol-6(su1006)]* | This paper | SNU45 | **Figure 3**; Available from the Xiao Lab |
| Strain, strain background (*C. elegans*) | *xwhEx34[P$_{ced-1}$ced-1::flag, sur-5:: gfp]/ ced-1(e1735)* | This paper | SNU47 | **Figure 3**; Available from the Xiao Lab |
| Strain, strain background (*C. elegans*) | *xwhEx35[P$_{ced-1}$ced-1(N962A)::flag, sur-5:: gfp]/ced-1(e1735)* | This paper | SNU48 | **Figure 3**; Available from the Xiao Lab |
| Strain, strain background (*C. elegans*) | *ced-6(xwh25)* III | This paper | SNU27 | **Figure 3**; Available from the Xiao Lab |
| Strain, strain background (*C. elegans*) | *ced-6(xwh25); xwhIs28[P$_{hsp-16}$trim-21::flag, sur-5::gfp]* | This paper | SNU49 | **Figure 3** **Figure 3—figure supplement 1**; Available from the Xiao Lab |
| Strain, strain background (*C. elegans*) | *ubc-21(xwh16); trim-21(xwh13)* | This paper | SNU17 | **Figure 6** **Figure 6—figure supplement 1**; Available from the Xiao Lab |
| Strain, strain background (*C. elegans*) | *src-1(xwh26); +/hT2* III | This paper | SNU28 | **Figure 4**; Available from the Xiao Lab |
| Strain, strain background (*C. elegans*) | *qxIs408[P$_{ced-1}$gfp::rab-5]* | Dr. Chonglin Yang (**Chen et al., 2013**) | N/A | **Figure 6** |
| Strain, strain background (*C. elegans*) | *trim-21(xwh12); qxIs408[P$_{ced-1}$gfp::rab-5]* | This paper | SNU50 | **Figure 6**; Available from the Xiao Lab |
| Strain, strain background (*C. elegans*) | *qxIs66[P$_{ced-1}$gfp::rab-7]* | Dr. Chonglin Yang (**Liu et al., 2012**) | N/A | **Figure 6** |
| Strain, strain background (*C. elegans*) | *trim-21(xwh12); qxIs66[P$_{ced-1}$gfp::rab-7]* | This paper | SNU51 | **Figure 6**; Available from the Xiao Lab |
| Strain, strain background (*C. elegans*) | *qxIs354[P$_{ced-1}$laat-1::gfp]* | Dr. Chonglin Yang (**Liu et al., 2012**) | N/A | **Figure 6** |
| Strain, strain background (*C. elegans*) | *trim-21(xwh12); qxIs354[P$_{ced-1}$laat-1::gfp]* | This paper | SNU52 | **Figure 6**; Available from the Xiao Lab |
| Strain, strain background (*C. elegans*) | *qxIs257[P$_{ced-1}$nuc-1::mcherry]* | Dr. Chonglin Yang (**Chen et al., 2013**) | N/A | **Figure 6** |
| Strain, strain background (*C. elegans*) | *trim-21(xwh12); qxIs257[P$_{ced-1}$nuc-1::mcherry]* | This paper | SNU53 | **Figure 6**; Available from the Xiao Lab |
| Strain, strain background (*C. elegans*) | *yqEx620[P$_{ced-1}$cpl-1::mchoint]* | Dr. Chonglin Yang (**Xu et al., 2014**) | N/A | **Figure 6** |
| Strain, strain background (*C. elegans*) | *trim-21(xwh12); yqEx620[P$_{ced-1}$cpl-1::mchoint]* | This paper | SNU54 | **Figure 6**; Available from the Xiao Lab |
| Strain, strain background (*C. elegans*) | *ced-1(e1735); trim-21(xwh13)* | This paper | SNU61 | **Figure 6**; Available from the Xiao Lab |
| Strain, strain background (*C. elegans*) | *ced-1(e1735); ubc-21(xwh16)* | This paper | SNU62 | **Figure 6—figure supplement 1**; Available from the Xiao Lab |

*Continued on next page*

*Continued*

| Reagent type (species) or resource | Designation | Source or reference | Identifiers | Additional information |
|---|---|---|---|---|
| Strain, strain background (*C. elegans*) | xwhIs36[P$_{hsp-16}$ced-1-ct::flag, sur-5::gfp] | This paper | SNU63 | ***Figure 7***; Available from the Xiao Lab |
| Strain, strain background (*C. elegans*) | ced-1(N962A)::flag(xwh17) I | This paper | SNU73 | ***Figure 4—figure supplement 1***; Available from the Xiao Lab |
| Strain, strain background (*C. elegans*) | ced-1(Y1019F)::flag(xwh17) I | This paper | SNU74 | ***Figure 4—figure supplement 1***; Available from the Xiao Lab |
| Strain, strain background (*C. elegans*) | ced-1(N962A)::flag(xwh17); ha::ubq-2(xwh20) | This paper | SNU75 | ***Figure 4—figure supplement 1***; Available from the Xiao Lab |
| Strain, strain background (*C. elegans*) | ced-1(N962A)::flag(xwh17); ha::ubq-2(xwh20) | This paper | SNU76 | ***Figure 4—figure supplement 1***; Available from the Xiao Lab |
| Strain, strain background (*C. elegans*) | xwhEx37[P$_{ced-1}$trim-21::flag, sur-5:: gfp] line1 | This paper | SNU64 | ***Figure 6—figure supplement 1***; Available from the Xiao Lab |
| Strain, strain background (*C. elegans*) | xwhEx38[P$_{ced-1}$trim-21::flag, sur-5:: gfp] line2 | This paper | SNU65 | ***Figure 6—figure supplement 1***; Available from the Xiao Lab |
| Strain, strain background (*C. elegans*) | xwhEx39[P$_{ced-1}$trim-21::flag, sur-5:: gfp] line3 | This paper | SNU66 | ***Figure 6—figure supplement 1***; Available from the Xiao Lab |
| Strain, strain background (*C. elegans*) | xwhEx40[P$_{trim-21}$trim-21::gfp, sur-5:: gfp] line1/trim-21(xwh13) | This paper | SNU67 | ***Figure 6—figure supplement 1***; Available from the Xiao Lab |
| Strain, strain background (*C. elegans*) | xwhEx41[P$_{trim-21}$trim-21::gfp, sur-5:: gfp] line2/trim-21(xwh13) | This paper | SNU68 | ***Figure 6—figure supplement 1***; Available from the Xiao Lab |
| Strain, strain background (*C. elegans*) | xwhEx42[P$_{trim-21}$trim-21::gfp, sur-5:: gfp] line3/trim-21(xwh13) | This paper | SNU69 | ***Figure 6—figure supplement 1***; Available from the Xiao Lab |
| Strain, strain background (*C. elegans*) | xwhEx43[P$_{ced-1}$vha-10::mcherry, sur-5:: gfp] line1/trim-21(xwh13) | This paper | SNU70 | ***Figure 7***; Available from the Xiao Lab |
| Strain, strain background (*C. elegans*) | xwhEx44[P$_{ced-1}$vha-10::mcherry, sur-5:: gfp] line2/trim-21(xwh13) | This paper | SNU71 | ***Figure 7***; Available from the Xiao Lab |
| Strain, strain background (*C. elegans*) | xwhEx45[P$_{ced-1}$vha-10::mcherry, sur-5:: gfp] line3/trim-21(xwh13) | This paper | SNU72 | ***Figure 7***; Available from the Xiao Lab |
| Strain, strain background (*C. elegans*) | xwhEx46[P$_{trim-21}$htrim21, sur-5:: gfp] line1/trim-21(xwh13) | This paper | SNU77 | ***Figure 6—figure supplement 1***; Available from the Xiao Lab |
| Strain, strain background (*C. elegans*) | xwhEx47[P$_{trim-21}$htrim21, sur-5:: gfp] line2/trim-21(xwh13) | This paper | SNU78 | ***Figure 6—figure supplement 1***; Available from the Xiao Lab |
| Strain, strain background (*C. elegans*) | xwhEx48[P$_{trim-21}$htrim21, sur-5:: gfp] line3/trim-21(xwh13) | This paper | SNU79 | ***Figure 6—figure supplement 1***; Available from the Xiao Lab |
| Strain, strain background (*C. elegans*) | xwhIs49[P$_{ced-1}$rde-1, rol-6(su1006)]/rde-1 | This paper | SNU81 | ***Figure 5***; Available from the Xiao Lab |
| Strain, strain background (*C. elegans*) | ujIs113 [P$_{pie-1}$H2B::mCherry, unc-119(+); P$_{nhr-2}$HIS-24::mCherry, unc-119(+)] | Dr. Chonglin Yang | JIM113 | ***Figure 6*** |
| Strain, strain background (*C. elegans*) | trim-21(xwh13); ujIs113 [P$_{pie-1}$H2B::mCherry, unc-119(+); P$_{nhr-2}$HIS-24::mCherry, unc-119(+)] | This paper | SNU80 | ***Figure 6***; Available from the Xiao Lab |
| Strain, strain background (*C. elegans*) | ubc-21(xwh15); smIs34[P$_{ced-1}$ced-1::gfp, rol-6(su1006)] | This paper | SNU55 | ***Figure 6—figure supplement 1***; Available from the Xiao Lab |
| Strain, strain background (*C. elegans*) | ubc-21(xwh15); qxIs408[P$_{ced-1}$gfp::rab-5] | This paper | SNU56 | ***Figure 6—figure supplement 1***; Available from the Xiao Lab |
| Strain, strain background (*C. elegans*) | qxIs68[P$_{ced-1}$mcherry::rab-7] | Dr. Chonglin Yang (***Cheng et al., 2015***) | N/A | ***Figure 6—figure supplement 1*** |
| Strain, strain background (*C. elegans*) | ubc-21(xwh15); qxIs68[P$_{ced-1}$mcherry::rab-7] | This paper | SNU57 | ***Figure 6—figure supplement 1***; Available from the Xiao Lab |
| Strain, strain background (*C. elegans*) | qxIs352[P$_{ced-1}$laat-1::mcherry] | Dr. Chonglin Yang (***Liu et al., 2012***) | N/A | ***Figure 6—figure supplement 1*** |

*Continued on next page*

*Continued*

| Reagent type (species) or resource | Designation | Source or reference | Identifiers | Additional information |
|---|---|---|---|---|
| Strain, strain background (*C. elegans*) | *ubc-21(xwh15); qxIs352[P_{ced-1}laat-1::mcherry]* | This paper | SNU58 | *Figure 6—figure supplement 1*; Available from the Xiao Lab |
| Strain, strain background (*C. elegans*) | *ubc-21(xwh15); qxIs257[P_{ced-1}nuc-1::mcherry]* | This paper | SNU59 | *Figure 6—figure supplement 1*; Available from the Xiao Lab |
| Strain, strain background (*C. elegans*) | *ubc-21(xwh15); yqEx620[P_{ced-1}cpl-1::mchoint]* | This paper | SNU60 | *Figure 6—figure supplement 1*; Available from the Xiao Lab |
| Strain, strain background (*C. elegans*) | *nck-1(xwh51)* | This paper | SNU83 | *Figure 5—figure supplement 1*; Available from the Xiao Lab |
| Strain, strain background (*C. elegans*) | *ced-6(xwh25); smIs34[P_{ced-1}ced-1::gfp, rol-6(su1006)]; xwhIs30[P_{ced-1}trim-21::mcherry, rol-6(su1006)]* | This paper | SNU85 | *Figure 3*; Available from the Xiao Lab |
| Strain, strain background (*C. elegans*) | *ced-6(xwh25); xwhIs32[P_{ced-1}trim-21::gfp, rol-6(su1006)]* | This paper | SNU86 | *Figure 3—figure supplement 1*; Available from the Xiao Lab |
| Strain, strain background (*C. elegans*) | *xwhIs53[P_{ced-1}ced-1::gfp, P_{odr-1}:: rfp]* | This paper | SNU87 | *Figure 5*; Available from the Xiao Lab |
| Strain, strain background (*C. elegans*) | *xwhIs54[P_{ced-1}ced-1(Y1019F)::gfp, rol-6(su1006)]* | This paper | SNU88 | *Figure 5*; Available from the Xiao Lab |
| Strain, strain background (*C. elegans*) | *trim-21(xwh13); xwhIs27[P_{ced-1}ced-1::flag, sur-5::gfp]* | This paper | SNU90 | *Figure 6—figure supplement 1*; Available from the Xiao Lab |
| Strain, strain background (*C. elegans*) | *snx-1(tm847)* | Dr. Chonglin Yang (*Chen et al., 2010*) | N/A | *Figure 6* |
| Strain, strain background (*C. elegans*) | *qxIs58[P_{ced-1}lmp-1::mcherry]* | Dr. Chonglin Yang (*Sasaki et al., 2013*) | N/A | *Figure 7—figure supplement 1* |
| Strain, strain background (*C. elegans*) | *snx-1(tm847); smIs34[P_{ced-1}ced-1::gfp, rol-6(su1006)]* | This paper | SNU91 | *Figure 7—figure supplement 1*; Available from the Xiao Lab |
| Strain, strain background (*C. elegans*) | *trim-21(xwh13); smIs34[P_{ced-1}ced-1::gfp, rol-6(su1006)]* | This paper | SNU92 | *Figure 7—figure supplement 1*; Available from the Xiao Lab |
| Strain, strain background (*C. elegans*) | *trim-21(xwh13); snx-1(tm847); smIs34[P_{ced-1}ced-1::gfp, rol-6(su1006)]* | This paper | SNU93 | *Figure 7—figure supplement 1*; Available from the Xiao Lab |
| Strain, strain background (*C. elegans*) | *smIs34[P_{ced-1}ced-1::gfp, rol-6(su1006)]; qxIs68[P_{ced-1}mcherry::rab-7]* | This paper | SNU94 | *Figure 7—figure supplement 1*; Available from the Xiao Lab |
| Strain, strain background (*C. elegans*) | *trim-21(xwh13); smIs34[P_{ced-1}ced-1::gfp, rol-6(su1006)]; qxIs68[P_{ced-1}mcherry::rab-7]* | This paper | SNU95 | *Figure 7—figure supplement 1*; Available from the Xiao Lab |
| Strain, strain background (*C. elegans*) | *snx-1(tm847); smIs34[P_{ced-1}ced-1::gfp, rol-6(su1006)]; qxIs68[P_{ced-1}mcherry::rab-7]* | This paper | SNU96 | *Figure 7—figure supplement 1*; Available from the Xiao Lab |
| Strain, strain background (*C. elegans*) | *trim-21(xwh13); snx-1(tm847); smIs34[P_{ced-1}ced-1::gfp, rol-6(su1006)]; qxIs68[P_{ced-1}mcherry::rab-7]* | This paper | SNU97 | *Figure 7—figure supplement 1*; Available from the Xiao Lab |
| Strain, strain background (*C. elegans*) | *smIs34[P_{ced-1}ced-1::gfp, rol-6(su1006)]; qxIs58[P_{ced-1}lmp-1::mcherry]* | This paper | SNU98 | *Figure 7—figure supplement 1*; Available from the Xiao Lab |
| Strain, strain background (*C. elegans*) | *trim-21(xwh13); smIs34[P_{ced-1}ced-1::gfp, rol-6(su1006)]; qxIs58[P_{ced-1}lmp-1::mcherry]* | This paper | SNU99 | *Figure 7—figure supplement 1*; Available from the Xiao Lab |
| Strain, strain background (*C. elegans*) | *snx-1(tm847); smIs34[P_{ced-1}ced-1::gfp, rol-6(su1006)]; qxIs58[P_{ced-1}lmp-1::mcherry]* | This paper | SNU100 | *Figure 7—figure supplement 1*; Available from the Xiao Lab |
| Strain, strain background (*C. elegans*) | *trim-21(xwh13); snx-1(tm847); smIs34[P_{ced-1}ced-1::gfp, rol-6(su1006)]; qxIs58[P_{ced-1}lmp-1::mcherry]* | This paper | SNU101 | *Figure 7—figure supplement 1*; Available from the Xiao Lab |
| Strain, strain background (*C. elegans*) | *trim-21(xwh13); xwhIs53[P_{ced-1}ced-1::gfp, P_{odr-1}:: rfp]* | This paper | SNU105 | *Figure 7—figure supplement 2*; Available from the Xiao Lab |

*Continued on next page*

*Continued*

| Reagent type (species) or resource | Designation | Source or reference | Identifiers | Additional information |
|---|---|---|---|---|
| Strain, strain background (*C. elegans*) | snx-1(tm847); xwhIs53[P$_{ced-1}$ced-1::gfp, P$_{odr-1}$:: rfp] | This paper | SNU106 | *Figure 7—figure supplement 2*; Available from the Xiao Lab |
| Strain, strain background (*C. elegans*) | xwhEx55[P$_{ced-1}$vha-10::mcherry/ xwhIs53[P$_{ced-1}$ced-1::gfp, P$_{odr-1}$:: rfp]] | This paper | SNU102 | *Figure 7—figure supplement 2*; Available from the Xiao Lab |
| Strain, strain background (*C. elegans*) | xwhEx56[P$_{ced-1}$vha-10::mcherry/ trim-21(xwh13); xwhIs53[P$_{ced-1}$ced-1::gfp, P$_{odr-1}$:: rfp]] | This paper | SNU103 | *Figure 7—figure supplement 2*; Available from the Xiao Lab |
| Strain, strain background (*C. elegans*) | xwhEx57[P$_{ced-1}$vha-10::mcherry/ snx-1(tm847); xwhIs53[P$_{ced-1}$ced-1::gfp, P$_{odr-1}$:: rfp]] | This paper | SNU104 | *Figure 7—figure supplement 2*; Available from the Xiao Lab |
| Strain, strain background (*C. elegans*) | ha::vha-10(xwh52) | This paper | SNU89 | *Figure 7—figure supplement 2*; Available from the Xiao Lab |
| Strain, strain background (*C. elegans*) | trim-21(xwh13); ha::vha-10(xwh52) | This paper | SNU107 | *Figure 7—figure supplement 2*; Available from the Xiao Lab |
| Strain, strain background (*C. elegans*) | snx-1(tm847); ha::vha-10(xwh52) | This paper | SNU108 | *Figure 7—figure supplement 2*; Available from the Xiao Lab |
| Strain, strain background (*C. elegans*) | xwhEx58[P$_{ced-1}$ced-1, sur-5:: gfp] line2/ ced-1(e1735) | This paper | SNU109 | *Table 3*; Available from the Xiao Lab |
| Strain, strain background (*C. elegans*) | xwhEx59[P$_{ced-1}$ced-1, sur-5:: gfp] line1/ ced-1(e1735) | This paper | SNU110 | *Table 3*; Available from the Xiao Lab |
| Strain, strain background (*C. elegans*) | xwhEx60[P$_{ced-1}$ced-1, sur-5:: gfp] line3/ ced-1(e1735) | This paper | SNU111 | *Table 3*; Available from the Xiao Lab |
| Strain, strain background (*C. elegans*) | xwhEx61[P$_{ced-1}$ced-1(N962A), sur-5:: gfp] line2/ced-1(e1735) | This paper | SNU112 | *Table 3*; Available from the Xiao Lab |
| Strain, strain background (*C. elegans*) | xwhEx62[P$_{ced-1}$ced-1(N962A), sur-5:: gfp] line1/ced-1(e1735) | This paper | SNU113 | *Table 3*; Available from the Xiao Lab |
| Strain, strain background (*C. elegans*) | xwhEx63[P$_{ced-1}$ced-1(N962A), sur-5:: gfp] line3/ced-1(e1735) | This paper | SNU114 | *Table 3*; Available from the Xiao Lab |
| Strain, strain background (*C. elegans*) | xwhEx64[P$_{ced-1}$ced-1(Y965F), sur-5:: gfp] line2/ced-1(e1735) | This paper | SNU115 | *Table 3*; Available from the Xiao Lab |
| Strain, strain background (*C. elegans*) | xwhEx65[P$_{ced-1}$ced-1(Y965F), sur-5:: gfp] line1/ced-1(e1735) | This paper | SNU116 | *Table 3*; Available from the Xiao Lab |
| Strain, strain background (*C. elegans*) | xwhEx66[P$_{ced-1}$ced-1(Y965F), sur-5:: gfp] line3/ced-1(e1735) | This paper | SNU117 | *Table 3*; Available from the Xiao Lab |
| Strain, strain background (*C. elegans*) | xwhEx67[P$_{ced-1}$ced-1(Y1019F), sur-5:: gfp] line2/ced-1(e1735) | This paper | SNU118 | *Table 3*; Available from the Xiao Lab |
| Strain, strain background (*C. elegans*) | xwhEx68[P$_{ced-1}$ced-1(Y1019F), sur-5:: gfp] line1/ced-1(e1735) | This paper | SNU119 | *Table 3*; Available from the Xiao Lab |
| Strain, strain background (*C. elegans*) | xwhEx69[P$_{ced-1}$ced-1(Y1019F), sur-5:: gfp] line3/ced-1(e1735) | This paper | SNU120 | *Table 3*; Available from the Xiao Lab |
| Strain, strain background (*Escherichia coli*) | OP50 | Dr. Chonglin Yang (*Chen et al., 2010*) | N/A | |
| Strain, strain background (*E. coli*) | DH5α | TaKaRa | Cat# 9057 | |
| Strain, strain background (*E. coli*) | HT115 | Dr. Chonglin Yang (*Chen et al., 2010*) | N/A | |
| Strain, strain background (*E. coli*) | BL21 | TaKaRa | Cat# 9126 | |
| Strain, strain background (*E. coli*) | BL21(DE3) | Solarbio | Cat# C1400 | |
| Strain, strain background (*Saccharomyces cerevisiae*) | Y2HGold | Clontech | Cat# 630498 | |

*Continued on next page*

Continued

| Reagent type (species) or resource | Designation | Source or reference | Identifiers | Additional information |
|---|---|---|---|---|
| Antibody | Anti-CED-1 (rabbit polyclonal) | Dr. Chonglin Yang (*Chen et al., 2010*) | N/A | IB(1:1000) Available from the Xiao Lab |
| Antibody | Anti-β-actin (mouse polyclonal) | This paper | N/A | IB(1:1000) Available from the Xiao Lab |
| Antibody | Anti-FLAG tag (DYKDDDDK) (rabbit polyclonal) | Sigma-Aldrich | Cat# SAB1306078 | IB(1:1000) |
| Antibody | Anti-Ub(P4D1) IgG1 (mouse monoclonal) | Santa Cruz Biotechnology | Cat# SC-8017; RRID:AB_628423 | IB(1:500) |
| Antibody | Anti-GST (mouse monoclonal) | Engibody | Cat# AT0027 | IB(1:1000) |
| Antibody | Anti-GST (mouse polyclonal) | This paper | N/A | IB(1:1000) Available from the Xiao Lab |
| Antibody | Anti-CED-1 (mouse polyclonal) | This paper | N/A | IB(1:1000) Available from the Xiao Lab |
| Antibody | Anti-FLAG M2 antibody (mouse monoclonal) | Sigma-Aldrich | Cat# F1804; RRID:AB_262044 | IB(1:1000) |
| Antibody | Anti-C-MYC-antibody (rabbit polyclonal) | Sigma-Aldrich | Cat# SAB4301136 | IB(1:1000) |
| Antibody | Anti-GFP (rabbit polyclonal) | Engibody | Cat# 1598 | IB(1:1000) |
| Antibody | Anti-GFP (mouse polyclonal) | This paper | N/A | IB(1:1000) Available from the Xiao Lab |
| Antibody | Anti-HA-Tag(C29F4) (rabbit monoclonal) | Cell Signaling Technology | Cat# 3724; RRID:AB_1549585 | IB(1:1000) |
| Antibody | Anti-CED-6 (rabbit polyclonal) | Dr. Chonglin Yang | N/A | IB(1:1000) Available from the Xiao Lab |
| Antibody | Anti-CED-6 (mouse polyclonal) | This paper | N/A | IB(1:1000) Available from the Xiao Lab |
| Antibody | Anti-P-tyrosine (P-Tyr-100) (mouse monoclonal) | Cell Signaling Technology | Cat# 9411; RRID:AB_331228 | IB(1:1000) |
| Antibody | Anti-NCK-1 (mouse polyclonal) | This paper | N/A | IB(1:1000) Available from the Xiao Lab |
| Antibody | Peroxidase-conjugated AffiniPure Goat Anti-Rabbit IgG(H+L) | Jackson ImmunoResearch | Cat# 111-035-003; RRID:AB_2313567 | IB(1:10,000) |
| Antibody | Peroxidase-conjugated AffiniPure Goat Anti-Mouse IgG(H+L) | Jackson ImmunoResearch | Cat# 115-035-003; RRID:AB_10015289 | IB(1:10,000) |
| Commercial assay or kit | Anti-FLAG M2 Affinity Gel | Sigma-Aldrich | Cat# A2220; RRID:AB_10063035 | |
| Commercial assay or kit | Anti-FLAG M2 Magnetic Beads | Sigma-Aldrich | Cat# M8823; RRID:AB_2637089 | |
| Commercial assay or kit | Pierce Streptavidin Magnetic Beads | Thermo Fisher Scientific | Cat# 88817 | |
| Commercial assay or kit | Ni-NTA Superflow | QIAGEN | Cat# 1018611 | |
| Commercial assay or kit | Glutathione Sepharose 4B | GE Healthcare | Cat# 17-0756 | |
| Commercial assay or kit | Glutathione High Capacity Magnetic Agarose Beads | Sigma-Aldrich | Cat# G0924 | |
| Commercial assay or kit | PureProteome Protein A/G Mix Magnetic Beads | Millipore | Cat# LSKMAGAG | |
| Commercial assay or kit | Pierce Anti-HA Magnetic Beads | Thermo Scientific | Cat# 88837 | |
| Chemical compound, drug | Glutathione, reduced | VWR AMRESCO | Cat# 0399 | |
| Chemical compound, drug | Cycloheximide (CHX) | INALCO SPA, Milan, Italy | Cat# 1758-9310 | |

*Continued on next page*

*Continued*

| Reagent type (species) or resource | Designation | Source or reference | Identifiers | Additional information |
| --- | --- | --- | --- | --- |
| Chemical compound, drug | MG-132 | Selleck | Cat# S2619 | |
| Chemical compound, drug | Imidazole | Millipore | Cat# 288-32-4 | |
| Chemical compound, drug | TRIzol Reagent | Ambion | Cat# 15596018 | |
| Commercial assay or kit | Pro-Q Diamond Phosphoprotein Gel Stain | Invitrogen | Cat# P33301 | |
| Commercial assay or kit | HiScript III RT SuperMix for qPCR (+gDNA wiper) | Vazyme | Cat# R323 | |
| Commercial assay or kit | ChamQ Universal SYBR qPCR Master Mix | Vazyme | Cat# Q711 | |
| Recombinant DNA reagent | pGBKT7-*ced-1-ct* | This paper | N/A | Available from the Xiao Lab |
| Recombinant DNA reagent | pGBKT7-*ubc-21* | This paper | N/A | Available from the Xiao Lab |
| Recombinant DNA reagent | pGBKT7-*ced-1-nt* | This paper | N/A | Available from the Xiao Lab |
| Recombinant DNA reagent | pGBKT7-*ced-1-ct-A* | This paper | N/A | Available from the Xiao Lab |
| Recombinant DNA reagent | pGBKT7-*ced-1-ct-B* | This paper | N/A | Available from the Xiao Lab |
| Recombinant DNA reagent | pGBKT7-*ced-1-ct-C* | This paper | N/A | Available from the Xiao Lab |
| Recombinant DNA reagent | pGBKT7-*ced-1(Y965F)-ct* | This paper | N/A | Available from the Xiao Lab |
| Recombinant DNA reagent | pGBKT7-*ced-1(Y1019F)-ct* | This paper | N/A | Available from the Xiao Lab |
| Recombinant DNA reagent | pGBKT7-*ced-1(N962A)-ct* | This paper | N/A | Available from the Xiao Lab |
| Recombinant DNA reagent | pGADT7-*trim-21* | This paper | N/A | Available from the Xiao Lab |
| Recombinant DNA reagent | pGADT7-*f43c11.7* | This paper | N/A | Available from the Xiao Lab |
| Recombinant DNA reagent | pGADT7-*k01g5.1* | This paper | N/A | Available from the Xiao Lab |
| Recombinant DNA reagent | pGADT7-*trim-21-ΔRING* | This paper | N/A | Available from the Xiao Lab |
| Recombinant DNA reagent | pGADT7-*trim-21-ΔBBOX* | This paper | N/A | Available from the Xiao Lab |
| Recombinant DNA reagent | pGADT7-*trim-21-ΔCC* | This paper | N/A | Available from the Xiao Lab |
| Recombinant DNA reagent | pGADT7-*trim-21-nt* | This paper | N/A | Available from the Xiao Lab |
| Recombinant DNA reagent | pGADT7-*trim-21-ct* | This paper | N/A | Available from the Xiao Lab |
| Recombinant DNA reagent | pGADT7-*ced-6-PTB* | This paper | N/A | Available from the Xiao Lab |
| Recombinant DNA reagent | pGADT7-*src-1* | This paper | N/A | Available from the Xiao Lab |
| Recombinant DNA reagent | pGADT7-*src-1(1-261aa)* | This paper | N/A | Available from the Xiao Lab |
| Recombinant DNA reagent | pGADT7-*src-1(229-533aa)* | This paper | N/A | Available from the Xiao Lab |
| Recombinant DNA reagent | pGADT7-*src-1-ΔSH2* | This paper | N/A | Available from the Xiao Lab |
| Recombinant DNA reagent | pGADT7-*src-1-ΔSH3* | This paper | N/A | Available from the Xiao Lab |
| Recombinant DNA reagent | pGADT7-*src-1-ΔTyrkc* | This paper | N/A | Available from the Xiao Lab |
| Recombinant DNA reagent | pGADT7-*nck-1* | This paper | N/A | Available from the Xiao Lab |
| Recombinant DNA reagent | pGEX-KG-*ced-1-ct* | This paper | N/A | Available from the Xiao Lab |
| Recombinant DNA reagent | pGEX-KG-*trim-21* | This paper | N/A | Available from the Xiao Lab |
| Recombinant DNA reagent | pGEX-KG-*ubc-21* | This paper | N/A | Available from the Xiao Lab |
| Recombinant DNA reagent | pGEX-KG-*ced-6* | This paper | N/A | Available from the Xiao Lab |
| Recombinant DNA reagent | pGEX-KG-*htrim21* | This paper | N/A | Available from the Xiao Lab |
| Recombinant DNA reagent | pET28a-*ced-1-ct-flag* | This paper | N/A | Available from the Xiao Lab |
| Recombinant DNA reagent | pET28a-*myc-trim-21* | This paper | N/A | Available from the Xiao Lab |

*Continued on next page*

*Continued*

| Reagent type (species) or resource | Designation | Source or reference | Identifiers | Additional information |
|---|---|---|---|---|
| Recombinant DNA reagent | pET28a-*ha-ubq-2* | This paper | N/A | Available from the Xiao Lab |
| Recombinant DNA reagent | pET28a-*ha-ubq-2(K48R)* | This paper | N/A | Available from the Xiao Lab |
| Recombinant DNA reagent | pET28a-*ha-ubq-2(K63R)* | This paper | N/A | Available from the Xiao Lab |
| Recombinant DNA reagent | pET28a-*uba-1* | This paper | N/A | Available from the Xiao Lab |
| Recombinant DNA reagent | pET28a-*myc-ubc-21* | This paper | N/A | Available from the Xiao Lab |
| Recombinant DNA reagent | pET28a-*ha-ced-6* | This paper | N/A | Available from the Xiao Lab |
| Recombinant DNA reagent | pET28a-*ced-1(Y965F)-ct-flag* | This paper | N/A | Available from the Xiao Lab |
| Recombinant DNA reagent | pET28a-*ced-1(Y1019F)-ct-flag* | This paper | N/A | Available from the Xiao Lab |
| Recombinant DNA reagent | pET28a-*ced-1(N962A)-ct-flag* | This paper | N/A | Available from the Xiao Lab |
| Recombinant DNA reagent | pET28a-*myc-src-1* | This paper | N/A | Available from the Xiao Lab |
| Recombinant DNA reagent | pET28a-*ha-vha-10* | This paper | N/A | Available from the Xiao Lab |
| Recombinant DNA reagent | pET28a-*myc-trim-21-ΔRING* | This paper | N/A | Available from the Xiao Lab |
| Recombinant DNA reagent | pET28a-*myc-trim-21-ΔBBOX* | This paper | N/A | Available from the Xiao Lab |
| Recombinant DNA reagent | pET28a-*myc-trim-21-ΔCC* | This paper | N/A | Available from the Xiao Lab |
| Recombinant DNA reagent | pET28a-*ha-nck-1* | This paper | N/A | Available from the Xiao Lab |
| Recombinant DNA reagent | pET28a-*ha-megf10-ct* | This paper | N/A | Available from the Xiao Lab |
| Recombinant DNA reagent | pcDNA3.1-*myc-src-1* | This paper | N/A | Available from the Xiao Lab |
| Recombinant DNA reagent | pcDNA3.1-*ha-ced-1-ct-miniturbo* | This paper | N/A | Available from the Xiao Lab |
| Recombinant DNA reagent | pcDNA3.1-*myc-ced-1-ct-miniturbo* | This paper | N/A | Available from the Xiao Lab |
| Recombinant DNA reagent | pcDNA3.1-*ha-nck-1* | This paper | N/A | Available from the Xiao Lab |
| Recombinant DNA reagent | pcDNA3.1-*myc-trim-21-miniturbo* | This paper | N/A | Available from the Xiao Lab |
| Recombinant DNA reagent | pPD49.26-P$_{ced-1}$*ced-1-flag* | This paper | N/A | Available from the Xiao Lab |
| Recombinant DNA reagent | pPD49.78-*trim-21-flag* | This paper | N/A | Available from the Xiao Lab |
| Recombinant DNA reagent | pPD49.83-*trim-21-flag* | This paper | N/A | Available from the Xiao Lab |
| Recombinant DNA reagent | pPD49.26-P$_{ced-1}$*trim-21-flag* | This paper | N/A | Available from the Xiao Lab |
| Recombinant DNA reagent | pPD49.26-P$_{ced-1}$*mcherry-ced-6* | This paper | N/A | Available from the Xiao Lab |
| Recombinant DNA reagent | pPD49.26-P$_{ced-1}$*trim-21-gfp* | This paper | N/A | Available from the Xiao Lab |
| Recombinant DNA reagent | pPD49.26-P$_{ced-1}$*ced-1(N962A)-flag* | This paper | N/A | Available from the Xiao Lab |
| Recombinant DNA reagent | pPD49.78-*ced-1-ct-flag* | This paper | N/A | Available from the Xiao Lab |
| Recombinant DNA reagent | pPD49.83-*ced-1-ct-flag* | This paper | N/A | Available from the Xiao Lab |
| Recombinant DNA reagent | pPD49.26-P$_{ced-1}$*vha-10-mcherry* | This paper | N/A | Available from the Xiao Lab |
| Recombinant DNA reagent | pPD95.77-P$_{trim-21}$*trim-21-mcherry* | This paper | N/A | Available from the Xiao Lab |
| Recombinant DNA reagent | pPD95.77-P$_{trim-21}$*trim-21-gfp* | This paper | N/A | Available from the Xiao Lab |
| Recombinant DNA reagent | pPD95.77-P$_{trim-21}$*trim21* | This paper | N/A | Available from the Xiao Lab |
| Recombinant DNA reagent | pPD49.26-P$_{ced-1}$*ced-1-gfp* | This paper | N/A | Available from the Xiao Lab |
| Recombinant DNA reagent | pPD49.26-P$_{ced-1}$*ced-1(Y1019F)-gfp* | This paper | N/A | Available from the Xiao Lab |
| Sequence-based reagent | *trim-21* sgRNA targeting sequence | This paper | N/A | GACTTCTCAAGTGAGGAGGATGG |
| Sequence-based reagent | *ubc-21* sgRNA targeting sequence | This paper | N/A | TCGCATTGGCACGGGTCACACGG |
| Sequence-based reagent | *ced-1-flag* sgRNA targeting sequence | This paper | N/A | TGCGAACAAAAAACGTGCTCAGG |

*Continued*

| Reagent type (species) or resource | Designation | Source or reference | Identifiers | Additional information |
|---|---|---|---|---|
| Sequence-based reagent | ha-ubq-2 sgRNA targeting sequence | This paper | N/A | AATCTTCGTCAAGACTCTGACGG |
| Sequence-based reagent | *ced-1(N962A)-flag* sgRNA targeting sequence | This paper | N/A | GGCCGAGAATTCCAGAATCCCCT |
| Sequence-based reagent | *ced-1(Y1019F)-flag* sgRNA targeting sequence | This paper | N/A | CCCAGACGACTACGCCTCCCTGG |
| Sequence-based reagent | *src-1* sgRNA targeting sequence | This paper | N/A | GCGATCGGGAGGCAGTGATATGG |
| Sequence-based reagent | *ha-ubq-2(K48R)* sgRNA targeting sequence | This paper | N/A | AATTTCAGGAAAGCAACTCGAGG |
| Sequence-based reagent | *ha-ubq-2(K63R)* sgRNA targeting sequence | This paper | N/A | TTGGTGCTCCGTCTTCGTGGAGG |
| Sequence-based reagent | *ced-6* sgRNA targeting sequence | This paper | N/A | GTCGGTGGAAATAATATTAATGG |
| Sequence-based reagent | *nck-1* sgRNA targeting sequence | This paper | N/A | ATACGATTATTTAGCACAAGAGG |
| Sequence-based reagent | *ha-vha-10* sgRNA targeting sequence | This paper | N/A | CAGTACCGAAAACCTTAAAATGG |
| Sequence-based reagent | QPCR, *tbg-1*, forward | This paper | N/A | cgtcatcagcctggtagaaca |
| Sequence-based reagent | QPCR, *tbg-1*, reverse | This paper | N/A | tgatgactgtccacgttgga |
| Sequence-based reagent | QPCR, *ced-1*, forward | This paper | N/A | ggatggactggaaaacattgtg |
| Sequence-based reagent | QPCR, *ced-1*, reverse | This paper | N/A | cggattcgcattgacattgg |
| Software, algorithm | SMART | EMBL | http://smart.embl-heidelberg.de/ | |
| Software, algorithm | ImageJ | NIH | https://imagej.nih.gov/ij/download.html | |
| Software, algorithm | GraphPad Prism 8 | GraphPad Software | https://www.graphpad.com/scientific-software/prism/ | |
| Software, algorithm | ClustalW2 | EMBL-EBI | https://www.ebi.ac.uk/Tools/msa/clustalw2/ | |
| Software, algorithm | ZEN 2 pro | ZEISS | https://www.zeiss.com/microscopy/int/products/microscope-software/zen.html | |

## *C. elegans* strains and genetics

The Bristol strain N2 of the nematode *C. elegans* was used as the wild-type. Strains of *C. elegans* were cultured at 20°C on nematode growth media (NGM) and maintained by standard protocols (*Brenner, 1974*). The mutant alleles used in this study were as follows: linkage group (LG) Ⅰ: *ced-1(e1735)*; LG Ⅱ: *trim-21(xwh12)*; LG Ⅲ: *ced-4(n1162)*, *ced-6(n1813)*, *ced-6(xwh25)*, *ced-7(n1892)*, *src-1(xwh26); +/hT2*, LG Ⅳ: *ced-2(n1994)*, *ced-3(n717)*, *ced-5(n1812)*; LG Ⅹ: *ced-8(n1891)*, *ubc-21(xwh15)*. To generate transgenic strains, we injected 1–30 ng/µl DNA into *C. elegans* germlines. Transgenic animals carrying extrachromosomal arrays (*xwhEx*) were generated by standard microinjection methods, and integrated genome arrays (*xwhIs*) were acquired by UV irradiation to achieve stable expression from arrays with low copy numbers. All strains used in this study from other LABs or carrying integrated/ extrachromosomal arrays were generated in our laboratories (backcrossed with N2 2–6 times) are listed in 'Key resources table'.

## Cell lines

Human embryonic kidney cells 293 (HEK293T) were obtained from FuHeng Cell Center (Shanghai). The cells were verified by STR profiling and tested to be free of mycoplasma contamination by stand PCR methods.

## RNAi experiments

Bacterial feeding assays were used for RNAi experiments (*Chen et al., 2010*). NGM plates containing 50 µg/ml ampicillin and 0.1 mM IPTG were seeded with bacteria expressing either control dsRNA or

dsRNA of genes. Worms at different growth stages were fed with bacteria expressing dsRNA for 48 hr and subjected to further analysis by immunoblotting. Embryos were fed bacteria expressing dsRNA until reaching the young adult stage, after which the different embryo or germline stage corpses were counted.

## Quantitative RT-PCR

Worms were washed off plates and washed several times with M9 buffer until the supernatant was clear. Worm pellets were resuspended in TRIzol reagent (Ambion, Austin, TX,, CAT# 15596018). Samples were frozen, thawed, and homogenized completely with tissue grinders. Total RNA was isolated by chloroform extraction, followed by ethanol precipitation and DNase treatment. cDNA was then synthesized using HiScript III RT SuperMix for qPCR (+gDNA wiper) (Vazyme, CAT# R323) according to the manufacturer's instructions. Quantitative RT-PCR was carried out using the ChamQTM Universal SYBR qPCR Master Mix (Vazyme, CAT# Q711) according to the manufacturer's instructions. Transcript quantification was normalized to *tbg-1*.

## MG-132/chloroquine treatment of worms

Worms were grown on NGM plates and subsequently transferred to plates containing different concentrations of MG-132 or CQ mixed with OP50. Immunoblotting was performed after 24 hr.

## Immunoblotting

Mix-staged worms were collected with M9 buffer and stored at –80°C. After thawing, the worms were resuspended in RIPA lysis buffer (25 mM Tris–HCl, pH 7.4, 150 mM NaCl, 1% sodium deoxycholate, 0.1% SDS) containing protease inhibitor cocktail, homogenized completely with tissue grinders, and centrifuged at $15,000 \times g$ at 4°C for 20 min to remove debris. Samples were quantified by the BCA method, then 15–30 µg samples containing 1×SDS loading buffer with 5% 2-mercaptoethanol were boiled at 100°C for 5 min. Worm lysates were loaded onto SDS-PAGE and transferred onto PVDF membranes, probed with primary antibodies (β-actin loaded control) and HRP-conjugated secondary antibodies, and developed with the Immobilon Western chemiluminescent HRP substrate (Millipore). PVDF membranes were made visible using a MiniChemi610 (SAGEcreation). All the antibodies used in this study are listed in 'Key resources table'.

## Co-immunoprecipitation

To detect the interaction of CED-1-CT with SRC-1, VHA-10, or NCK-1, and TRIM-21 with NCK-1, 293T cells in a 10 cm plate were transfected with 7 µg HA-CED-1-CT-miniTurbo or MYC-CED-1-CT-miniTurbo, 7 µg MYC-SRC-1 or HA-VHA-10 or HA-NCK-1, 7 µg MYC-TRIM-21-miniTurbo, and 7 µg HA-NCK-1 plasmids via 28 µg PEI. Biotin (0.5 mM) was added to a 10 cm plate after transfection for 48 hr to handle for 45 min. Cells were scraped off the plates, pelleted by centrifugation at $1000 \times g$ for 3 min, and washed with PBS buffer two times. Cells were resuspended in cell lysis buffer (25 mM Tris–HCl, pH 7.6, 150 mM NaCl, 10 mM $MgCl_2$, 1% NP-40) containing protease inhibitor cocktail, sonicated for 15 s, and centrifuged at $20,000 \times g$ at 4°C for 20 min to remove debris. The samples were quantitated using the BCA method. Proteins (2 mg) for co-IP were with pre-washed streptavidin magnetic beads (Thermo, 88817) and incubated with agitation at 4°C for 2 hr. Beads were washed three times with TBST buffer (50 mM Tris–HCl, pH 7.6, 150 mM NaCl, 0.1% Tween-20). Then, 20 µl of 2×SDS loading buffer with 5% 2-mercaptoethanol was added to the beads and boiled for 5 min at 100°C. The samples were analyzed by WB.

Mix-staged worms were collected and washed with M9 buffer. Samples were resuspended in lysis buffer (50 mM Tris–HCl, pH 7.4, 150 mM NaCl, 0.5% sodium deoxycholate, 1% NP-40, 10% glycerol) containing a protease inhibitor cocktail and were homogenized completely with tissue grinders, and centrifuged at $15,000 \times g$ and 4°C for 20 min to remove debris for co-IP. Proteins (4 mg) carrying FLAG-tag were immunoprecipitated with anti-FLAG M2 affinity gel (Sigma, A2220) and incubated with agitation at 4°C for 4–8 hr. Beads were then washed three times with TBS buffer and analyzed by WB. To detect the interaction of CED-6 with TRIM-21, or CED-1(WT, Y1019F) with NCK-1, worm lysates were incubated with anti-CED-6 or anti-GFP at 4°C for 4 hr. Pre-washed PureProteome Protein A/G Mix Magnetic Beads (Millipore, LSKMAGAG) were added and rotated for an additional 6 hr at 4°C. The beads were washed three times with TBS buffer and analyzed using WB. To detect the

interaction of CED-1 with VHA-10, worm lysates were incubated with anti-HA beads (Thermo, 88837) at 4°C for 6 hr. The beads were washed three times with TBS buffer and analyzed using WB.

## Yeast two-hybrid analysis

Each indicated gene cloned into pGADT7 or pGBKT7 was transformed into a Y2HGold yeast strain to examine protein–protein interactions according to the manufacturer's instructions (Clontech) for the Y2H system. Transformed yeasts were cultured on synthetic complete agar-Leu-Trp medium for 3–5 days. Single yeast colonies were transferred into the liquid culture medium and shaken for 8–12 hr before dropping onto -Leu-Trp and -Leu-Trp-His dropout medium to test for interactions. Images of yeast were taken after culturing at 30°C for 3–5 days. All plasmids used for the Y2H assays are listed in 'Key resources table'.

## CRISPR/Cas9-mediated genome editing

For the generation of null allele mutants and insertion of tags at specific sites, a previously described CRISPR/Cas9 method was used (*Paix et al., 2014*). Briefly, specific single-guide RNA (sgRNA) target sites were selected using the CRISPR design tool (http://crispr.mit.edu) and introduced into the vector ($pPD162\text{-}P_{eft\text{-}3}CAS9\text{-}P_{u6}sgRNA$) that expresses the CAS9 enzyme and sgRNA. Target sgRNA plasmid 20 ng/µl, target ssODN 2 µM (the repair oligo, containing 40–50 bp homologous arms), *dpy-10* sgRNA 20 ng/µl, and *dpy-10* ssODN 2 µM (the selection marker) were co-injected into worms. Then, 4 days later, dumpy or roller F1 worms were confirmed by PCR and Sanger sequencing. All the strains were backcrossed with N2 before use. The sgRNA target and the repairing oligo sequences used in this study are listed in 'Key resources table'.

## Heat shock assay

Worms were grown on NGM plates seeded with OP50 at 20°C. When multiple worms grew to adulthood, the samples were subjected to heat shock at 33°C for 1 hr and recovered at 20°C for 2 hr. Subsequently, worms were washed off NGM plates with M9 and washed twice with M9 buffer, then stored at –80°C for co-IP experiments.

## Purification of recombinant proteins

To purify recombinant proteins, cDNAs for target genes were cloned into pGEX-KG or pET28a vectors. GST-CED-1-CT$^{931\text{-}1111}$, GST-TRIM-21, GST-UBC-21, GST-CED-6, and GST-TRIM21 proteins were expressed in BL21 strain *E. coli* induced by IPTG. GST-tagged proteins in BL21 were collected by centrifugation and lysed in prokaryotic expression lysis buffer (25 mM Tris–HCl, pH 7.6, 150 mM NaCl, 0.5% NP-40) containing 1 mg/ml lysozyme and 1 mM PMSF for 30 min on ice, followed by sonication. GST-tagged proteins were then isolated using Glutathione Sepharose 4 B (GE Healthcare, 17-0756). After incubation at 4°C for 2 hr, beads were washed three times with prokaryotic expression lysis buffer and eluted with 20 mM or 30 mM glutathione reduced (AMERSCO, 0399). 6HIS-CED-1-CT-FLAG-6HIS, 6HIS-MYC-TRIM-21-6HIS, 6HIS-MYC-TRIM-21-NT$^{1\text{-}137}$-6HIS, 6HIS-MYC-TRIM-21-CT$^{137\text{-}292}$-6HIS, 6HIS-UBA-1-6HIS, 6HIS-HA-UBQ-2(ubiquitin region)–6HIS, 6HIS-MYC-UBC-21-6HIS, 6HIS-HA-CED-6-6HIS, 6HIS-CED-1-CT(N962A, Y965F, Y1019F)-FLAG-6HIS, 6HIS-MYC-SRC-1-6HIS, 6HIS-HA-VHA-10-6HIS, 6HIS-MYC-TRIM-21(ΔRING, ΔBBOX, ΔCC)–6HIS, 6HIS-HA-NCK-1-6HIS, and 6HIS-HA-MEGF10-CT-6HIS proteins were purified from the *E. coli* BL21(DE3) strain induced by IPTG. HIS-fusion-proteins in BL21(DE3) were collected by centrifugation and resuspended in prokaryotic expression lysis buffer containing 1 mg/ml lysozyme and 1 mM PMSF. After ultrasonication, HIS-fusion proteins were purified using Ni-NTA Superflow (QIAGEN). After incubation at 4°C for 4 hr, beads were eluted with 10/30/100/150 mM imidazole (Merck) in prokaryotic expression lysis buffer. Purified proteins were stored in a prokaryotic expression lysis buffer containing 10% glycerol at –80°C.

## GST pull-down assay

To investigate the protein–protein interactions in vitro, 2 µg GST-tagged proteins were immobilized on Glutathione High Capacity Magnetic Agarose Beads (Sigma, G0924) and incubated with 1 µg of 6HIS-fusion-proteins for 1 hr at 4°C in a binding buffer (25 mM Tris–HCl, pH 7.4, 150 mM NaCl, 0.5% NP-40). Beads were washed 3–5 times using washing buffer (25 mM Tris–HCl, pH 7.4, 300 mM NaCl, 0.5% NP-40) and boiled with 2×SDS loading buffer at 100°C for 5 min.

## Quantification and duration of embryonic and gonadal cell corpses

Cell corpses were quantified by button-like morphology using Nomarski optics. For somatic cell corpses, at least 15 embryos at each developmental stage (comma, 1.5-fold, 2-fold, 2.5-fold, 3-fold, and 4-fold) in each strain were scored as the head regions of embryonic cell corpses. Cell corpses in the germline meiotic region of one gonad arm in each of at least 15 animals were scored at indicated adult ages (12, 24, 36, 48, and 60 hr after the L4 larval stage). The average numbers of embryonic and gonadal cell corpses were shown and compared with those of other transgenic worms using unpaired *t*-tests.

Four-dimensional microscopy analysis was used to examine the cell corpse duration. For somatic cell corpses, embryos (two cell stage) were isolated from adult worms and placed in egg salt buffer (118 mM NaCl and 48 mM KCl), mounted on 2% agar pads, sealed with beeswax and Vaseline (1:1), and observed under Nomarski optics at 20°C. Images in 30 Z-sections (1.0 μm/section) were captured every minute for 400 min using an Axio Imager M2 microscope (ZEISS). Images were processed and viewed using ZEN 2 pro software (ZEISS). For gonadal cell corpses, adult worms were mounted in M9 buffer (3 g $KH_2PO_4$, 6 g $Na_2HPO_4$, 5 g NaCl, 1 ml 1 M $MgSO_4$, $H_2O$ to 1 l) containing 2 mM levamisole, sealed with beeswax and Vaseline (1:1), and observed at 20°C. Images were captured using an Axio Imager M2 microscope (ZEISS), and germline corpses were recorded.

## Ubiquitination of CED-1 in vitro and in vivo

To examine ubiquitin modification in vitro, 1 μg GST-CED-1-CT, 0.1 μg UBA-1, 0.25 μg MYC-UBC-21, 0.6 μg GST-TRIM-21, and 2 μg HA-UBQ-2 (WT, K48R, K63R) were incubated in ubiquitin ligation buffer (25 mM Tris–HCl, pH 7.4, 100 mM NaCl, 10 mM $MgCl_2$, 2 mM ATP, 0.5 mM DTT) at 20°C for 2 hr. The reaction was immobilized on pre-washed anti-FLAG M2 magnetic beads (Sigma, M8823) at 4°C for 1 hr and washed three times with TBS buffer. The beads were boiled for 5 min in 2×SDS loading buffer and analyzed by WB.

To confirm the ubiquitination of CED-1 in vivo (*Liu et al., 2018*), mix-staged worms (*ced-1::flag;ha::ubq-2, ced-1::flag;ha::ubq-2-K48R, ced-1::flag;ha::ubq-2-K63R*, or worms treated with tagged gene RNAi) were collected and washed in M9 buffer. Harvested worms were mixed with lysis buffer (50 mM Tris–HCl, pH 7.6, 150 mM NaCl, 0.5% sodium deoxycholate, 1% NP-40, 10% glycerol) containing a protease inhibitor cocktail, homogenized completely with tissue grinders, and centrifuged at 15,000 × *g* at 4°C for 20 min to remove debris. Samples of 2–4 mg proteins were immobilized with anti-FLAG M2 affinity gel and incubated with agitation at 4°C for 4 hr, washed three times with washing buffer (50 mM Tris–HCl, pH 7.6, 300 mM NaCl), and subjected to WB.

## In vitro phosphorylation assay and mass spectrometry of phosphorylation sites

To demonstrate that SRC-1 could be phosphorylated CED-1 in vitro, 1 μg MYC-SRC-1 and 2 μg CED-1-CT-FLAG (WT, Y1019F) were incubated in phosphorylation reaction buffer (20 mM Tris–HCl, pH 7.4, 100 mM NaCl, 12 mM $MgCl_2$, 1 mM ATP) at 25°C for 1 hr. The reactions were terminated by boiling the sample for 5 min with 1×SDS loading buffer and running SDS-PAGE gel followed by Coomassie brilliant blue staining, Pro-Q phosphoprotein staining, and immunoblotting to monitor phosphorylated tyrosine. SDS-PAGE gel was fixed in fixing solution (50% methanol, 10% acetic acid) for 30 min and washed three times with distilled water, for 10 min each time. The gel was then stained with Pro-Q Diamond Phosphoprotein Gel Stain (Invitrogen, P33301) according to the manufacturer's instructions for 90 min and destained with destaining buffer (20% acetonitrile, 50 mM sodium acetate, pH 4.0) three times for 30 min each time. Finally, the gel was washed with distilled water for 10 min and then scanned under UV irradiation.

To identify the phosphorylation site of CED-1, samples were separated by SDS-PAGE and visualized by Coomassie brilliant blue staining, and the target bands were then cut out. For in-gel tryptic digestion, the gel pieces were stained with 50 mM $NH_4HCO_3$ in 50% acetonitrile (v/v) until clear. Gel pieces were dehydrated with 100 μl of 100% acetonitrile for 5 min, the liquid was removed, and the gel pieces were rehydrated in 10 mM dithiothreitol and incubated at 56°C for 60 min. Gel pieces were again dehydrated in 100% acetonitrile, the liquid was removed, and the gel pieces were rehydrated with 55 mM iodoacetamide. The samples were incubated at room temperature in the dark for 45 min. The gel pieces were washed with 50 mM $NH_4HCO_3$ and dehydrated with 100% acetonitrile. They were

then rehydrated with 10 ng/μl trypsin resuspended in 50 mM $NH_4HCO_3$ on ice for 1 hr. The excess liquid was removed and the gel pieces were digested with trypsin at 37°C overnight. Peptides were extracted with 50% acetonitrile/5% formic acid, followed by 100% acetonitrile. Peptides were dried to completion and resuspended in 2% acetonitrile/0.1% formic acid. The tryptic peptides were dissolved in 0.1% formic acid (solvent A) and directly loaded onto a lab-made reversed-phase analytical column (15 cm length, 75 μm i.d.). The gradient comprised an increase from 7% to 25% solvent B (0.1% formic acid in 98% acetonitrile) over 18 min, 25% to 38% over 6 min, and climbing to 80% in 3 min, then holding at 80% for the last 3 min, all at a constant flow rate of 450 nl/min on an EASY-nLC 1000 UPLC system. The peptides were subjected to NSI source followed by tandem mass spectrometry (MS/MS) in OrbitrapFusion (Thermo) coupled online to the UPLC. The electrospray voltage applied was 2.0 kV. The m/z scan range was 350–1550 for a full scan, and intact peptides were detected in the Orbitrap at a resolution of 60,000. Peptides were then selected for MS/MS using an NCE setting of 35, and the fragments were detected in the Orbitrap at a resolution of 15,000 following a data-dependent procedure that alternated between one MS scan followed by 20 MS/MS scans with a 15.0 s dynamic exclusion. Automatic gain control (AGC) was set at 5E4. The resulting MS/MS data were processed using the Proteome Discoverer 1.3. Tandem mass spectra were searched against selected databases. Trypsin/P (or other enzymes, if any) was specified as a cleavage enzyme, allowing up to two missing cleavages. The mass error was set to 10 ppm for precursor ions and 0.02 Da for fragment ions. Carbamidomethyl on Cys was specified as a fixed modification, and oxidation on Met and modification were specified as variable modifications. Peptide confidence was set at a high value, and the peptide ion score was set at >20.

## Cycloheximide (CHX) treatment and protein degradation in vivo

Mix-staged worms were grown on NGM plates, and then transferred to liquid culture containing 100 ml S-Basal Medium (5.9 g NaCl, 50 ml 1 M $KPO_4$, pH 6.0, 1 ml 5 mg/ml cholesterol, $ddH_2O$ to 1 l), 300 μl 1 M $MgSO_4$, 300 μl 1 M $CaCl_2$, 1 ml 100× trace metal solution (0.346 g $FeSO_4 \bullet 7H_2O$, 0.93 g $Na_2EDTA$, 0.098 g $MnCl_2 \bullet 4H_2O$, 0.012 g $CuSO_4 \bullet 5H_2O$, $ddH_2O$ to 500 ml), and 1 ml 1 M potassium citrate (pH 6.0). A group of worms marked as untreated (0 hr). 0.2 mg/ml CHX ($C_{15}H_{23}NO_4$, INALCO SPA, Milan, Italy, 1758-9310) was added to the liquid culture. Worms were collected every 3 hr, twice. The collected worms (0, 3, and 6 hr) were resuspended in RIPA lysis buffer containing protease inhibitor cocktail, homogenized completely with tissue grinders, and centrifuged at 15,000 × *g* at 4°C for 20 min to remove debris. Proteins (0, 3, and 6 hr) were detected by immunoblotting using β-actin and CED-1 antibodies.

## Quantification of phagosomal markers and AO staining

The *trim-21*, *ubc-21* mutant worms carrying phagosomal markers (CED-1::GFP, GFP::RAB-5, GFP::RAB-7, mCHERRY::RAB-7, LAAT-1::GFP, LAAT-1::mCHERRY, NUC-1::mCHERRY, or CPL-1::mChiOnt) or transgenic worms carrying phagosomal markers that were treated with control and *vha-10* RNAi were mounted on 2% agar pads. Images of the total number of cell corpses and the number of cell corpses that were labeled with different phagosomal markers were captured using an Axio Imager M2 microscope (ZEISS). The percentage of embryonic or gonadal cell corpses labeled by phagosomal markers was determined by dividing the number of labeled cell corpses by the total number of cell corpses.

To detect lysosomal acidification, adult worms (36–48 hr after L4 molt) were soaked in 0.1 mg/ml AO containing a small amount of OP50 at room temperature and left in the dark for 1 hr. After treatment, worms were transferred to NGM plates seeded with OP50 at room temperature in the dark for 2 hr and mounted on 2% agar pads. Images were captured using DIC and fluorescence microscopy.

## Microscopy and time-lapse imaging

Worms tagged with fluorescence were imaged under DIC and fluorescence using an Axio Imager M2 microscope (ZEISS). Images were processed and viewed using ZEN 2 pro software (ZEISS). An Immersol 518F oil (Zeiss) was used. All the images were captured at 20°C (*Gan et al., 2019*).

Time-lapse imaging of HIS-24::mCherry in N2 and *trim-21* mutants was performed at 20°C under a ×100 oil objective using an Axio Imager M2 microscope (ZEISS). Images were focused on specific corpses and were taken every few minutes until the corpses disappeared.

## Immunoprecipitation, silver staining, and mass spectrometry analysis

Mix-staged worms ($P_{hsp-16}$ced-1-ct::flag) were grown on NGM plates seeded with OP50 at 20°C. Half of the plates were subjected to heat shock, and the rest were maintained at control conditions. Worms were harvested from the plates and washed three times with M9 buffer three times. Worms were then mixed with lysis buffer (50 mM Tris–HCl, pH 7.6, 150 mM NaCl, 0.5% sodium deoxycholate, 1% NP-40, 10% glycerol) containing a protease inhibitor cocktail, homogenized completely with tissue grinders, and centrifuged at 15,000 × $g$ at 4°C for 20 min to remove debris. Proteins were used for immuno-precipitation with an anti-FLAG M2 affinity gel. The reactions were incubated with agitation at 4°C for 8 hr. The bound proteins were washed three times with washing buffer (50 mM Tris–HCl, pH 7.6, 300 mM NaCl) and analyzed by WB.

For silver staining, IP samples were examined by SDS-PAGE. The gel was fixed in a formaldehyde fixing solution (40% methanol and 0.0185% formaldehyde) for 30 min. Later, the fixing solution was discarded, and the gel was washed twice with distilled water for 10 min each time. The gel was first incubated with 0.2 g/l $Na_2S_2O_3$ for 10 min, then incubated with 0.1% silver nitrate for 45 min. Then, the gel was soaked in thiosulfate developing solution (3% sodium carbonate, 0.0004% sodium thiosulfate, and 0.0185% formaldehyde). The process was terminated using 0.115 M citric acid.

Protein digestion (250 µg per sample) was performed according to the FASP procedure. Next, 100 µl 0.05 M iodoacetamide in UA buffer (8 M urea, 150 mM Tris–HCl, pH 8.0) was added to block reduced cysteine residues, and the samples were incubated for 20 min in the dark. The filter was washed with 100 µl UA buffer three times, and then twice with 100 µl 25 mM $NH_4HCO_3$. Finally, the protein suspension was digested with 3 µg trypsin (Promega) in 40 µl 25 mM $NH_4HCO_3$ overnight at 37°C, and the resulting peptides were collected as a filtrate. The peptide of each sample was desalted on C18 Cartridges (Empore SPE Cartridges C18 [standard density], bed I.D. 7 mm, volume 3 ml, Sigma), then concentrated by vacuum centrifugation and reconstituted in 40 µl of 0.1% (v/v) trifluoroacetic acid. MS experiments were performed on a Q Exactive mass spectrometer coupled to an EASY nLC (Proxeon Biosystems, now Thermo Fisher Scientific). The peptide (5 µg) was loaded onto a C18-reversed phase column (Thermo Scientific EASY Column, 10 cm long, 75 µm inner diameter, 3 µm resin) in buffer A (2% acetonitrile and 0.1% formic acid) and separated with a linear gradient of buffer B (80% acetonitrile and 0.1% formic acid) at a flow rate of 250 nl/min controlled by IntelliFlow technology over 60 min. MS data were acquired using a data-dependent top 10 method, dynamically choosing the most abundant precursor ions from the survey scan (300–1800 m/z) for HCD fragmentation. The determination of the target value was based on predictive automatic gain control (pAGC). The dynamic exclusion duration was 25 s. Survey scans were acquired at a resolution of 70,000 at m/z 200, and the resolution for HCD spectra was set to 17,500 at m/z 200. The normalized collision energy was 30 eV, and the underfill ratio, which specifies the minimum percentage of the target value likely to be reached at the maximum fill time, was defined as 0.1%. The instrument was run with peptide recognition mode enabled. MS experiments were performed in triplicate for each sample. The MS data were analyzed using MaxQuant software version 1.5.5.1. MS data were searched against the UniProt *Caenorhabditis elegans* database (27,767 entries). An initial search was performed using a precursor mass window of 6 ppm. The search followed an enzymatic cleavage rule of Trypsin/P and allowed a maximum of two missed cleavage sites and a mass tolerance of 20 ppm for fragment ions. Carbam-idomethylation of cysteines was defined as a fixed modification, whereas methionine oxidation was defined as a variable modification for database searching. The cutoff of the global false discovery rate (FDR) for peptide and protein identification was set to 0.01.

## Statistical analyses

Quantitative data were analyzed using GraphPad Prism 8 and Microsoft Excel (Microsoft Office). Error bars represent SEM. Two-tailed unpaired Student's *t*-tests were performed to compare the mean values between the two groups. Statistical significance is designated as *p<0.05, **p<0.01, and ***p<0.001.

## Acknowledgements

We thank Drs. Chonglin Yang, Drs. Ding Xue and the C. elegans Genetic Center for *C. elegans* strains. This work was partially supported by the National Natural Science Foundation Key Project of China (grant no. 91954114 to H Xiao), the National Natural Science Foundation of China (grant no. 31871387

to H Xiao), the Innovative Research Team for the Central Universities (grant no. GK202001004 to H Xiao), and Natural Science Foundation Youth Project of Shaanxi Province (grant no. 2022JQ208 to Q Zheng).

## Additional information

### Funding

| Funder | Grant reference number | Author |
|---|---|---|
| National Natural Science Foundation of China | 91954114 | Hui Xiao |
| National Natural Science Foundation of China | 31871387 | Hui Xiao |
| the Innovative Research Team for the Central Universities | GK202001004 | Hui Xiao |
| Natural Science Foundation Youth Project of Shaanxi Province | 2022JQ208 | Qian Zheng |

The funders had no role in study design, data collection and interpretation, or the decision to submit the work for publication.

### Author contributions

Lei Yuan, Data curation, Software, Formal analysis, Validation, Methodology, Writing – original draft; Peiyao Li, Data curation, Validation, Methodology, Writing – original draft; Huiru Jing, Data curation, Software, Methodology; Qian Zheng, Data curation, Software, Funding acquisition, Methodology; Hui Xiao, Conceptualization, Resources, Data curation, Formal analysis, Supervision, Funding acquisition, Validation, Investigation, Visualization, Methodology, Writing – original draft, Writing – review and editing

### Author ORCIDs

Lei Yuan ⓘ http://orcid.org/0000-0001-9305-4486

### Decision letter and Author response

Decision letter https://doi.org/10.7554/eLife.76436.sa1
Author response https://doi.org/10.7554/eLife.76436.sa2

## Additional files

### Supplementary files

• Transparent reporting form

### Data availability

All data generated or analysed during this study are included in the manuscript and supporting file.

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
