## [Editor Report]

This article will be of high interest to scientists interested in phagocytosis, the process of removal and degradation of dead cells and pathogens. The authors identify multiple signaling components that affect the protein level of a critical phagocytosis receptor, which disrupts the degradation of dead cells. The data are extensive and overall support the conclusions of the article, providing new insight into the regulation of phagocytosis.

---

## [Decision Letter]

**Decision letter after peer review:**

Thank you for submitting your article "*trim-21* Promotes Proteasomal Degradation of CED-1 for Apoptotic Cell Clearance in *C. elegans*" for consideration by *eLife*. Your article has been reviewed by 2 peer reviewers, and the evaluation has been overseen by a Reviewing Editor and Suzanne Pfeffer as the Senior Editor. The reviewers have opted to remain anonymous.

The reviewers have discussed their reviews with one another, and the Reviewing Editor asks you to please respond to the reviewers comments with a revised manuscript that addresses the points raised by the reviewers. As you can see, the reviewers request some additional documentation and analysis. Reviewer #2 finds it to be particularly important show co-localization of CED-1 and VHA-10 on phagosomes in wild type, trim-21, and retromer/snx mutants.

*Reviewer #2 (Recommendations for the authors):*

Yuan and co-authors reported in this manuscript that the phagocytic receptor CED-1 undergoes proteasomal degradation mediated by TRIM-21 and UBC-21, which is important for proper degradation of cell corpses. The authors provided evidence that CED-1 is polyubiquitinated and undergoes proteasome-mediated degradation. The polyubiquitination of CED-1 requires TRIM-21 and UBC-21 functions, while CED-1 degradation also requires CED-6, SRC-1, and NCK-1 that mediate TRIM-21 binding to CED-1. The authors further identified VHA-10 as an interacting protein of CED-1 and concluded that excess CED-1 interferes with lysosome acidification by binding VHA-10. However, this conclusion is not supported by their data. The overall imaging quality needs to be improved and multiple assays lack quantification analyses.

1. It has been shown previously that CED-1 is recycled from phagosome membranes to plasma membranes via retromer complex, and failure in recycling leads to lysosomal degradation of CED-1 (Chen et al., Science 2010). Here, the authors proposed that CED-1 is degraded via proteasome after engulfment is initiated, presumably on forming phagosomes (Figure 8). What is the relationship of retromer-dependent recycling, lysosome-dependent and proteosome-dependent degradation of CED-1? If the non-recycled CED-1 can be degraded by lysosomes, how its persistence interferes with V-ATPase as proposed by the authors? This point should be clearly addressed to demonstrate the role of proteosome-dependent degradation of CED-1 in apoptotic cell clearance.

2. The authors concluded that "excess CED-1 which accumulated in trim-21 mutants, bound to VHA-10 subunit, negatively affects the acidification of lysosomes and consequently blocked cell corpse degradation". However, their data cannot support this conclusion.

i) vha-10 RNAi causes very mild cell corpse clearance defects (Figure 7). Thus, it is not surprising that vha-10 RNAi did not affect cell corpse phenotype of trim-21 (which is even milder).

ii) Given the extremely weak phenotype in both trim-21 and vha-10 RNAi worms, the bypass or rescue experiments, which is indicated by differences of 2-3 cell corpses in average and has no other controls, are not convincing.

iii) To support the conclusion, the authors should show co-localization of CED-1 and VHA-10 on phagosomes in wild type, trim-21, and retromer/snx mutants.

iv) There is no data to support CED-1-VHA-10 binding negatively affects acidification of lysosomes.

3. Both trim-21(xwh13) and ubc-21(xwh16) mutants show very mild cell corpse phenotypes (Figure 6A-C, Figure 6-supplement 1A-F, H-M), which questions the importance of their role in cell corpse clearance.

4. The phagosome maturation data, especially images in Figure 6D and E are unclear. It is very difficult to see cell corpse labeling by phagosomal reporters in either wild type or trim-21 mutants and therefore not possible to comment on the conclusions. Images that show clear cell corpse labeling by phagosomal markers should be provided.

The authors showed that treatment of CQ or MG132 led to increased protein levels of CED-1. Are these two effects additive, or affect each other? Is lysosomal degradation of CED-1 affected by trim-21? Is polyubiquitination of CED-1 by TRIM-21 involved in lysosomal degradation of CED-1, for example, promoting CED-1 sorting into ILVs followed by degradation in lysosomes? On the other hand, does loss of ESCRT, such as in vps-37 RNAi, which is shown to block lysosomal degradation of CED-1 (Chen et al., Science 2010), affect polyubiquitination of CED-1 or the CED-1 degradation via proteasomes? To demonstrate that CED-1 is indeed degraded by proteasomes in addition to lysosomal degradation during phagosome maturation, the authors should examine CED-1::GFP levels on plasma membranes, phagosomal and phagolysosomal membranes in trim-21, retromer or snx mutant and in the double mutants that lack both TRIM-21 and retromer/SNX.

Overexpression of CED-1 reduces cell corpses in snx-1 mutants that affect CED-1 recycling (Chen et al., Science 2010). Does overexpression of CED-1 enhance the cell corpse phenotype in trim-21 mutant if it disrupts CED-1 degradation by proteasome?

In Figure 6-supplement 1A, ubc-21 appears to have significantly more germ cell corpses than trim-21, especially at 48 h and 60 h post L4, which suggests that additional E3 may be involved or that UBC-21 has additional roles than acting with TRIM-21. The authors should clarify this.

In Figure 2E, multiple bands were observed in lane 2-4 when CED-1-CT, TRIM-21, or UBC-21 is absent, which is inconsistent with the statement that "polyubiquitination of CED-1-CT was completely abolished by the removal of either UBC-21 or TRIM-21". In fact, similar bands, albeit with lower intensity, were also seen in the lane which did not contain CED-1-CT-FLAG. The authors should clarify these results.

Other points

1. Molecular weight should be indicted in western blot analyses.

2. The data quality in Figure 2F, G is insufficient, especially the anti-FLAG blot, and these data lack quantification analyses. The K48 and K63 issue should also be tested in vitro.

3. Quantification analysis should be provided for western blot assays that examine CED-1 levels in different genetic backgrounds or treatments (quantification analyses are missing in multiple figures).

4. To demonstrate that CED-6 mediates TRIM-21 recruitment to apoptotic cell surface, the authors should examine and quantify recruitment of TRIM-21 to apoptotic cells in ced-6 mutants and compare it with the wild type.

5. Does Y1019F affect NCK-1 binding to CED-1? It is difficult to understand if NCK-1 only binds phosphorylated CED-1 (supposedly at YXXL motif), while loss of NCK-1, but not Y1019F of CED-1 (failed to be phosphorylated), affects TRIM-21 binding.

How do N962A and Y1019F affect CED-1 polyubiquitination/degradation?

*Reviewer #3 (Recommendations for the authors):*

Yuan et al., identify ubiquitination and proteasomal degradation as an important mechanism for maintaining proper levels of the phagocytic receptor, Ced-1. They use a combination of genetics and molecular analyses to reveal multiple components of this pathway including the relevant E2 ubiquitin conjugating enzyme and E3 ubiquitin ligase, domains on Ced-1 and the E3 ligase (Trim-21) that interact, and signaling components required for their interaction. They additionally find that Ced-1 physically interacts with the V-ATPase, and propose that a lack of Ced-1 protein turnover inhibits V-ATPase function, leading to a block in phagosome maturation and persistence of cell corpses. This is an ambitious paper with screens for E2 and E3 enzymes, kinases, SH2 domain proteins and Ced-1 interactors, and generation of multiple new alleles by CRISPR/Cas9. Overall it provides new insight into Ced-1 regulation that is critical for its proper function, although some of the data could be improved to fully support the conclusions.

This paper will be of broad interest to researchers working on phagocytosis. It provides new links between phagocytic receptors and phagosome maturation, and the authors demonstrate conservation of this pathway between *C. elegans* and mammals.

Strengths

– The data are extensive and quantification and statistical analysis of much of the data presented.

– There is strong evidence for many of the conclusions, such as Ced-1 being regulated by proteasomal degradation, Ced-1 and Ced-6 interacting physically with Trim-21, multiple ced mutants leading to increased levels of Ced-1, defects in phagosome maturation in Trim-21 mutants and Ced-1/V-ATPase interactions.

– The identification of new Ced-1 interactors, particularly Trim21 and Vha-10, is very important to the field.

Weaknesses

– Some of the data are not convincing as presented, and further analysis and controls should be provided. Specific examples:

– Some of the ubiquitin blots are over-exposed (e.g. Figure 3K) and the conclusions are not clearly supported. Providing molecular size markers or arrows would be helpful (e.g. Figure 1C).

– It does not appear that endogenous Ced-1 is increasing in the trim-21; smls110 lane of the CED-1 blot (Figure 1L). Levels also do not appear to change in the bcls39 strain so an explanation is necessary. (Molecular markers or arrows indicating endogenous Ced-1 would help visualize the changes).

– In Figure 3B, levels in Ced-6 alone should be provided for comparison. It is possible that the effects are additive.

– Figure 3F – the colocalization is not convincing at the subcellular level. Higher resolution microscopy should be performed and individual cell types should be indicated. It should be stated if these cells are performing phagocytosis.

– It should be clearly stated if alleles used are null alleles (Figure 4 G,H, 5A, J, 6C and others) as this is important for conclusions based on double mutant analysis.

– Some of the findings with Src-1 have been previously demonstrated in *Drosophila* (Ziegenfuss et al. 2008) and mammals (Scheib et al. 2012) and the authors should clearly state what is consistent with those findings and what is new or different.

Below are some specific suggestions in addition to those provided above.

– Starting the results off with the findings on STAT does not fit well with the focus of the paper. It could be mentioned later when the kinase screen is described.

– The effects in Figure 1A (MG132) are much stronger than 1B (uba-1, ubq-2) – some interpretation should be provided.

– Figure 1C – it should be explained why the two versions of Ced-1::Flag are different sizes.

– The bands in supplemental figure 3D are highly variable so it is difficult to assess the levels. Consider repeating and quantifying this experiment.

– Figure 3F- it would also be interesting to determine if the colocalization of Ced-1 and Trim21 fluorescence was dependent on Ced6.

– Some discussion of how else src-1 and nck-1 might function in parallel to the two ced pathways would be of interest.

[Editors’ note: further revisions were suggested prior to acceptance, as described below.]

Thank you for resubmitting your work entitled "*trim-21* Promotes Proteasomal Degradation of CED-1 for Apoptotic Cell Clearance in *C. elegans*" for further consideration by *eLife*. Your revised article has been evaluated by Suzanne Pfeffer (Senior Editor) and a Reviewing Editor.

The manuscript has been much improved but there are some remaining issues, raised by Reviewer 2, that need to be addressed. None of the suggested experiments are required but textual changes that address these points are requested.

*Reviewer #2 (Recommendations for the authors):*

The revised manuscript has been improved substantially. The authors provided much more clear imaging and quantification analyses. They have addressed most of my concerns except for the following points that need to be further clarified.

The authors showed that:

1. "Recruitment and release of CED-1::GFP to (and from?) cell corpses in trim-21(8.29 ± 0.48 min) was similar to that in the WT (8.31 ± 0.25 min), but not to that of snx-1(>30 min) (Figure 7—figure supplement 1A). Moreover, "The CED-1::GFP levels in the trim-21 mutants were comparable to those on N2 in plasma membranes, phagosomal membranes, and phagolysosomal membranes, whereas snx-1 mutants and trim-21;snx-1 double mutants exhibited lower CED-1::GFP levels (Figure 7—figure supplemental 1B-E)"

These results and the interpretations need to be clarified or explained. Based on the provided data and the model presented in Figure 8, the ubiquitination of CED-1 by TRIM-21 occurs on forming phagosomes. If it fails, the transmembrane receptor CED-1 is presumably accumulated on the phagosomal surface. However, the authors showed that the release of CED-1 from cell corpse-containing phagosomes and the level of CED-1 on phagosomes and phagolysosomes were not altered in trim-21 mutants, which is probably due to the normal function of retromer-mediated recycling in trim-21 mutants. If this is the case, where does the excessive CED-1, which is a transmembrane receptor, localize in trim-21 mutants? The author showed that both CED-1-VHA-10 co-localization and interaction were increased in trim-21 mutants. Where does the enhanced interaction occur given that CED-1 levels on phagosomal and phagolysosomal membranes are unaltered? Moreover, as shown in Figure 7-supplement 2A and B, CED-1::GFP and VHA-10::mCherry co-localized on the phagosomal surface at a high level in both trim-21 and snx-1 mutants. What is the explanation for enhanced co-localization of them on the phagosomal surface where CED-1 levels are reduced in snx-1 and unaltered in trim-21?

2. The authors state that "Thus, excess CED-1, which accumulated in trim-21 mutants, bound to the VHA-10 subunit of V-ATPase, negatively affected the acidification of lysosomes, and consequently blocked cell corpse degradation."

By AO staining, the authors showed that phagosomal acidification is affected in trim-21 mutants and vha-10 RNAi worms, but they did not examine the acidification of lysosomes in either case. How is the conclusion of "negatively affected the acidification of lysosomes" supported? How did excessive CED-1 in trim-21 access to VHA-10 on lysosomes as indicated in the model (Figure 8)? This is also in contrast to the subcellular localization of the two proteins shown in Figure 7-supplement 2A, which is clearly not on the lysosome.

These points should be clarified. If the authors claim that "excess CED-1, which accumulated in trim-21 mutants, bound to the VHA-10 subunit of V-ATPase, negatively affected the acidification of lysosomes, and consequently blocked cell corpse degradation." They should provide direct evidence that lysosomal acidification is affected.

3. In the discussion part, the authors state that "we proposed that TRIM-21-mediated degradation of CED-1 could be effective in promoting the early stages of phagosome maturation, whereas retromer-mediated recycling of CED-1 from the phagosome surface back to the plasma membrane could be effective in the late stages."

This statement is also confusing. What is the early stage of phagosome maturation promoted by CED-1 degradation as the authors observed a late effect on phagolysosomal formation in trim-21 mutants? What are the late stages that are promoted by retromer-mediated recycling as effective recycling of the CED-1 receptor contributes to efficient engulfment of cell corpses upstream of phagosome maturation?

*Reviewer #3 (Recommendations for the authors):*

The authors have been very responsive to the previous reviews. I have no additional suggestions - it is an excellent contribution to the field.

---

## [Author Response]

Reviewer #2 (Recommendations for the authors):Yuan and co-authors reported in this manuscript that the phagocytic receptor CED-1 undergoes proteasomal degradation mediated by TRIM-21 and UBC-21, which is important for proper degradation of cell corpses. The authors provided evidence that CED-1 is polyubiquitinated and undergoes proteasome-mediated degradation. The polyubiquitination of CED-1 requires TRIM-21 and UBC-21 functions, while CED-1 degradation also requires CED-6, SRC-1, and NCK-1 that mediate TRIM-21 binding to CED-1. The authors further identified VHA-10 as an interacting protein of CED-1 and concluded that excess CED-1 interferes with lysosome acidification by binding VHA-10. However, this conclusion is not supported by their data. The overall imaging quality needs to be improved and multiple assays lack quantification analyses.1. It has been shown previously that CED-1 is recycled from phagosome membranes to plasma membranes via retromer complex, and failure in recycling leads to lysosomal degradation of CED-1 (Chen et al., Science 2010). Here, the authors proposed that CED-1 is degraded via proteasome after engulfment is initiated, presumably on forming phagosomes (Figure 8). What is the relationship of retromer-dependent recycling, lysosome-dependent and proteosome-dependent degradation of CED-1? If the non-recycled CED-1 can be degraded by lysosomes, how its persistence interferes with V-ATPase as proposed by the authors? This point should be clearly addressed to demonstrate the role of proteosome-dependent degradation of CED-1 in apoptotic cell clearance.

We thank the reviewer for these very insightful suggestions. To investigate the relationship between retromer-dependent recycling and lysosome- and proteosome-dependent degradation of CED-1, we treated N2 with MG-132 and CQ, and found that the effects were additive (please see, page 7; line 139-146; Figure 1—figure supplement 1B). Following treatment of both N2 and *trim-21* mutants with *vps-37* RNAi, the protein levels of endogenous CED-1 were significantly increased compared to those in controls (please see, page 12; line 243-250; Figure 1M and N). The CED-1 polyubiquitination was unchanged by *vps-37* RNAi treatment in *ced-1::flag; ha::ubq-2* worms (please see, page 14; line 300-304; Figure 2I and J). Additionally, the absence of *trim-21* had no effect on CED-1 recycling (please see, page 33; line 710-715; Figure 7—figure supplement 1A). We next confirmed that CED-1 is indeed degraded by proteasomes and undergoes lysosomal degradation during phagosome maturation. We found that CED-1::GFP levels in *trim-21* mutants were comparable to those of N2 in the plasma, phagosomal, and phagolysosomal membranes, whereas the *snx-1* mutants and *trim-21; snx-1* double mutants showed lower CED-1::GFP levels (please see, page 33-34; line 717-727; Figure 7—figure supplement 1B–E). We also found that overexpression of CED-1 enhanced the cell corpse phenotype in *trim-21* mutants (please see, page 28; line 596-598; Figure 6—figure supplement 1H). Furthermore, the interaction between CED-1 and VHA-10 was enhanced only in the *trim-21* mutant and not in the wild-type and *snx-1* mutant (please see, page 34; line 730-738; Figure 7—figure supplement 2C–E), indicating that non-recycled CED-1 degraded by lysosomes differed from CED-1 influenced by *trim-21*-mediated proteasomal degradation. Therefore, *trim-21*-mediated proteasomal degradation of CED-1 occurred independently of retromer-dependent recycling and lysosome-dependent degradation of CED-1.

2. The authors concluded that "excess CED-1 which accumulated in trim-21 mutants, bound to VHA-10 subunit, negatively affects the acidification of lysosomes and consequently blocked cell corpse degradation". However, their data cannot support this conclusion.i) vha-10 RNAi causes very mild cell corpse clearance defects (Figure 7). Thus, it is not surprising that vha-10 RNAi did not affect cell corpse phenotype of trim-21 (which is even milder).ii) Given the extremely weak phenotype in both trim-21 and vha-10 RNAi worms, the bypass or rescue experiments, which is indicated by differences of 2-3 cell corpses in average and has no other controls, are not convincing.iii) To support the conclusion, the authors should show co-localization of CED-1 and VHA-10 on phagosomes in wild type, trim-21, and retromer/snx mutants.iv) There is no data to support CED-1-VHA-10 binding negatively affects acidification of lysosomes.

The reviewer is absolutely right in pointing out that our data does not support the conclusion "excess CED-1 which accumulated in *trim-21* mutants, bound to VHA-10 subunit, negatively affects the acidification of lysosomes and consequently blocked cell corpse degradation". *vha-10* RNAi causes very mild cell corpse clearance defects, possibly through partial (knockdown) silencing of *vha-10*. To support the conclusion, we first used CRISPR-Cas9 to generate *vha-10* knockout worms and counted the number of cell corpses to determine whether the number was increased compared to that obtained using *vha-10* RNAi; however, the loss-of-function allele of *vha-10* led to embryonic lethality. We found that VHA-10 overexpression completely restored the acidification of phagosomes defects in *trim-21(xwh13)* mutants (please see, page 32-33; line 703-707; Figure 7J). In addition, co-localization of CED-1 and VHA-10 on phagosomes was significantly higher in *trim-21* and *snx-1* mutants than in the wild-type (please see, page 34; line 727-730; Figure 7—figure supplement 2A–B). The interaction between CED-1 and VHA-10 was enhanced only in the *trim-21* mutant and not in the wild-type and *snx-1* mutant (please see, page 34; line 730-738; Figure 7—figure supplement 2C–E). These results suggest that CED-1-VHA-10 binding negatively affects the acidification of lysosomes.

3. Both trim-21(xwh13) and ubc-21(xwh16) mutants show very mild cell corpse phenotypes (Figure 6A-C, Figure 6-supplement 1A-F, H-M), which questions the importance of their role in cell corpse clearance.

We thank the reviewer for this very helpful suggestion. The phosphatidylserine receptor PSR-1 plays an important role in recognizing phosphatidylserine during phagocytosis (Science. 2003 Nov 28; 302(5650): 1563-6.); however, the cell corpse phenotype of the *psr-1* mutant was relatively weak, indicating that additional engulfment receptors act upstream of CED-2/CED-5/CED-12 for cell corpse removal in *C. elegans*. Indeed, integrin INA-1 has been proposed to recognize and bind to PS (Curr Biol. 2010 Mar 23;20(6):477-86.). *trim-21* and *ubc-21* exhibited a relatively weak phenotype, suggesting that additional E3 ligases exist for proteasomal degradation of CED-1. Another possibility is that accumulation of CED-1 caused by *trim-21* or *ubc-21* partially affects the function of VHA-10 and does not cause complete loss of VHA-10 function, causing the phenotype to be weaker. Such weak phenotypes also suggest that appropriate levels of CED-1 for executing engulfment function were maintained through a precise mechanism, demonstrating their important roles in clearing apoptotic cells.

4. The phagosome maturation data, especially images in Figure 6D and E are unclear. It is very difficult to see cell corpse labeling by phagosomal reporters in either wild type or trim-21 mutants and therefore not possible to comment on the conclusions. Images that show clear cell corpse labeling by phagosomal markers should be provided.

We apologize for the unclear images of the clear cell corpse labeling by phagosomal markers. We have replaced the original images with clearer and magnified images of cell corpse labeling by phagosomal markers (please see, Figure 6D and E).

The authors showed that treatment of CQ or MG132 led to increased protein levels of CED-1. Are these two effects additive, or affect each other? Is lysosomal degradation of CED-1 affected by trim-21? Is polyubiquitination of CED-1 by TRIM-21 involved in lysosomal degradation of CED-1, for example, promoting CED-1 sorting into ILVs followed by degradation in lysosomes? On the other hand, does loss of ESCRT, such as in vps-37 RNAi, which is shown to block lysosomal degradation of CED-1 (Chen et al., Science 2010), affect polyubiquitination of CED-1 or the CED-1 degradation via proteasomes? To demonstrate that CED-1 is indeed degraded by proteasomes in addition to lysosomal degradation during phagosome maturation, the authors should examine CED-1::GFP levels on plasma membranes, phagosomal and phagolysosomal membranes in trim-21, retromer or snx mutant and in the double mutants that lack both TRIM-21 and retromer/SNX.Overexpression of CED-1 reduces cell corpses in snx-1 mutants that affect CED-1 recycling (Chen et al., Science 2010). Does overexpression of CED-1 enhance the cell corpse phenotype in trim-21 mutant if it disrupts CED-1 degradation by proteasome?

We thank the reviewer for these very insightful suggestions. To investigate the relationship between retromer-dependent recycling, lysosome- and proteosome-dependent degradation of CED-1, We treated N2 N2 with MG-132 and CQ, and found that the effects were additive (please see, page 7; line 139-146; Figure 1–figure supplement 1B). Following treatment of both N2 and trim-21 mutants with vps-37 RNAi, the protein levels of endogenous CED-1 were significantly increased compared to those in controls (please see, page 12; line 243-250; Figure 1M and N). The CED-1 polyubiquitination was unchanged by vps-37 RNAi treatment in ced-1::flag; ha::ubq-2 worms (please see, page 14; line 300-304; Figure 2I and J). Additionally, the absence of trim-21 had no effect on CED-1 recycling (please see, page 33; line 710-715; Figure 7–figure supplement 1A). We next conformed that CED-1 is indeed degraded by proteasomes and undergoes lysosomal degradation during phagosome maturation. We found that CED-1::GFP levels intrim-21 mutants were comparable to those of N2 in the plasma, phagosomal, and phagolysosomal membranes, whereas the snx-1 mutants and trim-21; snx-1 double mutants showed lower CED-1::GFP levels (please see, page 33-34; line 717-727; Figure 7–figure supplement 1B-E). We also found that overexpression of CED-1 enhanced the cell corpse phenotype in trim-21 mutants (please see, page 28; line 596-598; Figure 6–figure supplement 1H). Furthermore, the interaction between CED-1 and VHA-10 was enhanced only in the trim-21 mutant and not in the wild-type and snx-1 mutant (please see, page 34; line 730-738; Figure 7–figure supplement 2C–E), indicating that non-recycled CED-1 degraded by lysosomes differed from CED-1 influenced by trim-21-mediated proteasomal degradation. Therefore, trim-21-mediated proteasomal degradation of CED-1 occurred independently of retromer-dependent recycling and lysosome-dependent degradation of CED-1.

In Figure 6-supplement 1A, ubc-21 appears to have significantly more germ cell corpses than trim-21, especially at 48 h and 60 h post L4, which suggests that additional E3 may be involved or that UBC-21 has additional roles than acting with TRIM-21. The authors should clarify this.

We thank the reviewer for this very helpful suggestion. The phosphatidylserine receptor PSR-1 plays an important role in recognizing phosphatidylserine during phagocytosis (Science. 2003 Nov 28; 302(5650): 1563-6.); however, the cell corpse phenotype of the psr-1 mutant was relatively weak, indicating that additional engulfment receptors act upstream of CED-2/CED-5/CED-12 for cell corpse removal in *C. elegans*. Indeed, integrin INA-1 has been proposed to recognize and bind to PS (Curr Biol. 2010 Mar 23;20(6):477-86.). trim-21 and ubc-21 exhibited a relatively weak phenotype, suggesting that additional E3 ligases exist for proteasomal degradation of CED-1. Another possibility is that accumulation of CED-1 caused by trim-21 or ubc-21 partially affects the function of VHA-10 and does not cause complete loss of VHA-10 function, causing the phenotype to be weaker. Such weak phenotypes also suggest that appropriate levels of CED-1 for executing engulfment function were maintained through a precise mechanism, demonstrating their important roles in clearing apoptotic cells. Ubc-21 appeared to have significantly more germ cell corpses than trim-21, which has been explained in the revised manuscript (please see, page 26-27; line 572-575).

Other points1. Molecular weight should be indicted in western blot analyses.

We thank the reviewer for pointing out our oversight. We have indicated the molecular weights in the western blotting figures.

2. The data quality in Figure 2F, G is insufficient, especially the anti-FLAG blot, and these data lack quantification analyses. The K48 and K63 issue should also be tested in vitro.

We apologize for the confusion. We repeated the ubiquitination assay and validated the results of the K48- and K63-linked ubiquitination assay in vitro (please see, page 14; line 291-294; Figure 2E). We repeated and quantified the data shown in Figure 2F, G (please see, Figure 2F–H).

3. Quantification analysis should be provided for western blot assays that examine CED-1 levels in different genetic backgrounds or treatments (quantification analyses are missing in multiple figures).

We appreciate this suggestion. We quantified the results of western blotting performed to determine CED-1 levels in different genetic backgrounds or treatments.

4. To demonstrate that CED-6 mediates TRIM-21 recruitment to apoptotic cell surface, the authors should examine and quantify recruitment of TRIM-21 to apoptotic cells in ced-6 mutants and compare it with the wild type.

We thank the reviewer for this very helpful suggestion. We examined and quantified the recruitment of TRIM-21 to apoptotic cells in *ced-6* mutants and compared the value with that in the wild-type (please see, page 17-18; line 373-377; Figure 3—figure supplement 1G).

5. Does Y1019F affect NCK-1 binding to CED-1? It is difficult to understand if NCK-1 only binds phosphorylated CED-1 (supposedly at YXXL motif), while loss of NCK-1, but not Y1019F of CED-1 (failed to be phosphorylated), affects TRIM-21 binding.How do N962A and Y1019F affect CED-1 polyubiquitination/degradation?

We totally agree with the reviewer that it is unclear whether Y1019F affects NCK-1 binding to CED-1. We found that Y1019F of CED-1 was necessary for maintaining its interaction with NCK-1 (please see, page 26; line 556-562; Figure 5K). These results suggest that N962A affected the ubiquitination and degradation of CED-1 by affecting the interaction between CED-1 and CED-6, whereas Y1019F affected the ubiquitination and degradation of CED-1 by affecting the phosphorylation of CED-1 and thus the interaction between CED-1 and NCK-1.

Reviewer #3 (Recommendations for the authors):Yuan et al., identify ubiquitination and proteasomal degradation as an important mechanism for maintaining proper levels of the phagocytic receptor, Ced-1. They use a combination of genetics and molecular analyses to reveal multiple components of this pathway including the relevant E2 ubiquitin conjugating enzyme and E3 ubiquitin ligase, domains on Ced-1 and the E3 ligase (Trim-21) that interact, and signaling components required for their interaction. They additionally find that Ced-1 physically interacts with the V-ATPase, and propose that a lack of Ced-1 protein turnover inhibits V-ATPase function, leading to a block in phagosome maturation and persistence of cell corpses. This is an ambitious paper with screens for E2 and E3 enzymes, kinases, SH2 domain proteins and Ced-1 interactors, and generation of multiple new alleles by CRISPR/Cas9. Overall it provides new insight into Ced-1 regulation that is critical for its proper function, although some of the data could be improved to fully support the conclusions.This paper will be of broad interest to researchers working on phagocytosis. It provides new links between phagocytic receptors and phagosome maturation, and the authors demonstrate conservation of this pathway between *C. elegans* and mammals.Strengths– The data are extensive and quantification and statistical analysis of much of the data presented.– There is strong evidence for many of the conclusions, such as Ced-1 being regulated by proteasomal degradation, Ced-1 and Ced-6 interacting physically with Trim-21, multiple ced mutants leading to increased levels of Ced-1, defects in phagosome maturation in Trim-21 mutants and Ced-1/V-ATPase interactions.– The identification of new Ced-1 interactors, particularly Trim21 and Vha-10, is very important to the field.Weaknesses– Some of the data are not convincing as presented, and further analysis and controls should be provided. Specific examples:– Some of the ubiquitin blots are over-exposed (e.g. Figure 3K) and the conclusions are not clearly supported. Providing molecular size markers or arrows would be helpful (e.g. Figure 1C).

This is a very good point. We have repeated the experiment (please see, Figure 3L) and indicated the molecular size in all figures.

– It does not appear that endogenous Ced-1 is increasing in the trim-21; smls110 lane of the CED-1 blot (Figure 1L). Levels also do not appear to change in the bcls39 strain so an explanation is necessary. (Molecular markers or arrows indicating endogenous Ced-1 would help visualize the changes).

We thank the reviewer for this very helpful suggestion. This can be explained by the fact that the three transgenic integrated strains *smIs34*, *smIs110*, and *bcIs39* contained different copy counts of extrachromosomal arrays integrated into their chromosomes. As a result, the levels of CED-1 overexpression in *smIs34* and *bcIs39* varied. In *bcIs39*, exogenous CED-1 may be expressed at a higher level compared to that in *smIs34*, possibly masking TRIM-21's ability to degrade endogenous CED-1. In smIs110, CED-1ΔC::GFP (a GFP-fused CED-1 with the C-terminal region deleted) is still capable of recognizing neighboring apoptotic cells, suggesting that it competes with endogenous CED-1 for signal transduction, including recruiting TRIM-21 for its degradation. We have explained these points in the manuscript (please see, page 11-12; line 224-243) and indicated the molecular size in all figures.

– In Figure 3B, levels in Ced-6 alone should be provided for comparison. It is possible that the effects are additive.

We thank the reviewer this helpful suggestion. We have indicated the protein levels of endogenous CED-1 in the *ced-6* mutants alone and quantified endogenous CED-1 protein levels (please see, Figure 3B). These effects did not appear to be additive.

– Figure 3F – the colocalization is not convincing at the subcellular level. Higher resolution microscopy should be performed and individual cell types should be indicated. It should be stated if these cells are performing phagocytosis.

We thank the reviewer for this very good suggestion. We attempted to perform imaging using confocal microscopy but were unable to obtain a clear image because the expression was relatively weak and fluorescence was quickly quenched. We have replaced the original images with clearer images (please see, Figure 3F). *C. elegans* lacks the professional phagocytes, instead apoptotic cells are engulfed by many neighboring cell types, and apoptotic cells appeared as “button-like” under DIC microscopy, as indicated by the arrow in Figure 3F, and apoptotic cells surrounded by CED-1 or CED-6 were those recognized and internalized by phagocytes.

– It should be clearly stated if alleles used are null alleles (Figure 4 G,H, 5A, J, 6C and others) as this is important for conclusions based on double mutant analysis.

We agree with the reviewer and stated the alleles of the mutants in the revised manuscript (please see, Figure 3-Figure legend, Figure 4G and H-Figure legend, Figure 5A and J-Figure legend, Figure 5—figure supplement 1B-Figure legend, Figure 6C-Figure legend, Figure 7—figure supplement 1D-Figure legend, and Figure 7—figure supplement 2D-Figure legend).

– Some of the findings with Src-1 have been previously demonstrated in *Drosophila* (Ziegenfuss et al. 2008) and mammals (Scheib et al. 2012) and the authors should clearly state what is consistent with those findings and what is new or different.

We appreciate the reviewer for pointing out our Discussion should be expanded. We have revised the Discussion as recommended (please see, page 35-36; line 762-773).

Below are some specific suggestions in addition to those provided above.– Starting the results off with the findings on STAT does not fit well with the focus of the paper. It could be mentioned later when the kinase screen is described.

We thank the reviewer for this insightful suggestion. We have modified the manuscript by moving the findings on STAT to the kinase screen section (please see, page 21; line 441-443, 452-461; Figure 4—figure supplement 1C–E).

– The effects in Figure 1A (MG132) are much stronger than 1B (uba-1, ubq-2) – some interpretation should be provided.

We thank the reviewer for this very helpful suggestion. We have provided interpretation, stating that “The effects shown in Figure 1A (MG132) were much stronger than those shown in 1B (*uba-1, ubq-2*)” (please see, page 7; line 135-139).

– Figure 1C – it should be explained why the two versions of Ced-1::Flag are different sizes.

We apologize for the unclear description of our result. CED-1 has different isoforms, which has been explained this the revised manuscript (please see, Figure 1C-Figure legend).

– The bands in supplemental figure 3D are highly variable so it is difficult to assess the levels. Consider repeating and quantifying this experiment.

We agree with the reviewer. We have repeated the experiment and quantified the results (please see, Figure 3—figure supplement 1D).

– Figure 3F- it would also be interesting to determine if the colocalization of Ced-1 and Trim21 fluorescence was dependent on Ced6.

We thank the reviewer for this very helpful suggestion. We have examined and quantified the colocalization of CED-1 and TRIM-21 fluorescence in *ced-6* mutants and compare the values with those in the wild-type (please see, page 18; line 378-380; Figure 3F and G).

– Some discussion of how else src-1 and nck-1 might function in parallel to the two ced pathways would be of interest.

We appreciate the reviewer for suggesting that the Discussion should be expanded. We have revised the Discussion as recommended (please see, page 35-36; line 762-773).

[Editors' note: further revisions were suggested prior to acceptance, as described below.]

Reviewer #2 (Recommendations for the authors):The revised manuscript has been improved substantially. The authors provided much more clear imaging and quantification analyses. They have addressed most of my concerns except for the following points that need to be further clarified.The authors showed that:1. "Recruitment and release of CED-1::GFP to (and from?) cell corpses in trim-21(8.29 ± 0.48 min) was similar to that in the WT (8.31 ± 0.25 min), but not to that of snx-1(>30 min) (Figure 7—figure supplement 1A). Moreover, "The CED-1::GFP levels in the trim-21 mutants were comparable to those on N2 in plasma membranes, phagosomal membranes, and phagolysosomal membranes, whereas snx-1 mutants and trim-21;snx-1 double mutants exhibited lower CED-1::GFP levels (Figure 7—figure supplemental 1B-E)"These results and the interpretations need to be clarified or explained. Based on the provided data and the model presented in Figure 8, the ubiquitination of CED-1 by TRIM-21 occurs on forming phagosomes. If it fails, the transmembrane receptor CED-1 is presumably accumulated on the phagosomal surface. However, the authors showed that the release of CED-1 from cell corpse-containing phagosomes and the level of CED-1 on phagosomes and phagolysosomes were not altered in trim-21 mutants, which is probably due to the normal function of retromer-mediated recycling in trim-21 mutants. If this is the case, where does the excessive CED-1, which is a transmembrane receptor, localize in trim-21 mutants? The author showed that both CED-1-VHA-10 co-localization and interaction were increased in trim-21 mutants. Where does the enhanced interaction occur given that CED-1 levels on phagosomal and phagolysosomal membranes are unaltered? Moreover, as shown in Figure 7-supplement 2A and B, CED-1::GFP and VHA-10::mCherry co-localized on the phagosomal surface at a high level in both trim-21 and snx-1 mutants. What is the explanation for enhanced co-localization of them on the phagosomal surface where CED-1 levels are reduced in snx-1 and unaltered in trim-21?

We thank the reviewer for these very insightful thoughts. A major concern is that the level of CED-1 on phagosomes and phagolysosomes were not altered in *trim-21* mutants, whereas the co-localization and interaction of CED-1-VHA-10 were enhanced. A possible explanation for this might be that the level of excessive CED-1, which is localized on the individual phagosomal surface in each phagocyte and goes through the dynamic phagosome maturation process. As a result, a substantial difference of CED-1 localized to the phagosomes and phagolysosomes between *trim-21* mutants and WT worms cannot be captured by the fluorescent image quantification method. Although levels of CED-1 and interaction of CED-1-VHA-10 were increased in *trim-21* mutants lysis than in WT controls by the western blotting detection method, this reflects the change of a large number of phagosomes from massive amounts of phagocytes. Another possible explanation is that the excessive CED-1 in *trim-21* mutants accumulated on the phagosomal surface affects multiple stages of phagosome maturation process, which slows down the whole maturation process rather than a specific step, undetectable at the stage that we captured. As the mechanism determining which portions of CED-1 are ubiquitinated by TRIM-21 is unknown, we cannot distinguish portion of CED-1 regulated by TRIM-21 from the portion of CED-1 regulated by other mechanisms in *trim-21* mutants, making it difficult to follow where the excessive CED-1 is localized in *trim-21* mutants (please see, page 34-35; line 740-766).

We also found enhanced co-localization of CED-1 and VHA-10 on the phagosomal surface in both trim-21 and snx-1 mutants (Figure 7—figure supplement 2A–C); however, the CED-1 levels on the phagosomal surface are reduced in snx-1 and unaltered in trim-21 (Figure 7—figure supplement 1D–E). Given that V-type ATPases are trafficked to the phagosome and function to acidify its contents during the phagosome maturation process (Kinchen and Ravichandran, 2008), one possible explanation for these results is that CED-1 failed to recycle from phagosomes and cytosol back to the plasma membrane in snx-1 mutants, resulting in the enhanced co-localization of CED-1 and VHA-10 on the phagosomal surface despite no increased interaction between CED-1 and VHA-10. Unlike in snx-1 mutants, the enhanced co-localization of CED-1 and VHA-10 on the phagosomal surface is at least partially due to increased interaction between CED-1 and VHA-10 in trim-21 mutants. Another possible explanation is that the amount of excessive CED-1 degraded by TRIM-21 might be less than the amount of CED-1 recycled by retromer, as trim-21; snx-1 double mutants exhibited lower CED-1 levels (Figure 7—figure supplement 1B–E), which requires further investigation (please see, page 39-40; line 847-867; Figure 8).

2. The authors state that "Thus, excess CED-1, which accumulated in trim-21 mutants, bound to the VHA-10 subunit of V-ATPase, negatively affected the acidification of lysosomes, and consequently blocked cell corpse degradation."By AO staining, the authors showed that phagosomal acidification is affected in trim-21 mutants and vha-10 RNAi worms, but they did not examine the acidification of lysosomes in either case. How is the conclusion of "negatively affected the acidification of lysosomes" supported? How did excessive CED-1 in trim-21 access to VHA-10 on lysosomes as indicated in the model (Figure 8)? This is also in contrast to the subcellular localization of the two proteins shown in Figure 7-supplement 2A, which is clearly not on the lysosome.These points should be clarified. If the authors claim that "excess CED-1, which accumulated in trim-21 mutants, bound to the VHA-10 subunit of V-ATPase, negatively affected the acidification of lysosomes, and consequently blocked cell corpse degradation." They should provide direct evidence that lysosomal acidification is affected.

We thank the reviewer for this very helpful suggestion. A crucial step in phagosome maturation is the gradual acidification of the phagosomal lumen because an acidic environment promotes the activity of hydrolytic enzymes that degrade phagosomal contents. In *Caenorhabditis elegans*, cell corpse–containing phagosome acidification begins fairly early, with Rab5 positive early phagosomes staining weakly with acridine orange (AO), indicating that there are multiple modes of acidification depending on the stage of maturation (Kinchen et al., 2008). Later study revealed that RAB-2 and RAB-14 act partially redundantly to promote phagosome acidification and recruit lysosomes for phagolysosome formation for cell corpse degradation, whereas RAB-7 mediates fusion of lysosomes to phagosomes but is largely dispensable for the acidification of phagosomes in *C. elegans*, indicating that acidification of cell corpse–containing phagosomes does not appear to be dependent on efficient phagosome–lysosome fusion (Guo, Hu, Zhang, Jiang, and Wang, 2010; Lu and Zhou, 2012; Wang and Yang, 2016; Yu, Lu, and Zhou, 2008). However, the mechanism by which V-type ATPases regulate acidification of phagosomes containing apoptotic cells or apoptotic cell degradation has not been thoroughly investigated. We discovered that excessive CED-1 binding to the V-ATPase in *trim-21* mutant worms reduces acidification of cell corpse–containing phagosomes. However, future research is needed to determine the precise stage at which phagosome acidification is affected, as well as how excessive CED-1 binding to VHA-10 in *trim-21* mutants affects cell corpse degradation. We completely agree with the reviewer that we did not provide direct evidence that lysosomal acidification is affected. We revised this sentence in accordance with the reviewer's suggestion (please see, page 35-36; line 767-792; Figure 8).

3. In the discussion part, the authors state that "we proposed that TRIM-21-mediated degradation of CED-1 could be effective in promoting the early stages of phagosome maturation, whereas retromer-mediated recycling of CED-1 from the phagosome surface back to the plasma membrane could be effective in the late stages."This statement is also confusing. What is the early stage of phagosome maturation promoted by CED-1 degradation as the authors observed a late effect on phagolysosomal formation in trim-21 mutants? What are the late stages that are promoted by retromer-mediated recycling as effective recycling of the CED-1 receptor contributes to efficient engulfment of cell corpses upstream of phagosome maturation?

We apologize for the confusion. We rewrote the statement. We propose that TRIM-21 functions downstream of CED-6 and NCK-1 to mediate CED-1 degradation, preventing excessive CED-1 from affecting phagosome acidification and maturation, whereas retromer-mediated effective recycling of CED-1 contributes to efficient engulfment of cell corpses (please see, page 40; line 867-872).

References

Guo, P., Hu, T., Zhang, J., Jiang, S., and Wang, X. (2010). Sequential action of *Caenorhabditis elegans* Rab GTPases regulates phagolysosome formation during apoptotic cell degradation. *Proc Natl Acad Sci U S A, 107*(42), 18016-18021. doi:10.1073/pnas.1008946107

Kinchen, J. M., Doukoumetzidis, K., Almendinger, J., Stergiou, L., Tosello-Trampont, A., Sifri, C. D.,... Ravichandran, K. S. (2008). A pathway for phagosome maturation during engulfment of apoptotic cells. *Nat Cell Biol, 10*(5), 556-566. doi:10.1038/ncb1718

Kinchen, J. M., and Ravichandran, K. S. (2008). Phagosome maturation: going through the acid test. *Nat Rev Mol Cell Biol, 9*(10), 781-795. doi:10.1038/nrm2515

Lu, N., and Zhou, Z. (2012). Membrane trafficking and phagosome maturation during the clearance of apoptotic cells. *Int Rev Cell Mol Biol, 293*, 269-309. doi:10.1016/B978-0-12-394304-0.00013-0

Wang, X., and Yang, C. (2016). Programmed cell death and clearance of cell corpses in *Caenorhabditis elegans*. *Cell Mol Life Sci, 73*(11-12), 2221-2236. doi:10.1007/s00018-016-2196-z

Yu, X., Lu, N., and Zhou, Z. (2008). Phagocytic receptor CED-1 initiates a signaling pathway for degrading engulfed apoptotic cells. *PLoS Biol, 6*(3), e61. doi:10.1371/journal.pbio.0060061